# Haploblocks contribute to parallel climate adaptation following global invasion of a cosmopolitan plant

The role of rapid adaptation during species invasions has historically been minimized with the assumption that introductions consist of few colonists and limited genetic diversity. While overwhelming evidence suggests that rapid adaptation is more prevalent than originally assumed, the demographic and adaptive processes underlying successful invasions remain unresolved. Here we leverage a large whole-genome sequence dataset to investigate the relative roles of colonization history and adaptation during the worldwide invasion of the forage crop, *Trifolium repens* (Fabaceae). We show that introduced populations encompass high levels of genetic variation with little evidence of bottlenecks. Independent colonization histories on different continents are evident from genome-wide population structure. Five haploblocks—large haplotypes with limited recombination—on three chromosomes exist as standing genetic variation within the native and introduced ranges and exhibit strong signatures of parallel climate-associated adaptation across continents. Field experiments in the native and introduced ranges demonstrate that three of the haploblocks strongly affect fitness and exhibit patterns of selection consistent with local adaptation across each range. Our results provide strong evidence that large-effect structural variants contribute substantially to rapid and parallel adaptation of an introduced species throughout the world.

Invasive species threaten ecosystems, agriculture, health and culture. The cost of controlling the spread of these species is immense[1–3], averaging US$26.8 billion per year globally. Yet, why certain introduced species become invasive is unclear. Despite substantial effort, research has identified few consistent predictors of invasion[4–8]. The roles that introduction history and evolutionary processes such as natural selection play in invasions have historically been neglected[9–12]. However, recent literature stemming from large-scale experiments and the genomic revolution suggests that rapid evolution may shape invasion success—particularly in species that have been widely introduced and represent important components of ecosystems across the globe[13–17].

An early assumption in invasion biology posited that introductions involved severe bottlenecks that purge genetic variation and constrain adaptation[10,18–20]. However, many invasions do not fit this classic expectation[21,22], especially for human-associated species that are repeatedly introduced. Repeated introductions and admixture between divergent genotypes can even increase genetic diversity in the introduced range[22,23]. Natural selection and rapid adaptation have also been increasingly documented across invasive species[14,16–18,24,25]. This paradigm shift leads to the questions that we address in this study. Specifically, how do introduction history and admixture shape population structure during invasions? What is the genetic architecture of adaptation during invasions? And, does parallel adaptation to climate occur across geographically disparate introductions?

Theoretical models predict that the first steps of rapid adaptation should involve mutations of large effect[26]. Limited studies on the genetics of adaptation during invasions generally support this[16,17], although quantitative genomic approaches tend to bias detection towards

✉e-mail: paul.battlay@monash.edu; Nicholas.kooyers@louisiana.edu

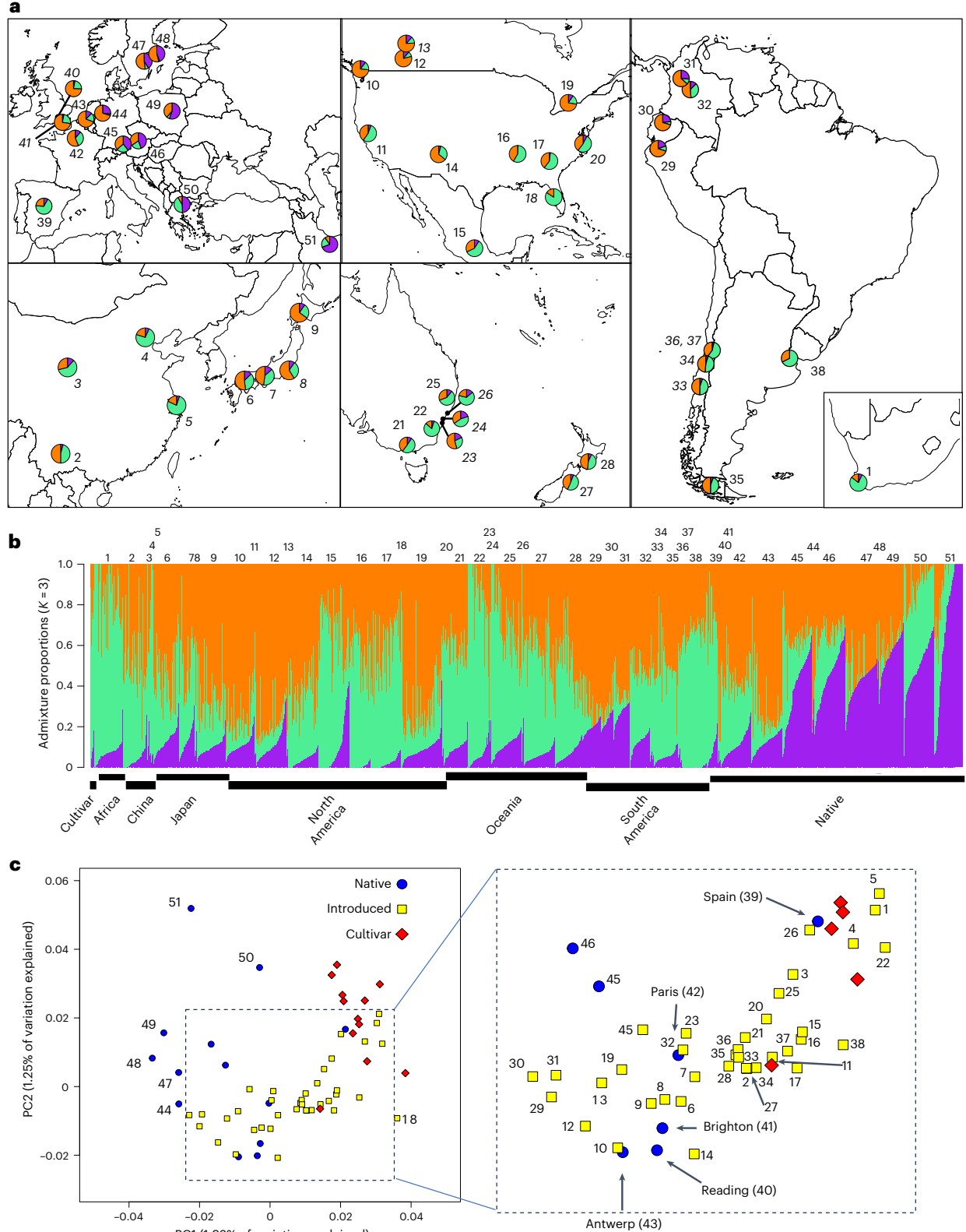

**Fig. 1 | Population structure across worldwide populations of white clover.**
**a**, NGSadmix ancestries mapped across worldwide sampling. Each pie chart within map inserts reflects the average ancestries ($K = 3$) from a given wild population with orange, purple and green corresponding to different ancestries. Cultivars are not included. Numbers represent each population; normal typeset indicates population have >30 samples, italics indicates a population has <10 samples. **b**, Barplots depict ancestry output from the most likely $K$ value ($K = 3$) for 2,660 individuals. Individuals are organized along the $x$ axis by population sorted by continent, longitude and ancestry values. Numbers above barplots indicate populations from the above map. **c**, PCA visualizing population structure across ranges. Colour and shape indicate geographic location of each population as shown in the legend. Insert amplifies area of overlap between native and introduced populations. Numbers refer to populations from **a** and **b**. The 'Spain' point refers to samples from four cities with limited population structure.

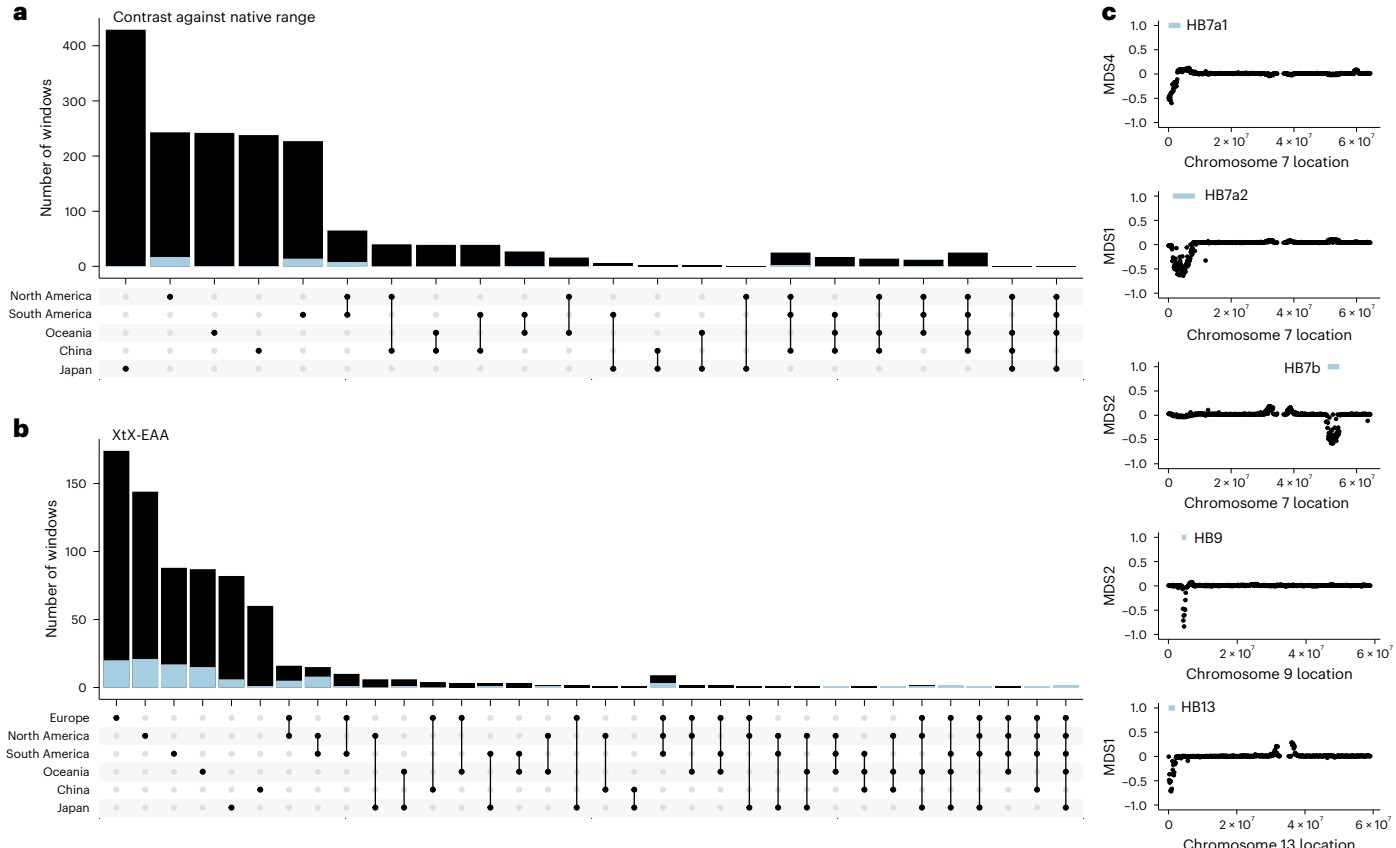

**Fig. 2 | Signatures of structural variants are enriched for patterns of parallel selection across regions where white clover has been introduced. a**, Upset plot depicting outlier windows for native–introduced region contrasts. **b**, Upset plot depicting windows corresponding to climate adaptation in each range (outliers for XtX statistic and correlations with at least one climate variable). Blue portions of bars in **a** and **b** correspond to genomic windows within haploblocks, while black portions of bars represent non-haploblock regions. **c**, Five haploblocks (putative structural variants indicated by blue bars above the region) identified as outliers on MDS axes summarizing local population structure along chromosomes.

large-effect variants[27,28]. Yet many traits critical for range expansion, such as growth rate, size and dispersal, are polygenic in diverse plants. Adaptation via structural variants may reconcile these observations. Structural variants often suppress recombination, allowing clusters of co-adapted small-effect alleles to be inherited as a single segregating unit[29], and have been associated with rapid adaptation[16,30,31]. However, structural variants are difficult to link to fitness across native and introduced ranges, because reciprocal transplants are logistically challenging and are rarely combined with large-scale genomic analyses.

Increasing globalization has resulted in repeated introduction of human-commensal species worldwide. These species often encounter similar selection pressures throughout their ranges (for example, altered climate regimes, release from herbivores or loss of mutualists), thus some degree of parallel phenotypic and molecular adaptation might be expected[32]. However, introduction history and demography can shape genetic variation through factors such as the timing and order of allele arrival (priority effects[33]). Admixture among genetically differentiated native populations in introduced areas can also create unique combinations of alleles[34]. Finally, if selected traits are polygenic, different loci may underlie the same phenotype across regions, weakening signatures of parallel evolution. Effectively parsing introduction history from adaptation requires genomic data from populations spanning comparable climatic gradients in multiple regions.

White clover (*Trifolium repens*) is an outcrossing legume native to Europe and western Asia, introduced globally as a forage and cover crop. Domesticated between 1000 and 1200 AD in present-day Spain, it spread across Western Europe and the British Isles in the mid-1600s[35]. Introductions to North and South America, South Africa, Australia,

Japan and China occurred by the late 1800s via European colonial expansion and probably involved both landraces and wild accessions[36]. Modern cultivars have been developed in North America, China, Australia and New Zealand using germplasm sourced from across the world[37], and these cultivars were widely distributed after 1950[38]. Previous simple sequence repeats (SSR)-based studies show high genetic diversity in both native and introduced populations[39]. Studies of a key defence polymorphism, cyanogenesis, have documented recurrent adaptive clines forming across climatic and urban–rural gradients in native and introduced regions, suggesting rapid post-introduction adaptation[14,32,40–42]. Likewise, additional climate-associated genetic clines have been documented across North America[43].

Here we investigated how introduction history and adaptation interact to shape the global invasion of white clover across diverse climatic regions. Using population-genomic data from six continents, we reconstructed invasion history and identified signals of selection. We sequenced the genomes of 2,660 individuals from 13 native populations, 39 populations across five introduced ranges and 12 widely used cultivars (Fig. 1 and Supplementary Table 1; mean coverage 1.01×) using a low-coverage whole-genome strategy. This approach enables precise and accurate estimation of site frequency spectra and allele frequencies[44,45], supporting analyses of population structure, selection signatures, haploblock detection and genome-wide association studies (GWAS)[46–49]. We independently validate fitness effects of identified variants by conducting four transcontinental field trials using a globally diverse set of accessions. Finally, we performed controlled growth chamber experiments to explore gene expression patterns, providing insight into the biological functions of candidate genes.

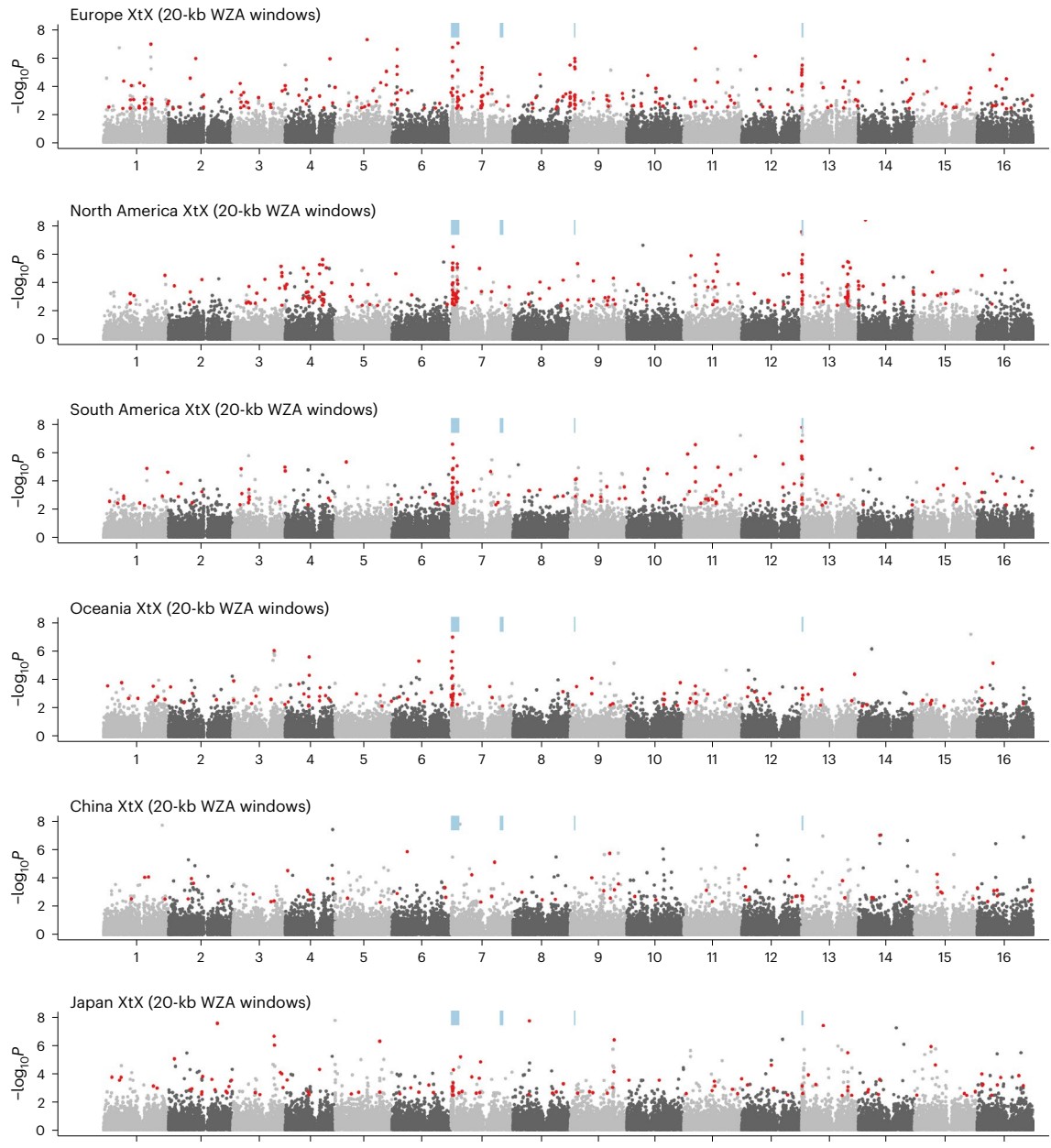

**Fig. 3 | Haploblocks exhibit molecular signals of selection following introduction.** Empirical *P* values for enrichment of XtX (an $F_{ST}$-like statistic that includes a correction for population structure) in 20-kbp windows across the genome using the WZA within each white clover range. Numbers along the *x* axis indicate chromosome number. Red points indicate XtX-EAA outliers, windows that are in top 1% of WZA scores for XtX and correlation (Kendall's Tau) with at least one climate variable. Blue bars indicate haploblock locations.

## Results

### Introduction history

White clover does not exhibit a classic population bottleneck signature in any introduced region. Genetic diversity is high in both the native and introduced ranges, with no clear difference in $\pi$ (native $\pi_{Avg} = 0.016$, introduced $\pi_{Avg} = 0.015$; Welch's analysis of variance (ANOVA): $F_{1,21.47} = 2.58$, $P = 0.12$) or $\theta_w$ (native $\theta_{Avg} = 0.023$, introduced $\theta_{Avg} = 0.019$; Welch's ANOVA: $F_{1,13.89} = 1.23$, $P = 0.29$). Despite this, there is twofold variation in diversity among populations within the same range (Extended Data Fig. 1; $\pi_{Range} = 0.013$–$0.025$). Genome-wide Tajima's *D* values are negative across both the native and introduced ranges, consistent with a recent population expansion (native $D_{Avg} = -0.70$, introduced $D_{Avg} = -0.60$). This pattern aligns with the recent worldwide spread of *T. repens*. However, Tajima's *D* does not differ between native and introduced ranges (Welch's ANOVA: $F_{1,17.3} = 0.30$, $P = 0.59$). Demographic modelling of effective population size ($N_e$) over the past 1,000 years reveals notable variation among populations, with historic increases in $N_e$ in most cases. However, this variation does not correspond to native versus introduced status and there are no signatures of recent bottlenecks or expansions (Extended Data Fig. 1). These results are consistent with the colonization of each introduced area involving repeated introductions of a high number of genetically diverse individuals.

We examined genetic differentiation between populations in native and introduced ranges to better understand the independence of introduction events, different sources of introductions and potential patterns of introgression between introduced ranges. Consistent with high worldwide levels of genetic diversity and limited

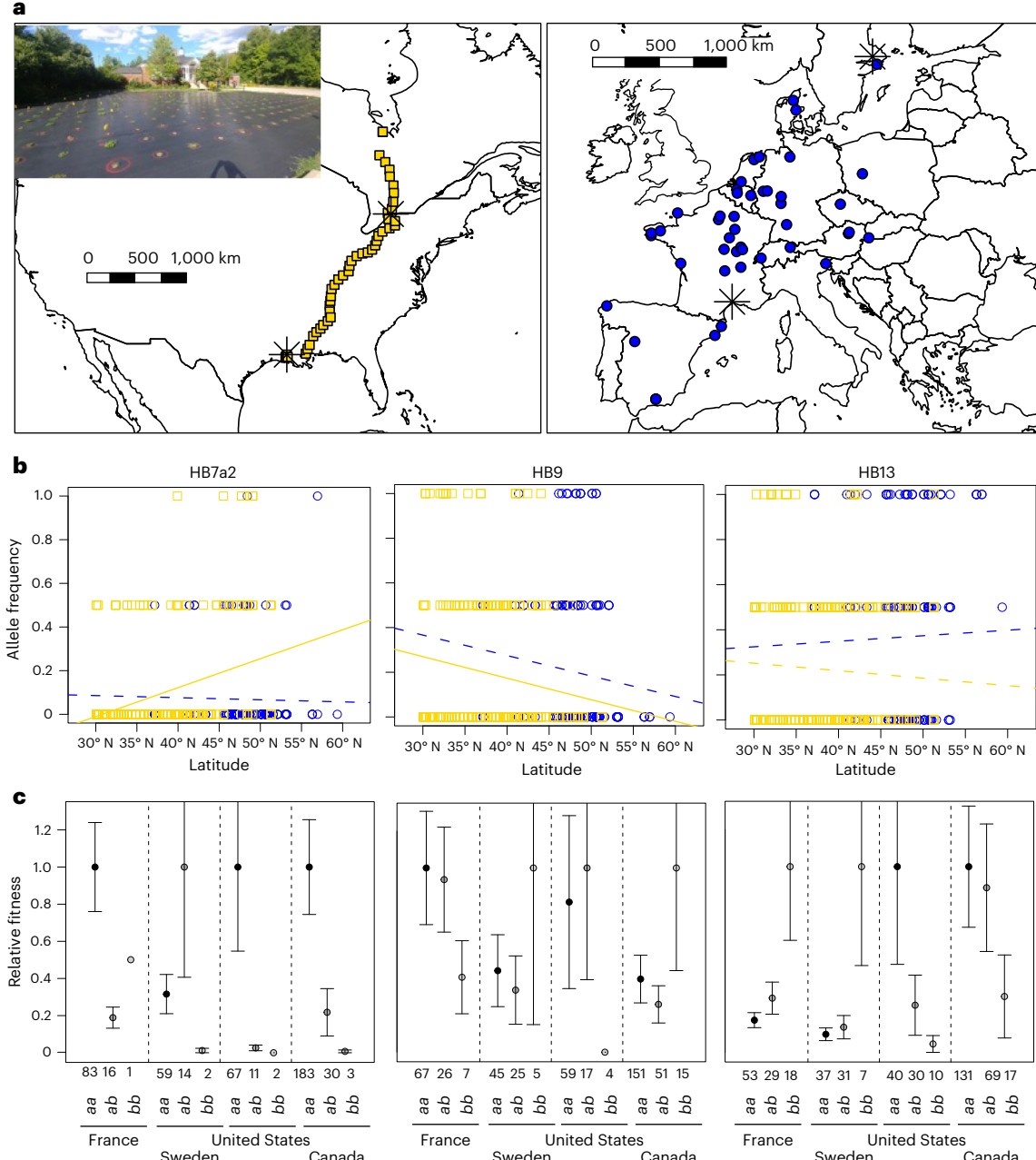

**Fig. 4 | Impacts of haploblock variation on fitness across four field common gardens. a**, Experimental design of the transcontinental field experiment. Points represent the 96 populations that were planted into each garden. Black asterisks are the locations of each garden. Insert picture is of the Mississauga, Ontario, Canada, garden. **b**, Alternative haploblock allele frequency for each individual from European (blue circles) populations or North American (gold squares) pooled across all four common gardens. Regression lines model allele frequency by latitude in Europe (blue) and North America (gold) with solid lines indicating statistically significant (two-sided ANOVA, $P < 0.05$) latitudinal clines and dash lines indicating non-significant regressions. **c**, Average relative fitness for each haploblock genotypes where the $a$ allele represents the reference allele[67] and the $b$ allele represents that alternative allele. Numbers directly above genotypes are the number of samples included in each category. Relative fitness was calculated from total seed mass and standardized by the genotype with the highest fitness within each garden. Error bars represent standard error around the mean.

bottlenecks, differentiation among populations was low (worldwide average weighted pairwise $F_{ST} = 0.027$). Pairwise genetic differentiation was as strong within native and introduced regions as between regions (Extended Data Fig. 2). A strong isolation-by-distance pattern was evident in the native range (Mantel's $r = 0.82$, $P = 0.001$), with weaker patterns within introduced regions (North America—Mantel's $r = 0.18$, $P = 0.10$; South America—Mantel's $r = 0.55$, $P = 0.002$). These results support several introductions from the native region accompanied by subsequent gene flow across each introduced region.

To better parse population structure, we conducted admixture analyses with NGSadmix[50] using putatively neutral sites (four-fold degenerate sites). The most likely number of idealized populations was $K = 3$ (ref. 51). All populations contained all three ancestral gene pools (ancestries) reflecting high within-population variation. These ancestries were strongly represented in different areas of the native range, reflecting latitudinal and longitudinal patterns of isolation-by-distance (Fig. 1). Higher order $K$ values (for example, $K = 4$, 6; Extended Data Fig. 3) further subdivide the native range along a latitudinal gradient.

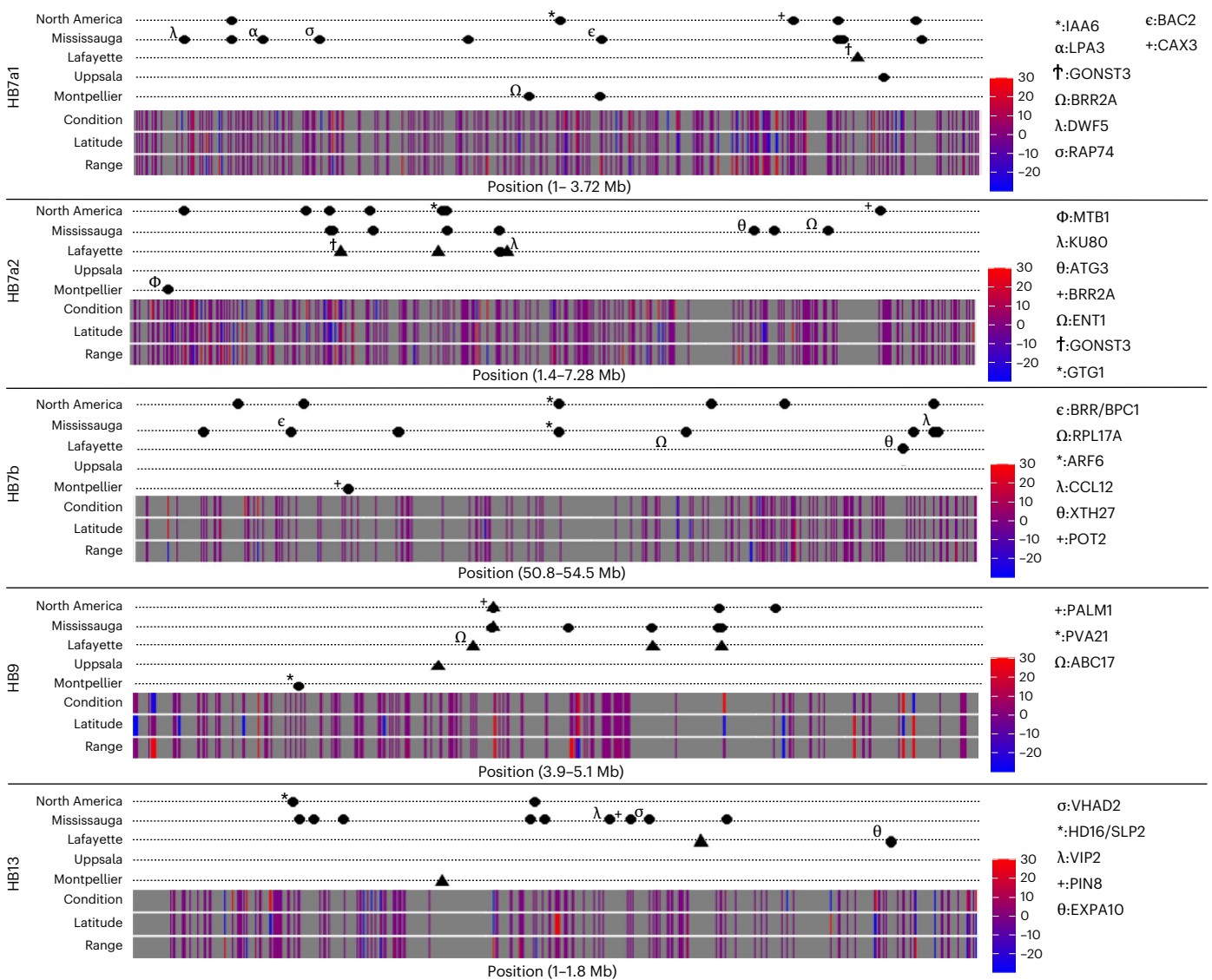

**Fig. 5 | Associations with fitness and differential expression within haploblocks.** GWAS hits for survival to flowering (triangles) and total seed mass (circles) are plotted above expression ribbons. Each dotted line corresponds to a GWAS analysis within a particular garden (Mississauga, Lafayette, Uppsala or Montpellier) or a GWAS analysis that merged the North American gardens (North America). Hits that either fall within a coding sequence or are within 10 kb of a known homologue are marked with a Greek or mathematical symbol as shown in the legend to the right of each graph. Differential expression ribbons visualize expression patterns for each gene within comparisons between condition (drought versus well-watered), latitude (low versus high) and range (native versus North American). Heatmaps for differential expression for each gene range from downregulated (blue) to upregulated (red). HB7a1, HB7a2 and HB7b are all found on chromosome 7, while HB9 is on chromosome 9 and HB13 is on chromosome 13. The position on chromosomes is indicated beneath differential expression heatmaps for each chromosome. Full descriptions of fitness GWAS hits and differentially expressed genes can be found in Supplementary Tables 3 and 5, respectively.

Such structure in the native range suggests that it should be possible to identify the major contributing sources for each introduction.

We compared ancestries of populations within the native and introduced ranges to infer colonization history and admixture. North American populations have ancestries most closely related to Spain in the south and France and Great Britain in the north. High-elevation populations in South America (for example, Medellin, Bogota and Quito) and Japanese populations resemble high-latitude populations in North America (that is, more orange ancestry; Fig. 1). Lower elevation southern populations in South America, as well as Australian populations, New Zealand populations, Chinese populations and South Africa, resemble southern populations in North America with similar ancestry coefficients to Spain (more green; Fig. 1). The similarities between different introduced areas probably reflect a shared introduction history as western European nations brought white clover to these areas, but may also reflect post-introduction admixture between regions, or ecological sorting due to shared climate or biotic selection factors. For instance, Japanese and Chinese populations have very divergent ancestries which probably reflect differences in introduction history. However, parallel differences within continents, such as those observed in North and South America, may reflect contemporary admixture or ecological sorting across climatic gradients.

To better determine the primary sources for each introduced region, we conducted a principal component analysis (PCA). Similarity in PC space closely corresponds to NGSadmix ancestries at $K = 3$. There is differentiation among populations from native and introduced regions (Fig. 1c; PERMANOVA: $F_{1,49} = 4.7$, $P = 0.039$), with a limited number of native populations from western Europe (Spain, Britain, France and Belgium) overlapping in PC space with the introduced populations. Similarity in PC space probably reflects colonization

history and it is notable that there is no clear clustering of different introduction regions. For instance, Canadian populations (Toronto, Calgary, Edmonton and Vancouver) are located next to British, French and Belgian populations, probably reflecting the introduction of white clover to these regions during French and UK colonization. Likewise, other North American populations are located midway between Spanish, French and British populations, reflecting greater Spanish ancestry.

Introduction history alone does not explain the patterns observed in the PCA—introgression with modern agricultural cultivars could shape patterns of genome-wide population structure. To test this, we included 12 modern cultivars developed in North America, Australia and New Zealand using germplasm collected from North America, Australia, France, Spain and New Zealand. Surprisingly, cultivars clustered separately from introduced and native populations aside from the Spanish populations (PERMANOVA: $F_{2,60} = 22.1$, $P = 0.001$, Fig. 1c). With the exception of Grasslands Huia, cultivars are closely related to the Spanish populations and introduced populations from hot climates (Extended Data Fig. 3). Thus, the cultivars do not necessarily reflect the regions where each cultivar originated, but instead tend to have similar genetic compositions to one another. Nearly all these cultivars were derived from field populations bred for resistance to drought and other environmental stressors. Conversely, Grasslands Huia, a New Zealand-derived cultivar, is closely related to other New Zealand wild populations. Thus, although admixture between cultivars and introduced populations clearly occurs, substantial differentiation from natural populations persists.

## Genomic basis of adaptation

Given the proliferation of white clover across diverse habitats, an important question is: what role has adaptation played in the spread of *T. repens*? Selection in introduced regions could favour different alleles that allow adaptation to new conditions in the introduced range and/or that underlie traits that promote rapid invasion. We identified genomic regions with allele frequency differentiation between the native and each of the five introduced regions using genome scans in 20-kilobase pair (kb) windows (BayPass contrast[52]; Extended Data Fig. 4 and Supplementary Table 2). Highly differentiated regions of the genome (top 1% of windows) overlapped between the native–introduced comparisons more than expected by chance (hypergeometric test: $P ≤ 0.00001$; Fig. 2), with the exception of the Europe–Japan contrast (hypergeometric test: $P = 0.16$). These shared patterns of differentiation between introduced regions provide evidence for parallel selection pressures across introduced regions. However, no differentiated genomic windows were shared across all five introductions (Fig. 2) and few were shared across four regions (27 windows; 1.6% of windows that are an outlier for any contrast). Consistent with the admixture analysis, North and South America share the most differentiated windows (128 windows, 29% of outlier windows). These results highlight parallel signatures of selection during range expansion across introduced regions.

Selection can also cause rapid adaptation to the environmental heterogeneity within each introduced range. We examined genomic regions underlying climatic adaptation in each introduced region by performing genome scans to identify 20-kb windows enriched for sites showing both extreme population allele frequency differentiation (BayPass XtX[53]) and correlations with climate[54] (Fig. 3). In each range, between 15% and 52% of XtX outlier windows were also outliers for correlations with at least one of six minimally correlated climate variables (XtX-EAA windows). In all ranges, this overlap was greater than would be expected by chance (hypergeometric test: $P ≤ 7.01 × 10^{-31}$), indicating the importance of rapid adaptation to local climate post-introduction (Extended Data Fig. 5). Across ranges, we observed signatures of genetic parallelism in climate adaptation—the outlier XtX-EAA climate adaptation windows overlapped between ranges more often than expected by chance for all between-range comparisons (hypergeometric test:

$P < 0.013$). There was also some overlap between the windows identified in the contrast analysis and the XtX analyses (native range, 8.6%; introduced ranges, 7%). This pattern may be expected given that the sampled introduced ranges tend to have warmer climates than most of the native range (mean annual temperature: native 10.3 °C, introduced 13.8 °C, $P = 0.006$) and thus regions under climate-associated selection should be differentiated from the native range.

The most notable peaks in each of the genome scans were extended regions of differentiation (haploblocks) on chromosomes 7, 9 and 13. Two partially overlapping haploblocks on chromosome 7 (HB7a1 and HB7a2) and one on chromosome 13 (HB13) were shared among the Europe–North America and Europe–South America contrasts (Extended Data Fig. 4). Allele frequencies within haploblocks HB7a1, and HB13 were strongly associated with climate variables across all ranges, while HB7a2 and HB9 showed strong associations in some ranges but not others. The breadth and synteny of these regions suggest that large structural variants may underlie convergent patterns of differentiation. We used a local PCA of population-genomic data[16,55,56] to identify potential structural variants (inversions and translocations) across the genome. Local PCA has been shown to be a powerful method to identify haploblocks using WGS low-coverage data[49] (Methods). Haploblock regions contained stretches of windows with divergent population structure that clustered into three groups in the PCA (consistent with three genotypes). The middle cluster, which contained putative heterozygous individuals, exhibited higher levels of local nucleotide diversity compared with homozygous individuals in the other two clusters. Corresponding clustering and heterozygosity patterns were also observed in local PCAs using single nucleotide polymorphism (SNP) data from the higher-coverage Toronto population (Extended Data Fig. 6). Haploblock regions exhibited elevated linkage disequilibrium (LD) compared with neighbouring genomic regions (Extended Data Fig. 6), and these LD blocks were reduced when examined within putative homozygous individuals. These genomic signals (PCA cluster, heterozygosity and LD patterns) are consistent with structural variants identified in other species[16,56].

We identified signatures of five putative structural variants among 2,660 white clover samples (Fig. 2 and Extended Data Fig. 6). Haploblocks HB7a1, HB7a2, HB7b, HB9 and HB13 were 3.7, 7.1, 3.7, 1.2 and 1.8 megabase pairs in size, and contained 591, 1,014, 398, 152 and 227 genes, respectively. All haploblock reference and alternative alleles are found in nearly all the populations suggesting that haploblocks existed as standing genetic variation in the native range before introduction. Despite elevated LD across haploblocks, there are still high levels of polymorphism within each block, suggesting that haploblocks are old. However, allele frequencies differed between the introduced and native ranges for HB7a2 ($t_{49} = -3.1$, $P = 0.003$), HB9 ($t_{49} = 2.1$, $P = 0.036$) and HB13 ($t_{49} = -2.2$, $P = 0.03$) indicative of the different colonization history and environmental conditions in each range. Haploblocks have higher levels of within-range differentiation (XtX) than non-haploblock regions across every range, except for China (Extended Data Fig. 7), consistent with relatively strong selection on haploblocks by climatic variation within regions following introduction.

Haploblock regions have stronger signals of selection and parallelism across invasions than non-haploblock regions. Despite covering <2% of the genome, haploblocks contain, respectively, 14.8% and 6% of outlier windows for XtX-EAA and contrast scans. This represents a significant enrichment for XtX-EAA scans in all ranges (hypergeometric test: $P ≤ 0.028$) and for contrast scans between the native range and North and South America (Extended Data Fig. 7; 14% and 12% of contrast outlier windows, respectively; hypergeometric test: $P ≤ 9.67 × 10^{-32}$). Furthermore, 29% and 10% of parallel windows (windows that were outliers in more than one range) for XtX-EAA and contrast scans, respectively, were found within haploblocks, marking a substantial enrichment in these regions relative to the rest of the genome (hypergeometric test: $P ≤ 9.09 × 10^{-16}$). These results suggest that large structural

variants played an important and often parallel role in range expansion following introduction.

### Characterization of adaptive haploblocks

To independently validate evidence of selection on the haploblocks following introduction, we conducted a transcontinental field experiment using diverse populations from the native and introduced ranges coupled with a GWAS. The experiment included common gardens at low and high latitudes in the native range (Uppsala, Sweden; Montpellier, France) and the introduced North American range (Lafayette, LA, United States; Mississauga, Ontario, Canada). Each garden was planted with replicate plants from the same 96 natural populations; 47 populations collected along a latitudinal gradient in North America[57] and 49 collected across Europe[41]. Using the same low-coverage whole-genome sequence approach as above, we genotyped 569 individuals for each of the five haploblocks. Frequencies of the reference and alternative haploblock alleles matched expectations from the worldwide dataset. We observed latitudinal clines in the predicted directions in North America for HB7a2, HB7b and HB9 (Fig. 4). We did not expect latitudinal clines for HB13 or HB7a1 because allelic variation at these haploblocks does not differ between high- and low-latitude populations in the native and eastern North American ranges.

We examined whether allelic variation at each haploblock influenced survival in the first year, growth rate and fecundity (total seed mass). There were significant garden × haploblock genotype effects on fitness consistent with haploblocks conferring local adaptation in the directions expected from the above genome scans (Fig. 4c, Extended Data Fig. 8 and Supplementary Table 3). The strongest association was for HB13, where the alternative haploblock was strongly favoured in the native gardens, but the reference haploblock was strongly favoured in both North American gardens (ANOVA, garden × genotype: $X^2 = 9.6$, $P < 0.0001$). Allelic variation at HB13 is highly predictive of fecundity in the Louisiana garden (Lafayette $r^2 = 0.28$), and predicts more variation than a genomic covariance matrix. Likewise, the HB9 alternative haploblock was marginally favoured in the colder garden in both Europe and North America, while the reference haploblock was favoured in the warmer gardens in both ranges (ANOVA, garden × genotype: $X^2 = 2.6$, $P = 0.05$). Notably, the alternative allele for other haploblocks (HB7a1, HB7a2 and HB7b) are at much lower frequencies in North American and European populations, reducing our power to detect associations with fitness. Nevertheless, patterns at each haploblock still largely fit predictions established from allele frequencies. For instance, plants homozygous for the alternative HB7a2 allele had 92% greater survival in the first year in the Canadian common garden, but none of these homozygotes survived the first year in the Louisiana garden (ANOVA, garden × genotype: $X^2 = 7.5$, $P = 0.059$; Extended Data Fig. 8). Allelic variation at HB7a2 is also moderately predictive of fecundity in both the Canadian and Louisiana gardens (Toronto $r^2 = 0.05$, Lafayette $r^2 = 0.17$). These analyses provide experimental support that selection on haploblocks has driven rapid adaptation within introduced ranges.

We next evaluated which genes within each haploblock could be driving differences in fitness between gardens by conducting separate GWAS within the native and introduced gardens. This method is likely limited for identifying specific genes underlying fitness differences because there is elevated LD within haploblocks; however, there is substantial variation within haploblocks, which allowed us to identify distinct peaks of phenotype–genotype association. Loci in each haploblock were strongly associated with the ability to flower and total seed mass (Fig. 5, Extended Data Fig. 9 and Supplementary Table 4). Most hits were observed in the North American gardens due to sample size differences between gardens, and the analysis probably only detected a subset of fitness-associated genes as a result of limited sample size of some haplotype genotypes. The number of hits exceeded the genome-wide expectation for each haploblock for at least one fitness measure (Extended Data Fig. 10). All hits were located within 10 kb of annotated genes, but only two hits fell directly within the coding sequence of a predicted gene. The abundance of hits near predicted genes, yet the scarcity within coding sequence, is consistent with fitness-related SNPs being in regulatory regions (for example, promoter regions). Moreover, the number and location of fitness-associated SNPs within haploblocks suggests that there are multiple genomic regions under selection within each haploblock, and that differential expression may be an important driver of adaptive phenotypic differences.

Genes near fitness-associated SNPs within the haploblocks correspond to stress resistance, defence and flowering, matching expectations from gene ontology (GO) analyses of haploblock regions (Supplementary Table 5). Of the multiple fitness-associated SNPs within the HB7a1 haploblock, one of the most prominent was found downstream of *IAA6* ($P = 1.32 \times 10^{-5}$, $\beta = -0.97$), a gene encoding a key regulator of auxin responses, phototaxis and development in *Arabidopsis*[58]. The two GWAS hits underlying survival to flowering on HB7a2 were associated with *MT1B* ($P = 2.18 \times 10^{-7}$, $\beta = 1.24$) and *GTG1* ($P = 2.10 \times 10^{-6}$, $\beta = -1.187$)—genes associated with water stress responses, root growth and light responses[59,60]. Two GWAS hits underlying survival to flowering on HB7b were within the coding sequence of *ARF6* ($P = 2.24 \times 10^{-6}$, $\beta = -1.73$); *ARF6* encodes a transcription factor involved in flower maturation in *Arabidopsis*[61]. Notably, several genes associated with photoperiodic control and flowering in other species are associated with survival to flowering within the HB13 haploblock including hits downstream of *Hd16* ($P = 3.61 \times 10^{-5}$, $\beta = -1.03$)[62] and *SLP2* ($P = 3.61 \times 10^{-5}$, $\beta = -1.03$)[63]. Identification of these genes suggests that each haploblock contains ecologically important variation underlying adaptation following invasion and provides specific targets for downstream functional analysis.

We further validated fitness-associated SNPs within haploblocks using a manipulative RNA-seq experiment conducted in growth chambers. We evaluated genome-wide differential expression between high- and low-latitude white clover populations from the native and introduced range in dry-down and well-watered conditions. The water availability treatment was selected because differential mortality between common gardens was hypothesized to be associated with the divergent water regimes. While elevated, differentially expressed genes were not over-represented within haploblocks and had similar magnitude expression changes compared with the rest of the genome for all comparisons (treatment, range or latitude; Extended Data Fig. 10). However, a high percentage of hits in the fitness GWAS above were differentially expressed in at least one comparison (survival to flowering GWAS hits—38.0%, 68 of 179 genes; total seed mass GWAS hits—30.6%, 11 of 36 genes). These genes were relatively uniformly distributed across the different haploblocks, in which survival to flowering GWAS hits represented 25–48% of differentially expressed genes across haploblocks. This group included 12 genes with clear orthologues (survival to flower—*ARC11*, *ATG3*, *CCL12*, *ENT1*, *EXPA10*, *IAA6*, *MT1B*, *PIN8*, *RAP74*, *RPL19*, *SLP2*; total seed mass—*GONST3*) which were associated with fitness in GWAS analyses and had differential expression across drought treatment, range and latitude (ANOVA, treatment × range × latitude: $P_{adj} < 0.0001$; Supplementary Table 6). Several of these genes (including *IAA6*, *MT1B*, *ATG3*, *EXPA10* and *RPL19*) have been associated with drought and oxidative stress in other species[59,64–67]. In sum, the same genes identified in the fitness GWAS have different patterns of expression in populations from different ranges and latitudes. This result is consistent with *cis*-regulatory changes underlying rapid adaptation following introduction, but does not exclude the possibility that haploblocks also include ecologically important variation in protein-coding regions.

## Discussion

We demonstrate that the worldwide invasion of white clover has been achieved through a complex pattern of global colonization and

rapid adaptation. While population structure reflects some aspect of colonization history and independent introduction events, our demographic analyses are consistent with white clover undergoing repeated introductions, followed by admixture among diverse ancestral haplotypes. This complex introduction history has maintained substantial genetic diversity and high effective population sizes in introduced populations. Our results match a growing literature documenting introduction histories that include many source populations and repeated introductions throughout an invasion[23,68–70]. Further parsing the relative contributions of founder events, admixture and expansion will probably require historical sampling and more complex demographic models[16,70,71].

There is strong evidence that climate-related selection has occurred in introduced ranges around the world. Within each range, genomic windows exhibiting extreme variation in allele frequency were enriched for correlations with the environment, demonstrating the key role that adaptation to climate has played during introduction. Moreover, selection scans for local adaptation and divergence from the native range show remarkable parallelism. The strongest and most parallel signatures of adaptation come from just a few haploblocks that also exhibit classic genomic signatures of structural rearrangements (inversions and translocations). Allelic variation within haploblocks is strongly associated with differences in relative fitness between common gardens in the native (Europe) and introduced (North America) range, demonstrating that haploblocks underlie patterns of local adaptation that have evolved in the last 400 years. Variation within these haploblocks suggests that the molecular basis of these differences lies in differential expression of key genes involved in developmental timing, stress tolerance and defence.

The identification of large-effect haploblocks driving rapid parallel adaptation provides key insights into the genomics of rapid adaptation. Our results complement decades of empirical studies documenting clines in inversion polymorphisms in insects[31,72–75], mollusks[76,77], fish[78], mammals[79] and plants[16,30,80], including following recent invasions in *Drosophila*[81] and *Ambrosia*[16,80]. Likewise, theoretical studies have long-predicted that large-effect loci and inversions should underlie rapid adaptation. Our study validates the adaptive importance of these haploblocks, using common gardens to demonstrate the contemporary fitness benefits and trade-offs associated with haploblocks. Three notable observations stem from our system that contribute to our understanding of structural variants and rapid parallel evolution. First, our results suggest that the haploblocks are contributing disproportionately to local adaptation compared with SNPs within other windows. Second, we find substantial diversity within each haploblock allele including variation linked to fitness; this suggests not only that structural variants are old, but also that large structural variants can accumulate different locally beneficial alleles[82,83]. Third, unlike theoretical models of adaptive walks that rely on de novo mutation, each identified haploblock exists as standing genetic variation in the native range, and repeated introductions facilitated their spread to different regions around the globe. Within the context of an invasion, lag periods preceding rapid expansion following introduction may not only be an opportunity for demographic increase and sorting, but also an opportunity for additional input of standing variation from the native or other introduced ranges.

## Conclusions

Our results demonstrate the power and importance of rapid adaptation during an invasion. We find that despite a complex introduction history, strong selection acts to generate both parallel and non-parallel signatures across invasive regions with structural variants playing a key role in local adaptation. We suggest that divergent selection and adaptation are probably the norm for human-commensal species, with large-effect variants present as standing genetic variation in the native range contributing to invasion success globally.

## Methods

### Population genomics dataset

Our dataset includes low-coverage whole-genome sequences from 2,660 samples collected from 50 different cities and surrounding rural areas spanning the native range in Europe and Western Asia (12 cities) as well as introductions to North America (11 cities), South America (10 cities), Japan (4 cities), China (4 cities), Oceania (8 cities) and Africa (1 city). These samples were originally collected as part of the Global Urban Evolution Project from 2016 to 2019[42]. Each city was treated as a single population and sample sizes for each population ranged from 5 to 120. This heterogeneity in sample size was intentional as we wanted to include a number of cities with high sampling for better estimates of site frequency spectra and population-genomic statistics (31 cities; average 80.74, standard deviation 17.7 individuals). We then added further cities with lower sampling that we deemed as important areas for understanding colonization history (19 cities; average 5.95, standard deviation 0.23 individuals). Additionally, we sequenced 32 samples collected from four cities in Spain (A Coruña, Granada, Salamanca and San Sebastian)[41], as well as 12 popular cultivars bred in the United States (Durana, Patriot, Renovation, Merit, Pilgrim, Osecola, LA-S1 and CA Ladino), Australia (Irrigation) and New Zealand (Crau, Grassland Huia and Grasslands Pitua). Cultivars are still introduced today across crop fields, as forage crops, in public parks and as bait by deer hunters. Details on library construction and sequencing for new samples are described in the Supplementary Information. Environmental data for each sampling location were extracted from BIOCLIM using the raster v.3.6-26 package in R. Importantly, although some samples within the population-genomic overlap with those in another recent paper[42], the research questions, bioinformatic and statistical analysis, results and conclusions are all distinct and new.

### Analysis of demography and worldwide population structure

Sequences were processed using a common pipeline (https://github.com/James-S-Santangelo/glue_dnaSeqQC) and aligned to a chromosome-level genome assembly[84]. For demographic analysis, we extracted four-fold degenerate sites using the Degeneracy Pipeline (https://github.com/tvkent/Degeneracy) and used all sites for genome scans. We assessed population genomic diversity, differentiation and structure using genotype likelihoods in ANGSD v.0.929 (ref. 85). To examine genetic diversity within each population, we first calculated genotype likelihoods and site allele frequency likelihoods (SAF) for each population independently using only four-fold degenerate sites (-GL 1 -doMaf 2 -doCounts 1 -dumpCounts 2 -baq 2 -minQ 20 -minMapQ 30 -doSaf 1 -sites 4fold.sites). One-dimensional site frequency spectra were used to calculate thetas ($\theta_w$ and $\theta_\pi$) using realSFS saf2theta and thetaStat do_stat. We recalculated genotype likelihoods and SAFs for each population using the reference genome to assign major and minor alleles (-GL 1 -doMaf 2 -minMaf 0.05 -doCounts 1 -dumpCounts 2 -baq 2 -minQ 20 -minMapQ 30 -doSaf 5 -doMajorMinor 4 -sites 4fold.sites) for estimating differentiation using Hudson's Fst (realSFS fst index -whichFst 1). Average number of SNPs per population for these analyses was 10,784,068 (s.d. 865,692).

We identified signatures of bottlenecks by comparing genetic diversity statistics and Tajima's *D* between native and each introduced region. Models including covariates for population sample size and number of sites do not qualitatively change conclusions. We estimated $N_e$ through time using a coalescent framework implemented in EPOS[86], focusing on population contractions in the last 1,000 years as signatures of bottlenecks. We investigated patterns of genetic differentiation within and among populations across the native and introduced ranges by calculating pairwise weighted and unweighted $F_{ST}$ values using ANGSD[87,88]. Isolation by distance and isolation by environment in native and introduced ranges were assessed via Mantel tests using the mantel() function within the vegan library[89] with Haversine geographic

distance matrices via distm() function within the geodist library and climatic distance matrices using the dist() function in the vegan library.

We examined worldwide population structure and individual ancestry using NGSadmix[50]. Genotype likelihoods were re-estimated treating all samples as a single population and adding a minor allele frequency cutoff of 0.05 (-minMaf). This resulted in 533,655 sites. NGSadmix runs included three to eight replicates of $K = 1–8$ using 10,000 iterations per replicate (-maxIter). To determine the most likely number of clusters, we examined standard deviations in likelihoods at each $K$ and used the method described in ref. [51] to identify the most likely number of ancestral clusters and the uppermost level of population structure. To better dissect introduction history, we examined patterns of nested population structure using PCA. We used PCAngsd[90] to generate a variance–covariance matrix using genotype likelihoods and estimated allele frequencies (pcangsd.py), and then extracted the eigenvectors (the principal components) of the covariance matrix using eigen() function in R. To examine potential clustering within the PCA, we conducted PERMANOVA using the adonis2() function within the vegan library[89]. Differences in number of samples, sequencing coverage, batch effects from sequencing runs have limited impact on our inferences of population structure (Supplementary Information). Additionally, distance-based pruning of our dataset and removing haploblocks do not alter population structure (Supplementary Figs. 2 and 3).

## Genome scans for signatures of selection

We identified regions of the genome under selection using two separate approaches. First, we contrasted allele frequencies in the native range with those in each invasive range. Second, we looked for relationships between allele frequency and climate within each individual range as evidence of local climate adaptation. Genotype likelihoods were calculated in ANGSD (-GL 1 -doGlf 2 -doMajorMinor 4 -doMaf 2 -baq 2 -minQ 20 -minMapQ 30 -SNP_pval 1e-6 -minMaf 0.05) in each range (Europe, North America, South America, Oceania, China and Japan) for climate adaptation scans or pair of ranges for contrast scans. We then estimated population allele frequencies for these sites in each population individually using ANGSD (-GL 1 -doGlf 2 -doMajorMinor 4 -doMaf 2 -doCounts 1 -baq 2 -minQ 20 -minMapQ 30 -minMaf 0). Allele frequencies for sites were only used if they were callable for all populations in a particular scan (14.7–22.7 M sites per range).

We used the BayPass contrast statistic[52] to summarize allele frequency differentiation at each site between European populations and populations from an invasive range while correcting for population structure. Enrichment of contrast outliers was calculated for non-overlapping 20-kb windows using the weighted-$Z$ analysis (WZA[91]) and outlier windows were defined as the 1% tail of the distribution of WZA window scores.

We tested for genomic regions with greater differentiation than expected by chance within each native range while accounting for genome-wide population structure using the BayPass core model. For these genome scans, we generated population covariance omega matrices for each range in BayPass v.2.2 (refs. [52,53]) using 10,000 sites sampled from outside annotated genes. We then ran the BayPass core model to quantify allele frequency divergence between populations within each range while accounting for population structure using the omega matrix (XtX). Next, correlations between population allele frequencies in each range and six minimally correlated bioclimatic variables (BIO1, BIO2, BIO8, BIO12, BIO15 and BIO19 from the WorldClim dataset[54]) were quantified using the absolute value of Kendall's Tau. In each range, we used WZA to identify non-overlapping 20-kb windows that were enriched for outliers for the XtX statistic and correlations with each bioclimatic variable. Outlier windows for each statistic were defined as the 1% tail of the distribution of WZA window scores. Outlier windows that overlapped between genome scans were identified, and their enrichment relative to a hypergeometric distribution was tested in R.

## Haploblock identification

We identified haploblocks—population-genomic signatures of large structural variants—using local PCA, which has proved reliable in a range of genomic datasets (for example, refs. [16,55,56]) including those with low-coverage whole-genome sequencing data[49]. We modified the method described by ref. [55] to use covariance matrices from PCAngsd v1.10 (ref. [90]), which were calculated in 100-kb windows from beagle files generated in ANGSD v.0.929 (5) (-GL 2 -doMajorMinor 1 -doCounts 1 -doGLF 2 -SNP_pval 1e-6 -doMaf 2 -doGeno -1 -doPost 1 -minMapQ 30 -minQ 20 -trim 5 -minMaf 0.05 -minInd 665 -geno_minDepth 2 -setMinDepthInd 2 -uniqueOnly 1). Local population structure along each chromosome was analysed on five multidimensional scaling (MDS) axes and outliers were identified from the 5% corners of each pair of MDS axes. We selected MDS scan regions for further analysis on the basis of the presence of clusters of a particular outlier in a chromosomal region, that is, stretches of a chromosome where the population structure was both similar and extreme. In total, ten such regions were analysed, but five were excluded on the basis of lack of clustering in the local PCA and/or patterns of heterozygosity incongruous with a structural variant. Heterozygosity was also calculated for each sample in each candidate region using ANGSD (-dosaf 1 -minMapQ 30 -minQ 20 -trim 5 -GL 2) and realSFS v.0.929 (ref. [92]) (-fold 1). After filtering out samples with less than 0.4× coverage, putative structural variants were identified by the presence of three clusters of samples along a single principal component axis, indicative of two homozygous and one heterozygous inversion genotype. Samples were assigned to clusters manually. We validated haploblock genotyping first by looking for the presence of significantly elevated heterozygosity in the middle (heterozygote) cluster (Wilcoxon test $P < 0.0003$ for all heterozygote versus homozygote comparisons). Second, we performed LD scans with ngsLD v.1.2.0 (ref. [93]) (--min_maf 0.05 --max_kb_dist 0) on 5,000 randomly sampled sites from each chromosome containing a haploblock. For each haploblock, LD scans were run on a set of samples homozygous for the more common haploblock allele, as well as a random set of samples of the same size. We further tested our haploblock genotyping using 109 samples from Toronto that were sequenced to a coverage of -10×. We called SNPs from alignments of these samples using FreeBayes v.1.3.6 (ref. [94]) and filtered them using VCFtools v.0.1.15 (ref. [95]) (--minQ 30 --minGQ 20 --minDP 5 --max-alleles 2 --max-missing 0.7). To identify GO terms enriched in haploblocks, the topGO library[96] was used with Fisher's exact test, the 'weight01' algorithm and $P < 0.05$ to assess significance.

## Associations between haploblocks and fitness

We examined patterns of local adaptation and the genomics of adaptation using four common gardens located in the southern and northern region of the native range (Montpellier, France and Uppsala, Sweden, respectively) and the southern and northern regions within the North American introduced range (Lafayette, United States, and Mississauga, Canada, respectively). This study was originally reported by ref. [57], but the sequencing and GWAS for this work is new to this study. Common gardens were conducted for 2 years at each site, 2020–21 in North American gardens and 2021–22 in European gardens. Seedlings from the same lines were planted in each garden. Seeds were collected from 49 white clover populations spanning a 27° latitudinal gradient in Europe and from 47 additional populations spanning a 21° latitudinal gradient in North America. Seeds were grown for a single refresher generation and outcrossed via hand-pollination within each population. We established four to six outbred lines per population before randomizing and planting directly into the natural soil of a cultivated lawn at each site. Survival was surveyed and mature fruit were collected weekly. We report two measures of fitness: 'survival to flowering' is a binary variable that indicates whether a plant was able to flower during the 2-yr experiment and represents both viability and ability to mate; 'total seed mass' reflects both viability and fecundity as plants that did not produce any seeds had no seed mass.

 

We generated low-coverage whole-genome sequences for 569 samples across the four gardens using the same library construction and bioinformatics pipelines as above. We estimated haploblock genotypes by performing local PCAs as above on each previously identified haploblock region including all lcWGS samples. The first two principal components of genetic variation across population-genomic and common garden samples for each haploblock region were visualized and used to assign common garden samples to genotype clusters.

We assessed whether haploblocks were associated with adaptation following introduction via a three-pronged approach. We first validated our population genomics dataset by using linear models (lm()) to identify associations between haploblock genotype and latitude of collection site. We then qualitatively compared whether clines in the native and introduced region in these gardens matched the population-genomic dataset. Second, we examined how haploblock variation impacts fitness across gardens. We modelled survival to flower and total seed mass in separate univariate generalized linear models with garden, genotype and garden × genotype interaction as factors. GLMs were implemented using glm() and statistical significance of each factor was assessed using Anova() with Type III sum of squares in the car library[97]. Survival to flower was modelled with a binomial error distribution and a logit link function. Total seed mass was log(+1) transformed and modelled with a Gaussian distribution and identity link function. Finally, we calculated relative fitness from total seed mass data for each haploblock to better understand the strength of selection acting on each haploblock within each garden. Relative fitness for each haploblock was calculated by dividing each individual value for total seed mass by the average value of total seed mass for the genotype with the highest fitness in the garden.

To identify the genes and phenotypes potentially under selection, we conducted GWAS with a genotype-likelihood framework implemented in ANGSD. We conducted independent GWAS for the two fitness traits in each garden. Then, we pooled European gardens and North American gardens and conducted GWAS for each trait in each pooled sample. Genotype likelihoods were estimated for each range (174 individuals from the native European range and 395 individuals from the introduced North American range) in ANGSD (-GL 1 -minMaf 0.05 -minMapQ 30 -minQ 20). GWAS used a hybrid model (-doAsso 5) which first uses a score statistic to evaluate the joint maximum likelihood estimate between a trait and an observed marker[98]. If the chi-square test falls below a particular threshold (-hybridThres 0.05), a latent genotype model with an expectation-maximization algorithm is fit[45]. We controlled for population structure by adding the first 20 principal components as covariates. Principal components were generated in PCAngsd as above. In GWAS for each combined range, we also added garden as a covariate. To account for multiple tests, we used a conservative Bonferroni correction. We used permutation analyses to determine whether the number of fitness GWAS hits exceeded expectations from the rest of the genome (Supplementary Information).

**Differential expression analysis.** We performed a manipulative experiment to examine variation in expression between white clover populations in the native and introduced range under dry down and well-watered conditions. This study was first reported by ref. 99 and we narrow our focus here to differential expression analyses in identified haploblock regions. We grew seeds collected from three or four populations from low latitude and high latitude in the European and North American ranges (14 populations total). Seeds for each population had been pooled from >25 different maternal lines, and we grew one to three seeds from each population in the control and well-watered treatments (47 total samples). Thus, biological replication occurred at the population level. Plants were grown for 6 weeks to accumulate aboveground and belowground biomass. At 6 weeks, all pots were saturated with water by bottom-watering. Plants in the control (well-watered) flats received periodic watering according to

our standard greenhouse protocol. Plants in the dry down treatment did not receive additional water. Each day, we assessed soil moisture in each pot using a SMT150T soil moisture meter (Dynamax). Leaf tissue from two healthy adult leaves was flash frozen in liquid nitrogen 10 days after the dry down treatment began from plants in both the well-watered and control treatment. Library construction, sequencing and bioinformatics details in the Supplementary Information.

We used DESeq2 (ref. 100) to test for differences in transcript abundance between dry down and well-watered treatment groups, between the North American and European ranges, and between high- and low-latitude populations. We controlled for volumetric water content at time of tissue collection by treating it as a covariate in each of the DEseq2 models. We used two different models to examine differential patterns of gene expression across treatments, range and latitude. The first included all interactions (treatment × range × latitude). The second set of models were univariate models examining differential expression across treatment, range and latitude separately. Genes were categorized as differentially expressed if false discovery rate (FDR) was <0.1. We evaluated whether transcribed genes located within haploblocks were more or less differentially expressed than in other regions of the genome by resampling across the genome. Briefly, the same number of genes found within each haploblock were randomly sampled across the genome 10,000 times while preserving synteny. The number of genes with an FDR < 0.1 for each of the 10,000 sampled haploblocks was summed and the average log2Foldchange was calculated, which were then used to create a null distribution for each haploblock region of the expected number of differentially expressed genes and their relative log2Foldchange.

### Ethics and inclusion

This study involves worldwide collection and sequencing of plant germplasm. All collectors were given opportunity to collaborate and obtain authorship. All collections were properly permitted with local authorities.

### Reporting summary

Further information on research design is available in the Nature Portfolio Reporting Summary linked to this article.

## Data availability

Low-coverage whole-genome sequences for accessions used in population genomics analyses can be found as fastq files in the NCBI SRA database (Bioprojects: PRJNA1081485, PRJNA1179961). Metadata and fitness data from the four-way common garden study can be found on Dryad[101] and associated low-coverage whole-genome sequences are in the NCBI SRA database (Biooproject: PRJNA1098360). Raw fastq files from RNA-seq expression experiment can be found in the NCBI SRA database (Bioproject: PRJNA1131002). Source data are provided with this paper.

## Code availability

All code from this paper is available via GitHub at github.com/pbattlay/glue-invasions (ref. 102).

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

## Acknowledgements

This work would not have been possible without white clover collections by 287 fellow scientists in the Global Urban Evolution network. Support for field work was provided by A. Daugereaux, the UL-Ecology Center and numerous field and laboratory assistants in each common garden. The Louisiana Optical Network Infrastructure provided computational support. The use of genetic data originating from Ecuador was approved by the Ecuadorian Ministerio del Ambiente (access permit MAE-DNB-2018-0106 and transfer permit ATM-CM-2018-0106-001–2019). Funding was provided by an NSERC CGS Doctoral Award to L.J.A.; NSF grants OIA-1920858 and DBI-2244712 to N.J.K.; CNRS-University of Toronto PhD Student Travel Grant to M.T.J.J. and C.V.; NSERC Discovery Grant (RGPIN-2016-06063), Canada Research Chair (950-231981), NSERC Steacie Fellowship (544292) and NSERC RTI Grant (EQPEQ 423691) to M.T.J.J.; and ARC (DP220102362) and HFSP (RGP0001) grants to K.A.H.

## Author contributions

P.B., B.T.H., J.I.M.-R., J.S.S., L.J.A., P.Y.K., R.W.N., M.T.J.J., K.A.H. and N.J.K. conceptualized and designed the experiments. J.S.S., L.J.A., S.G.I., F.A., D.N.A., J.A., A.B., M.S.C., S.D., M.F.-A., W.G., C.G.-L., P.E.G., G.R.H., C. Lampei, C. Lara, A.L.-L., D.S.L., T.J.S.M., N.M., M.M.B., A.M., M.M., J.P., V.P., J.A.M.R., D.J.R., R.S.R., J.K.R., A.C.S., K.S.W., I.T., A.V.W., M.T.J.J. and N.J.K. provided resources for the experiments. L.J.A., N.K., A.P., A.T., C.V., F.V., C.S., C.M.P., R.A., P.Y.K., M.T.J.J. and N.J.K. performed the experiments. P.B., B.T.H., J.I.M.-R. and N.J.K. curated datasets. P.B., B.T.H., J.I.M.-R., J.S.S., J.W., A.E.C., M.F. and N.J.K. conducted analysis of the data. L.J.A., C.V., M.T.J.J., K.A.H. and N.J.K. acquired funding for the experiments. M.T.J.J., K.A.H. and N.J.K. were the project administrators and provided supervision. The original draft was written by P.B., B.T.H., M.T.J.J., K.A.H. and N.J.K. Review and editing was provided by all authors and all authors approved the final version of this paper.

## Funding

## Competing interests

The authors declare no competing interests.

## Additional information

**Extended data** is available for this paper at https://doi.org/10.1038/s41559-025-02751-2.

**Correspondence and requests for materials** should be addressed to Paul Battlay or Nicholas J. Kooyers.

Paul Battlay[1,38] ✉, Brandon T. Hendrickson[2,38], Jonas I. Mendez-Reneau[2], James S. Santangelo ®[3], Lucas J. Albano[4], Jonathan Wilson ®[1], Aude E. Caizergues[4], Nevada King[2], Adriana Puentes[5], Amelia Tudoran ®[5], Cyrille Violle ®[6], Francois Vasseur[6], Courtney M. Patterson[2], Michael Foster[7], Caitlyn Stamps[2], Simon G. Innes[2], Rémi Allio[8], Fabio Angeoletto[9], Daniel N. Anstett ®[10,11], Julia Anstett ®[12], Anna Bucharova[13], Mattheau S. Comerford[14], Santiago David ®[12], Mohsen Falahati-Anbaran[15], William Godsoe[16], César González-Lagos ®[17], Pedro E. Gundel[18,19], Glen Ray Hood[20], Christian Lampei ®[13], Carlos Lara[21], Adrián Lázaro-Lobo[22], Deleon Silva Leandro[23], Thomas J. S. Merritt ®[24], Nora Mitchell ®[25], Mitra Mohammadi Bazargani[15], Angela Moles ®[26], Maureen Murúa[27],

**Juraj Paule**[28]**, Vera Pfeiffer**[29]**, Joost A. M. Raeymaekers** ⓘ [30]**, Diana J. Rennison** ⓘ [31]**, Rodrigo S. Rios** ⓘ [32]**, Jennifer K. Rowntree**[33]**, Adam C. Schneider**[34]**, Kaitlin Stack Whitney** ⓘ [35]**, Ítalo Tamburrino**[36]**, Acer VanWallendael**[37]**, Paul Y. Kim** ⓘ [7]**, Rob W. Ness**[4]**, Marc T. J. Johnson** ⓘ [4]**, Kathryn A. Hodgins** ⓘ [1,39] **& Nicholas J. Kooyers** ⓘ [2,39] ✉

[1]Monash University, Melbourne, Victoria, Australia. [2]University of Louisiana, Lafayette, LA, USA. [3]University of California, Berkeley, CA, USA. [4]University of Toronto Mississauga, Mississauga, Ontario, Canada. [5]Swedish University of Agricultural Sciences, Uppsala, Sweden. [6]Université Montpellier, CNRS, EPHE, IRD, Montpellier, France. [7]Grambling State University, Grambling, LA, USA. [8]UMR CBGP, INRAE, CIRAD, IRD, Institut Agro, Université Montpellier, Montpellier, France. [9]Universidade Federal de Rondonópolis, Rondonópolis, Brazil. [10]Michigan State University, Lansing, MI, USA. [11]Cornell University, Ithaca, NY, USA. [12]University of British Columbia, Vancouver, British Colombia, Canada. [13]Philipps University Marburg, Marburg, Germany. [14]University of Massachusetts, Boston, MA, USA. [15]NTNU University Museum, Trondheim, Norway. [16]Lincoln University, Lincoln, New Zealand. [17]Universidad Adolfo Ibáñez, Santiago, Chile. [18]Centro de Ecología Integrativa, Instituto de Ciencias Biológicas, Universidad de Talca, Talca, Chile. [19]Universidad de Buenos Aires, Buenos Aires, Argentina. [20]Wayne State University, Detroit, MI, USA. [21]Universidad Católica de la Santísima Concepción, Concepción, Chile. [22]Biodiversity Research Institute IMIB, Mieres, Spain. [23]Federal University of Mato Grosso, Cuiabá, Brazil. [24]Laurentian University, Sudbury, Ontario, Canada. [25]University of Wisconsin - Eau Claire, Eau Claire, WI, USA. [26]UNSW Sydney, Kensington, New South Wales, Australia. [27]Universidad Mayor, Santiago, Chile. [28]Freie Universität Berlin, Berlin, Germany. [29]Washington State University, Pullman, WA, USA. [30]Nord University, Bodø, Norway. [31]University of California, La Jolla, CA, USA. [32]Universidad de La Serena, La Serena, Chile. [33]University of Plymouth, Plymouth, UK. [34]University of Wisconsin, La Crosse, WI, USA. [35]Rochester Institute of Technology, Rochester, NY, USA. [36]Universidad de Chile, Santiago, Chile. [37]North Carolina State University, Raleigh, NC, USA. [38]These authors contributed equally: Paul Battlay, Brandon T. Hendrickson. [39]These authors jointly supervised this work: Kathryn A. Hodgins, Nicholas J. Kooyers. ✉e-mail: paul.battlay@monash.edu; Nicholas.kooyers@louisiana.edu

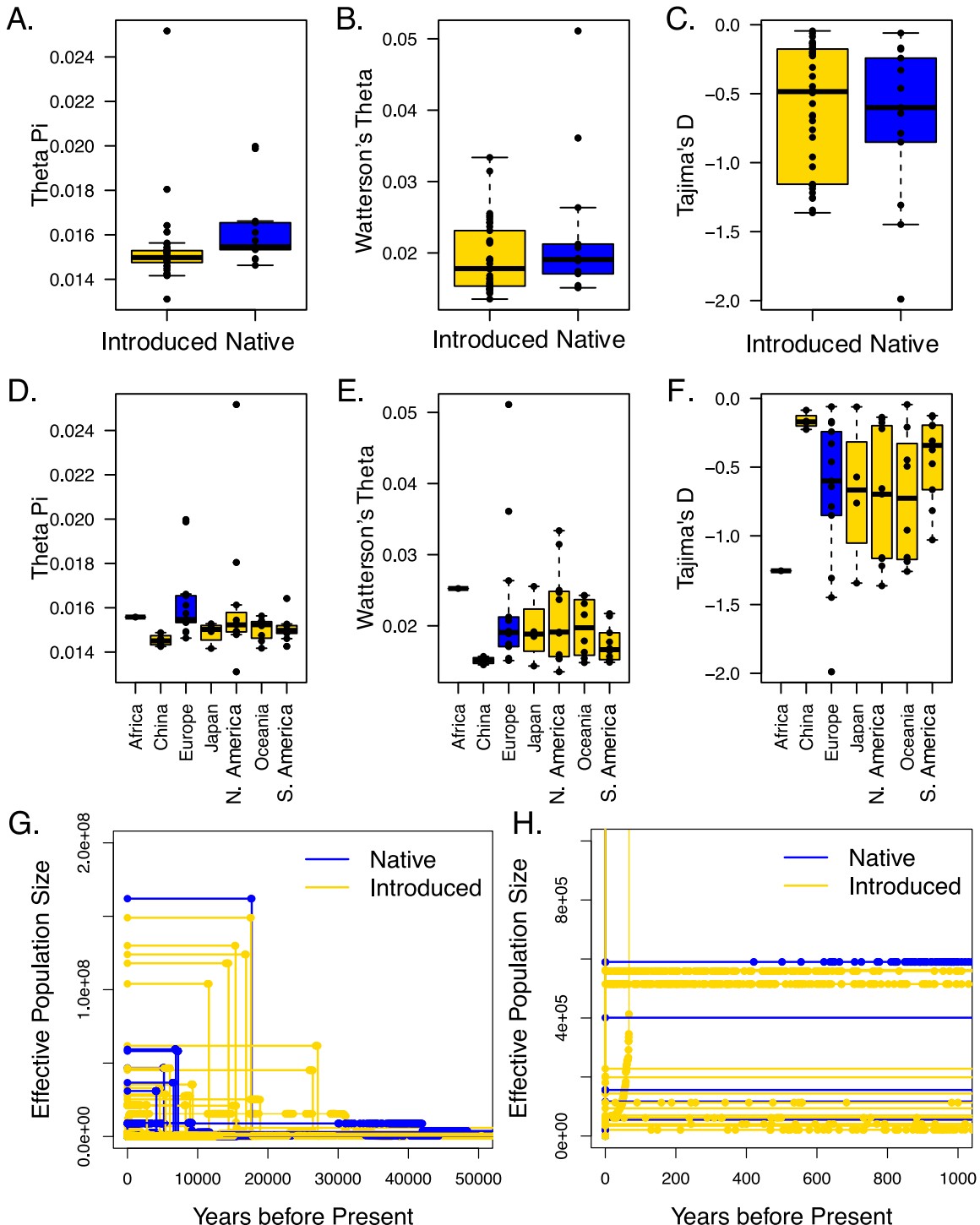

**Extended Data Fig. 1 | Population genetic summary statistics across native and introduced ranges. (a–c)** Boxplots compare measures of genetic diversity (π and θ$_w$) as well as Tajima's D between native and introduced ranges. Sample size include 13 native populations and 38 introduced populations. **(d–f)** Boxplots further parse introduced population into individual introduction events. Each point represents the genome-wide average for a single population. Box edges in boxplots represent the interquartile range, center line represents the median, and upper and lower whiskers are the largest value either greater or less than,

respectively, 1.5 times the interquartile range. There are 1, 4, 13, 4, 11, 8 and 10 populations included in the Africa, China, Europe, Japan, N. America, Oceania, and South America boxplots, respectively. **(g, h)** Coalescent reconstructions of effective population size through time as estimated through EPOS[86]. Neither native nor introduced populations exhibit any signatures of bottlenecks following introduction. Instead, most populations show signs of population expansion.

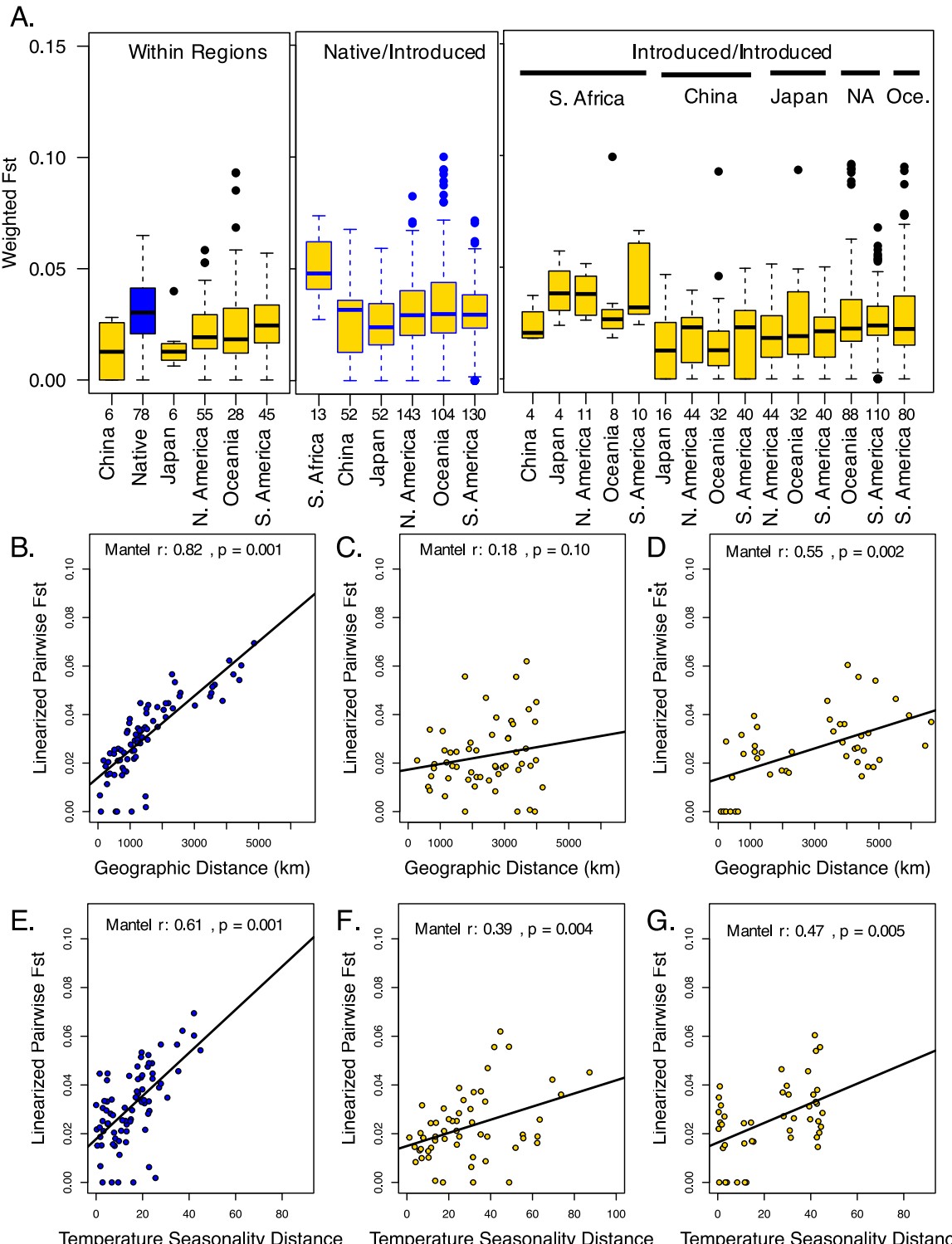

**Extended Data Fig. 2 | Population structure within and among native and introduced ranges. a**. Weighted pairwise fst within each range, weighted pairwise fst values between native and introduced populations, and weighted pairwise fst values between populations in different introduced ranges. Pairwise fst values are generally low and similar across all worldwide populations. Number of pairwise comparisons included in each boxplot is presented on x-axis below each box. Box edges in boxplots represent the interquartile range, center line represents the median, and upper and lower whiskers are the largest value either greater or less than, respectively, 1.5 times the interquartile range. **b–g**. Mantel tests for isolation by distance (**b-d**) and isolation by environment (**e-g**) across the native range (**b,e**), North America (**c,f**) and South America (**d,g**). Mantel tests are two-sided.

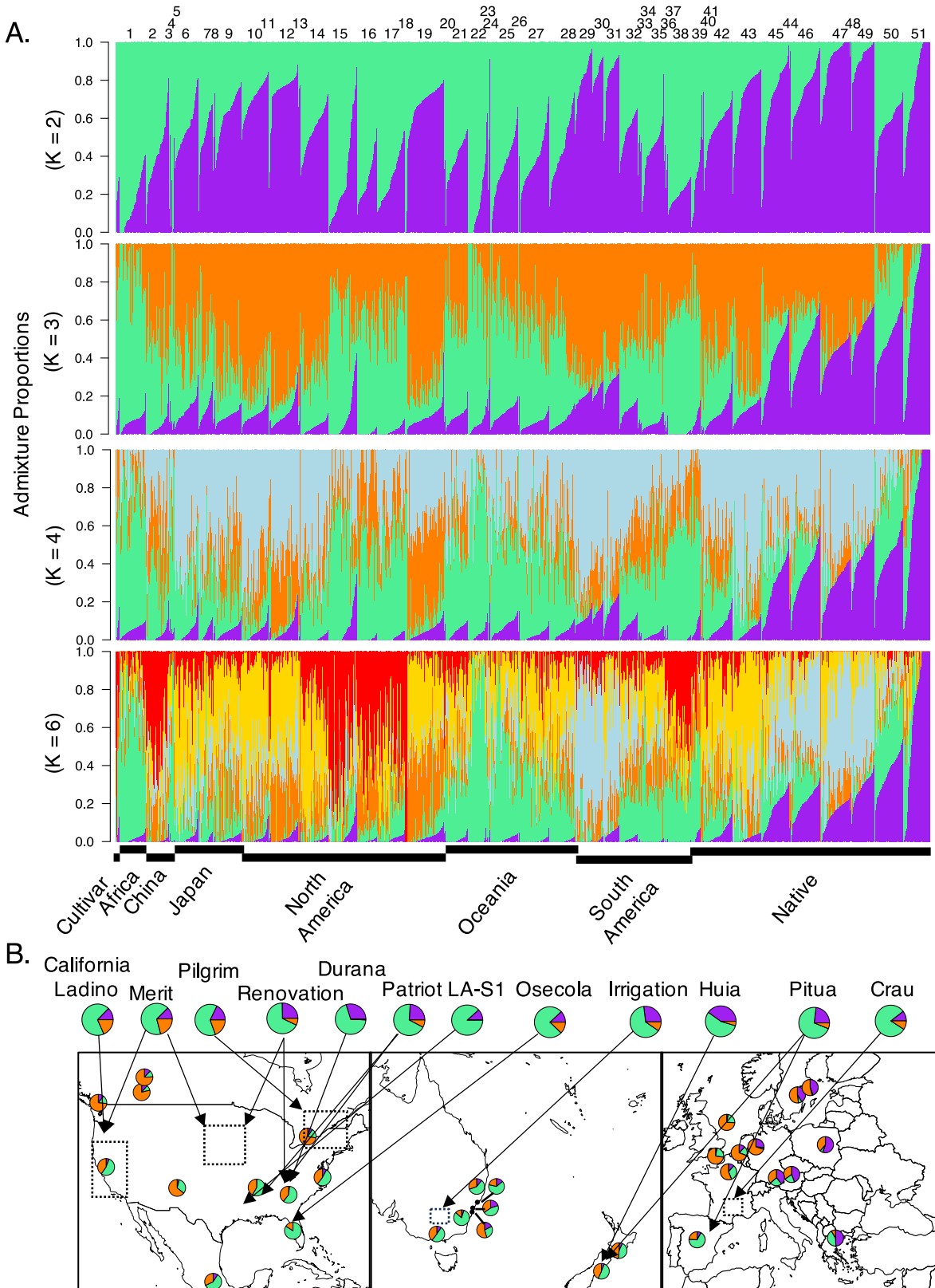

**Extended Data Fig. 3 | NGSadmix assumed population ancestry mapped across worldwide sampling and cultivars. a.** Barplots depicts ancestry output from K=2,3,4, and 6 K-values. Best K was K=3. Individuals are organized along the x-axis by population sorted by continent, longitude, and ancestry values. **b**. A map of cultivar origins and NGSadmix assumed ancestries. Each pie chart within map inserts reflects the average ancestries (K=3) from a given wild population (not from cultivars). Pie charts outside the map reflect the ancestry for each cultivar. Arrows indicate the relative locations where wide populations were collected to generate each cultivar. Note that some cultivars were generated from crosses between wild populations from multiple areas. Dotted boxes reflect when the location of the originating wild population is a general location. Note that Durana and Renovation stem from plants collected in Georgia, USA, LA-S1 and Patriot originate from plants in Louisiana, USA and Mississippi, USA respectively, and Pitua is the result of crossing between Spanish and New Zealand accessions.

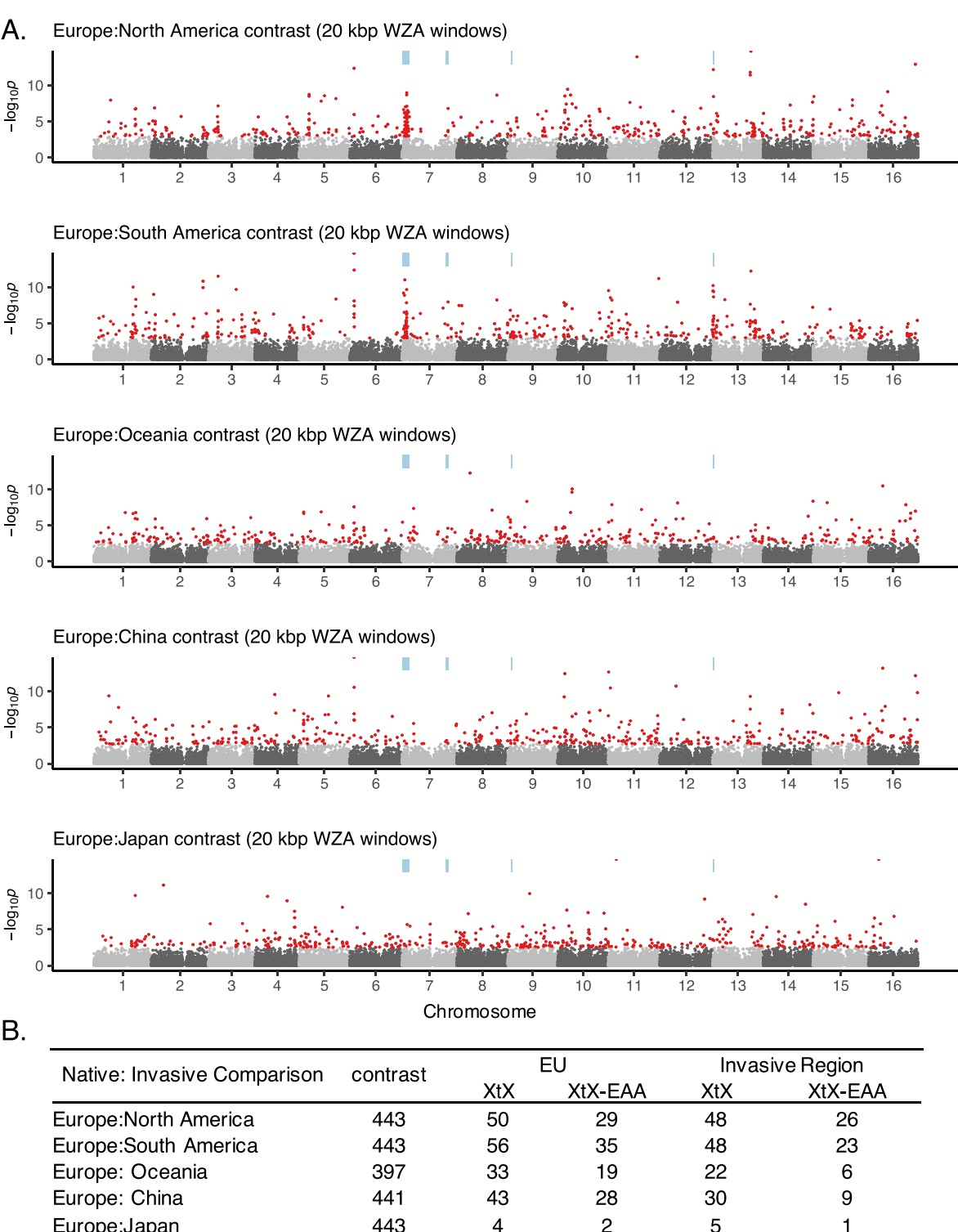

| Native: Invasive Comparison | contrast | EU | | Invasive Region | |
|---|---|---|---|---|---|
| | | XtX | XtX-EAA | XtX | XtX-EAA |
| Europe:North America | 443 | 50 | 29 | 48 | 26 |
| Europe:South America | 443 | 56 | 35 | 48 | 23 |
| Europe: Oceania | 397 | 33 | 19 | 22 | 6 |
| Europe: China | 441 | 43 | 28 | 30 | 9 |
| Europe:Japan | 443 | 4 | 2 | 5 | 1 |

**Extended Data Fig. 4 | Genome scan for differentiated regions between Europe and each invasive range. a.** Empirical $p$-values for enrichment of $C_2$ (contrast) in 20 kbp windows using the WZA between Europe and each invasive range. Red points indicate the top 1% of WZA scores. Blue bars indicate haploblock locations.

**b.** Overlap between contrast outlier 20 kbp windows and outlier 20 kbp windows within each introduced range (XtX) or across climatic gradients within each range (XtX-EAA).

**A.**

| All | Bio1 | Bio2 | Bio8 | Bio12 | Bio15 | Bio19 |
|---|---|---|---|---|---|---|
| HB7a1 | 0.139 | 0.013 | 0.263 | 0.188 | 0.022 | 0.045 |
| HB7a2 | 0.009 | 0.259 | 0.173 | 0.149 | 0.336 | -0.06 |
| HB7b | -0.17 | 0.024 | -0.2 | 0.007 | -0.04 | 0.096 |
| HB9 | -0.05 | 0.022 | 0.011 | -0.07 | 0.008 | -0.09 |
| HB13 | 0.099 | 0.032 | 0.113 | 0.101 | 0.102 | -0.01 |

| Europe | Bio1 | Bio2 | Bio8 | Bio12 | Bio15 | Bio19 |
|---|---|---|---|---|---|---|
| HB7a1 | -0.5 | 0.046 | 0.595 | -0.14 | 0.351 | -0.63 |
| HB7a2 | -0.08 | 0.295 | 0.357 | 0.388 | 0.171 | 0.016 |
| HB7b | 0.107 | -0.17 | -0.23 | 0.443 | -0.11 | 0.534 |
| HB9 | 0.242 | 0.545 | 0.061 | 0.121 | 0.182 | 0.061 |
| HB13 | -0.44 | -0.11 | 0.137 | -0.23 | 0.321 | -0.26 |

| North America | Bio1 | Bio2 | Bio8 | Bio12 | Bio15 | Bio19 |
|---|---|---|---|---|---|---|
| HB7a1 | 0.273 | 0.236 | 0.273 | -0.16 | 0.164 | -0.13 |
| HB7a2 | -0.24 | 0.018 | -0.02 | -0.09 | 0.236 | -0.35 |
| HB7b | -0.29 | 0.212 | -0.14 | -0.25 | -0.14 | -0.21 |
| HB9 | 0.367 | 0.257 | -0.15 | 0.073 | -0.15 | -0.04 |
| HB13 | 0.055 | 0.164 | -0.24 | -0.02 | 0.164 | 0.091 |

| South America | Bio1 | Bio2 | Bio8 | Bio12 | Bio15 | Bio19 |
|---|---|---|---|---|---|---|
| HB7a1 | 0.227 | -0.13 | 0.378 | 0.428 | -0.03 | 0.025 |
| HB7a2 | 0.244 | -0.2 | 0.556 | 0.333 | -0.11 | -0.29 |
| HB7b | -0.2 | 0.422 | -0.42 | -0.2 | 0.333 | 0.333 |
| HB9 | -0.18 | 0.09 | -0.23 | -0.05 | 0.09 | 0.045 |
| HB13 | 0 | -0.36 | 0.225 | 0.315 | -0.27 | -0.14 |

| Oceania | Bio1 | Bio2 | Bio8 | Bio12 | Bio15 | Bio19 |
|---|---|---|---|---|---|---|
| HB7a1 | 0.571 | 0.429 | 0.857 | 0.214 | 0.571 | 0.071 |
| HB7a2 | -0.18 | -0.11 | -0.04 | -0.11 | 0.182 | 0.036 |
| HB7b | -0.57 | -0.04 | -0.42 | -0.79 | -0.19 | -0.49 |
| HB9 | -0.19 | 0.189 | -0.04 | 0.113 | -0.11 | -0.19 |
| HB13 | 0.5 | 0.357 | 0.786 | 0.429 | 0.5 | 0.143 |

| China | Bio1 | Bio2 | Bio8 | Bio12 | Bio15 | Bio19 |
|---|---|---|---|---|---|---|
| HB7a1 | 0.236 | -0.24 | -0.24 | 0.236 | 0.236 | 0.236 |
| HB7a2 | -0.18 | 0.183 | -0.55 | -0.18 | 0.183 | -0.18 |
| HB7b | 0.236 | -0.24 | -0.24 | 0.236 | 0.236 | 0.236 |
| HB9 | 0.183 | -0.18 | 0.548 | 0.183 | -0.55 | 0.183 |
| HB13 | -1 | 1 | -0.67 | -1 | 0.333 | -1 |

| Japan | Bio1 | Bio2 | Bio8 | Bio12 | Bio15 | Bio19 |
|---|---|---|---|---|---|---|
| HB7a1 | -0.33 | -0.67 | -0.33 | 0 | 0 | 0.333 |
| HB7a2 | 0 | 0.333 | 0 | -0.33 | -0.33 | 0 |
| HB7b | 0 | -0.33 | 0 | 0.333 | -0.33 | 0.667 |
| HB9 | -0.67 | -1 | -0.67 | -0.33 | -0.33 | 0.667 |
| HB13 | -0.67 | -0.33 | -0.67 | -1 | -0.33 | 0 |

Kendall's Tau

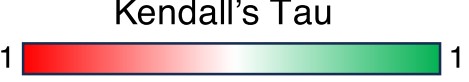

-1 1

**B.**

| Region | XtX | Bio1 | Bio2 | Bio8 | Bio12 | Bio15 | Bio19 | All |
|---|---|---|---|---|---|---|---|---|
| Europe | 446 | 82 | 75 | 66 | 32 | 52 | 72 | 232 |
| North America | 449 | 108 | 18 | 41 | 48 | 17 | 29 | 207 |
| South America | 448 | 27 | 30 | 56 | 71 | 10 | 17 | 140 |
| Oceania | 400 | 21 | 55 | 27 | 22 | 27 | 19 | 116 |
| China | 448 | 26 | 26 | 32 | 26 | 16 | 26 | 69 |
| Japan | 449 | 28 | 30 | 28 | 27 | 32 | 23 | 110 |

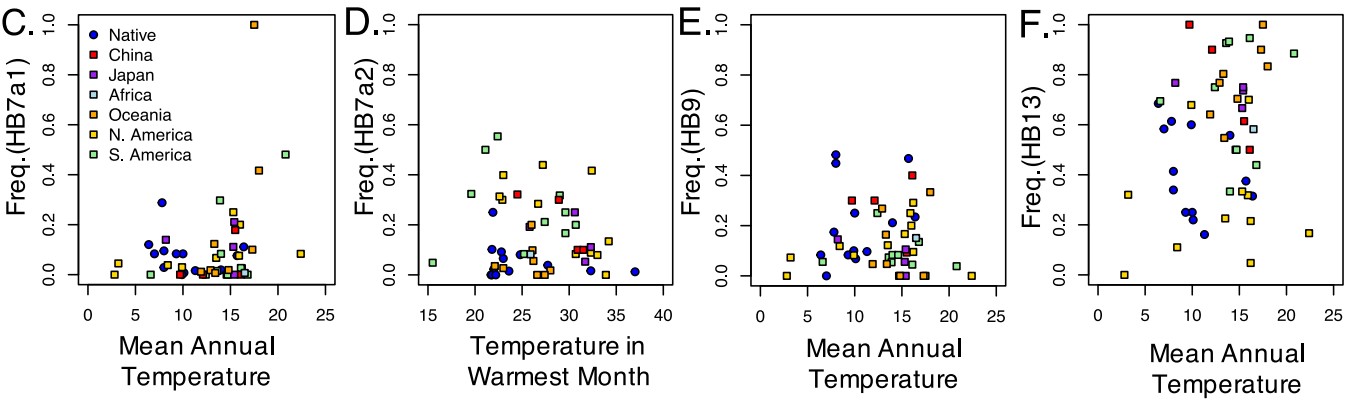

Extended Data Fig. 5 | See next page for caption.

**Extended Data Fig. 5 | Climatic correlations with haploblocks and genome-wide variation. a**. Associations between haploblocks and climatic variables from the WORLDCLIM database. **b**. Overlap between outlier 20kbp XtX windows within each introduced range and outlier windows associated with each climatic variable. Abbreviations: Bio1: Annual Mean Temperature, Bio2: Mean Diurnal Range, Bio8: Temperature in the Wettest Quarter, Bio12: Annual Precipitation,

Bio15: Precipitation Seasonality, Bio19: Precipitation in the Coldest Quarter. **c–f**. Relationships between haploblock allele frequencies and climatic variables. Each point is a single population and is color coded by native or introduced region. Allele frequencies correspond to the frequency of the alternative (non-reference) allele.

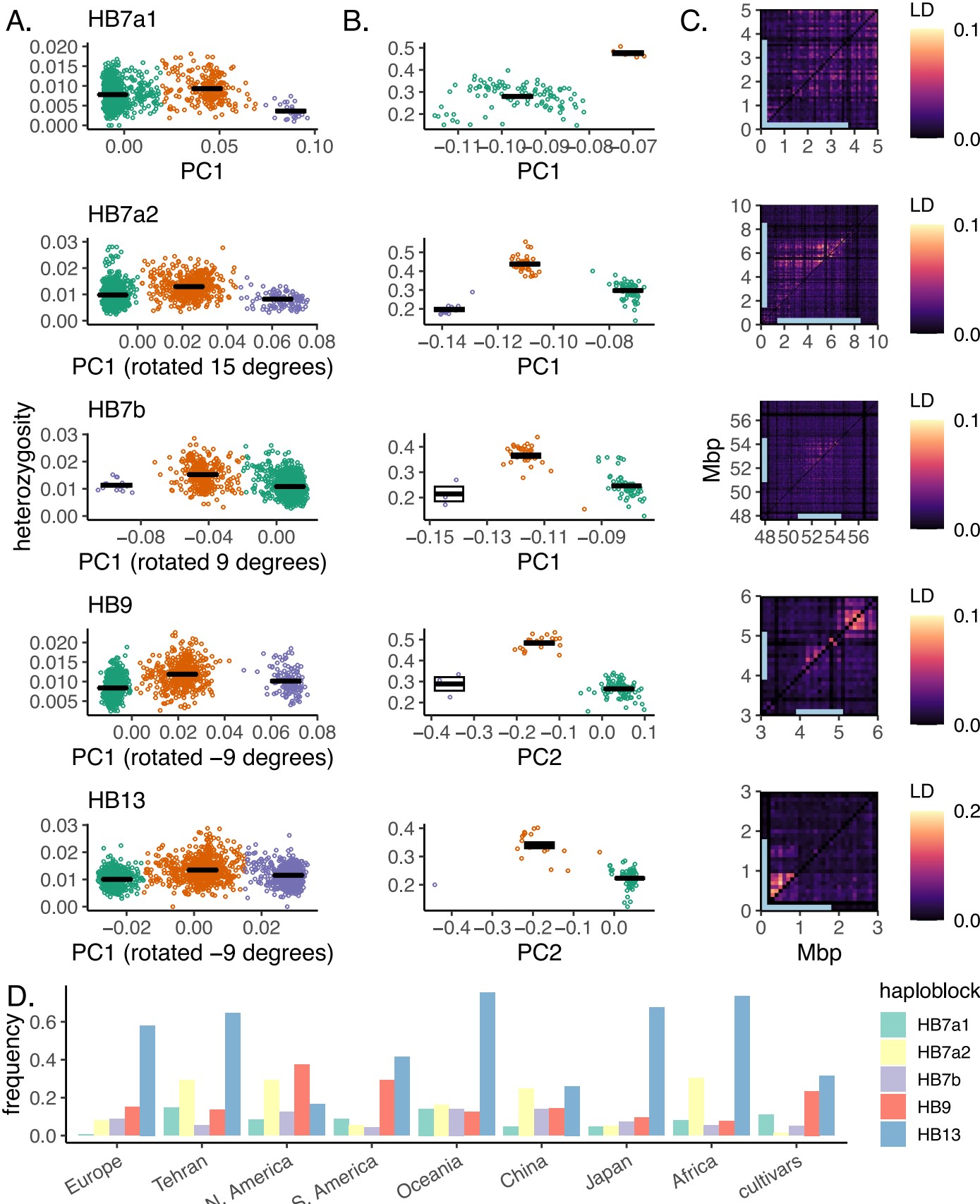

**Extended Data Fig. 6 | See next page for caption.**

**Extended Data Fig. 6 | Five haploblocks–population-genomic signatures of structural variants. a**. Haploblock clusters for the worldwide population genomics dataset. Three clusters indicative of two homozygous (green and purple) and one heterozygous (orange) structural variant genotypes separate along the first principal component of genetic variation across each haploblock, and furthermore putative heterozygotes show significantly elevated heterozygosity (two-sided Wilcoxon test p < 0.0003 for all heterozygote vs. homozygote comparisons; boxes denote mean ± SEM for each cluster). For better visualization, y-axes have been cropped removing 7, 1, 1, 0 and 6 outlier points for hb7a1, hb7a2, hb7b, hb9 and hb13 plots respectively. Sample sizes (clusters from left to right): HB7a1 1825/250/29, HB7a2 1528/462/111, HB7b 1765/324/15, HB9 1525/441/130, HB13 688/783/633. **b**. Haploblock clusters are apparent using local PCAs and heterozygosity values derived from SNP data for 109 higher-coverage (~10X) Toronto samples. Colors reflect genotypes assigned from the global dataset. Note that there were no samples genotyped as homozygous for the rare HB7a1 allele in Toronto. Boxes denote mean ± SEM for each cluster. Sample sizes (clusters from left to right): HB7a1 109/9/0, HB7a2 64/41/13, HB7b 80/35/3, HB9 93/22/3, HB13 93/24/1. **c**. Local patterns of linkage disequilibrium (the second highest value in each 100kb window) corresponding to haploblock regions (blue bars) are present in a random sample of individuals (top triangle; matching sample size of bottom triangle) but absent in samples homozygous for the common haploblock allele (bottom triangle). **d**. Estimated allele frequencies of each haploblock.

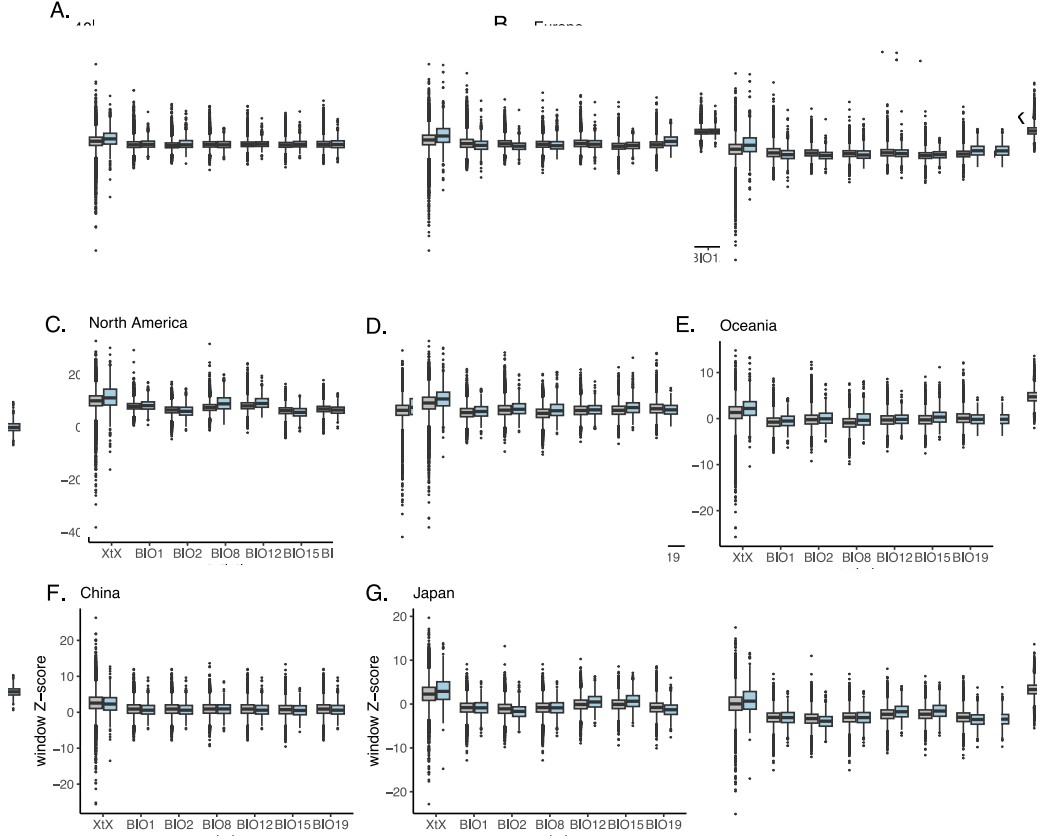

**Extended Data Fig. 7 | Comparisons of selective signatures within haploblock and non-haploblock windows across the genome. a**. Distribution of WZA 20 kbp window scores for contrasts between Europe and each invasive range for non-haploblock (gray) and haploblock (light blue) windows. **b–g**. Distribution of XtX statistics and Kendall's Tau for several climatic variables from the WorldClim dataset for each region. For boxplots, box edges represent the interquartile range, the center line in the box is the median, and the whiskers represent 1.5 times less or greater than the interquartile range. Abbreviations: Bio1: Annual Mean Temperature, Bio2: Mean Diurnal Range, Bio8: Temperature in the Wettest Quarter, Bio12: Annual Precipitation, Bio15: Precipitation Seasonality, Bio19: Precipitation in the Coldest Quarter. EU = Europe, nAM = North America, sAM = South America, OC = Oceania, CN = China, and JP = Japan. Sample sizes (non-haploblock/ haploblock): A: EU-nAM 42694/703, EU-sAM 42679/706, EU-OC 35184/560, EU-CN 42517/696, EU-JP 42788/703; B: 43915/728; C: 44235/731; D: 44099/728; E: 39375/650; F: 44162/727; G: 44257/731.

| Structural Variant | Fitness Measure | Genotype | | | Garden | | | Genotype:Garden | | |
|---|---|---|---|---|---|---|---|---|---|---|
| | | $X^2$ | df | p-value | $X^2$ | df | p-value | $X^2$ | df | p-value |
| HB7a1 | Survival (1st Year) | 0.1 | 1 | 0.751 | **60.15** | **3** | **5.46E-13** | 2.93 | 3 | 0.402 |
| HB7a2 | Survival (1st Year) | 0.56 | 1 | 0.454 | **50.92** | **3** | **5.08E-11** | **7.46** | **3** | **0.059** |
| HB7b | Survival (1st Year) | 0.08 | 1 | 0.773 | **58.86** | **3** | **1.03E-12** | 5.75 | 3 | 0.124 |
| HB9 | Survival (1st Year) | 0.28 | 1 | 0.595 | **44.06** | **3** | **1.47E-09** | 0.8 | 3 | 0.851 |
| HB13 | Survival (1st Year) | 0.05 | 1 | 0.825 | **30.43** | **3** | **1.12E-06** | 1.02 | 3 | 0.797 |
| | | | | | | | | | | |
| HB7a1 | Survival (Flowering) | 2.32 | 1 | 0.128 | **70.53** | **3** | **3.28E-15** | 5.72 | 3 | 0.126 |
| HB7a2 | Survival (Flowering) | 0.01 | 1 | 0.944 | **60.33** | **3** | **5.00E-13** | 2.31 | 3 | 0.51 |
| HB7b | Survival (Flowering) | 0.77 | 1 | 0.38 | **64.63** | **3** | **6.03E-14** | 2.87 | 3 | 0.412 |
| HB9 | Survival (Flowering) | 0.16 | 1 | 0.692 | **53.15** | **3** | **1.70E-11** | 2.15 | 3 | 0.543 |
| HB13 | Survival (Flowering) | 0.07 | 1 | 0.787 | **35.15** | **3** | **1.13E-07** | 0.79 | 3 | 0.852 |
| | | F | df | p-value | F | df | p-value | F | df | p-value |
| HB7a1 | Total Seed Mass | No Variation in Sweden Garden | | | | | | | | |
| HB7a2 | Total Seed Mass | **7.41** | **1** | **0.007** | **7.08** | **3** | **1.54E-04** | 2.31 | 3 | 0.078 |
| HB7b | Total Seed Mass | 0.04 | 1 | 0.849 | **6.66** | **3** | **2.65E-04** | 0.47 | 3 | 0.703 |
| HB9 | Total Seed Mass | 0.77 | 1 | 0.382 | **7.04** | **3** | **1.63E-04** | 0.85 | 3 | 0.47 |
| HB13 | Total Seed Mass | 7.75 | 1 | 0.006 | **4.76** | **3** | **0.003** | 7.65 | **3** | **7.43E-05** |
| | | | | | | | | | | |
| HB7a1 | Absolute Fitness | 0.32 | 1 | 0.57 | **32.29** | **3** | **2.20E-16** | **2.8** | **3** | **0.04** |
| HB7a2 | Absolute Fitness | 1.2 | 1 | 0.274 | **30.2** | **3** | **2.20E-16** | 1.86 | 3 | 0.135 |
| HB7b | Absolute Fitness | 1.14 | 1 | 0.286 | **31.29** | **3** | **2.20E-16** | **2.56** | **3** | **0.054** |
| HB9 | Absolute Fitness | 0.04 | 1 | 0.841 | **22.67** | **3** | **1.03E-13** | 0.35 | 3 | 0.79 |
| HB13 | Absolute Fitness | 1.33 | 1 | 0.249 | **6.68** | **3** | **2.03E-04** | 9.59 | **3** | **3.74E-06** |

**Extended Data Fig. 8 | Associations between fitness variables and haploblock genotypes across four common gardens.** ANOVA results based on a type-III sum-of-squares and two-sided tests. Both survival measures were modeled within a generalized linear model with a binomial distribution and logit link function. Total Seed Mass does not include individuals that did not survive to flowering. Absolute fitness is measured as total seed mass with individuals not surviving to flowering having zero total seed mass. Total seed mass and relative fitness were log+1 transformed to improve model fit. Bold values indicate statistically significant factors at p < 0.05.

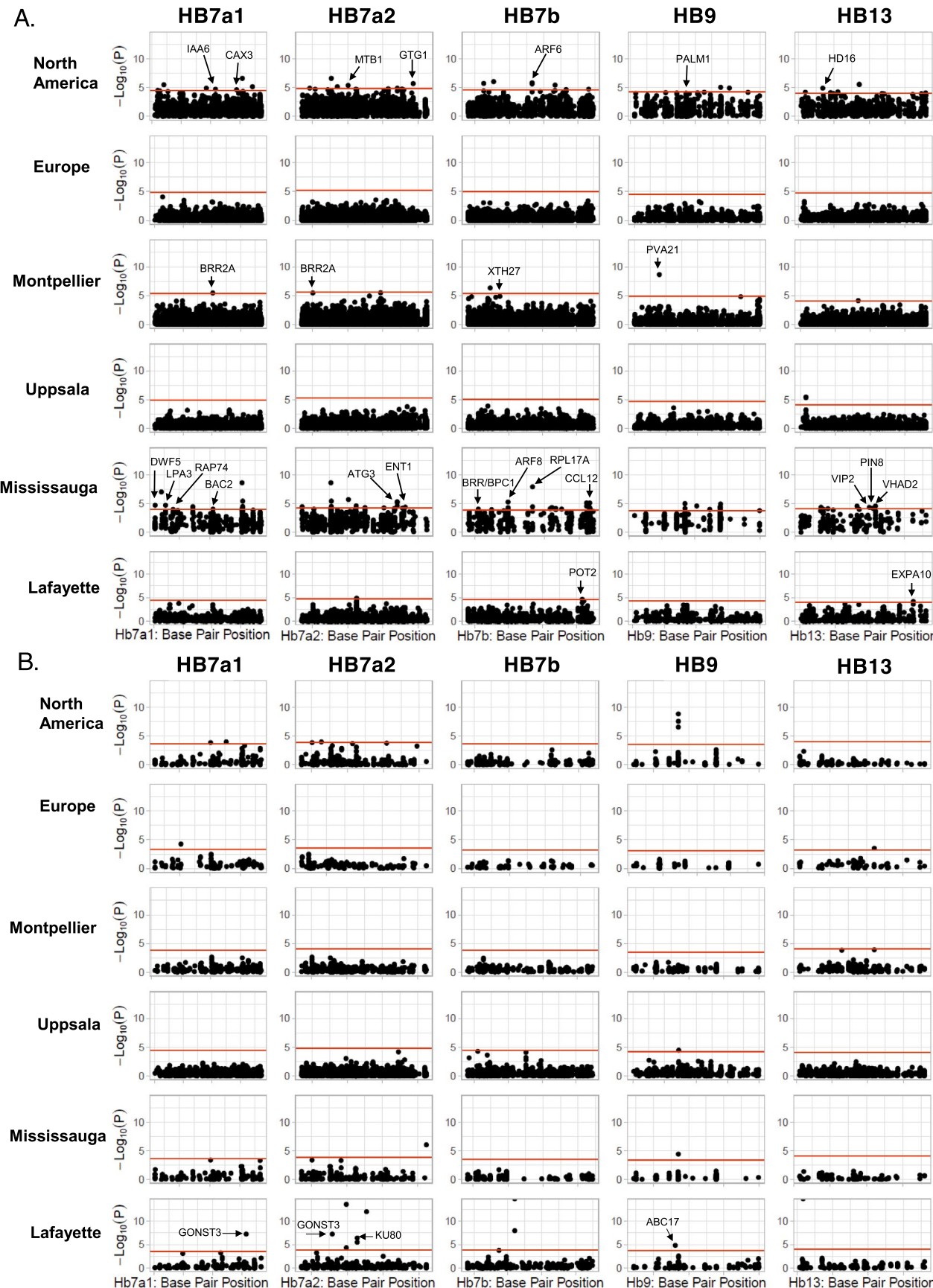

**Extended Data Fig. 9 | Manhattan plots summarizing associations with fitness across haploblocks. a**. Associations between haploblock genotypes with survival to flowering. **b**. Associations between haploblock genotypes with total seed mass (including zeros for plants that did not survive to flowering). P-values correspond to score statistic (a two-sided test) and are not corrected for multiple comparisons. The Bonferroni corrected significance threshold (horizontal red line) is specific to each haploblock and garden. Gene names are given only for hits landing within coding sequence of annotated genes.

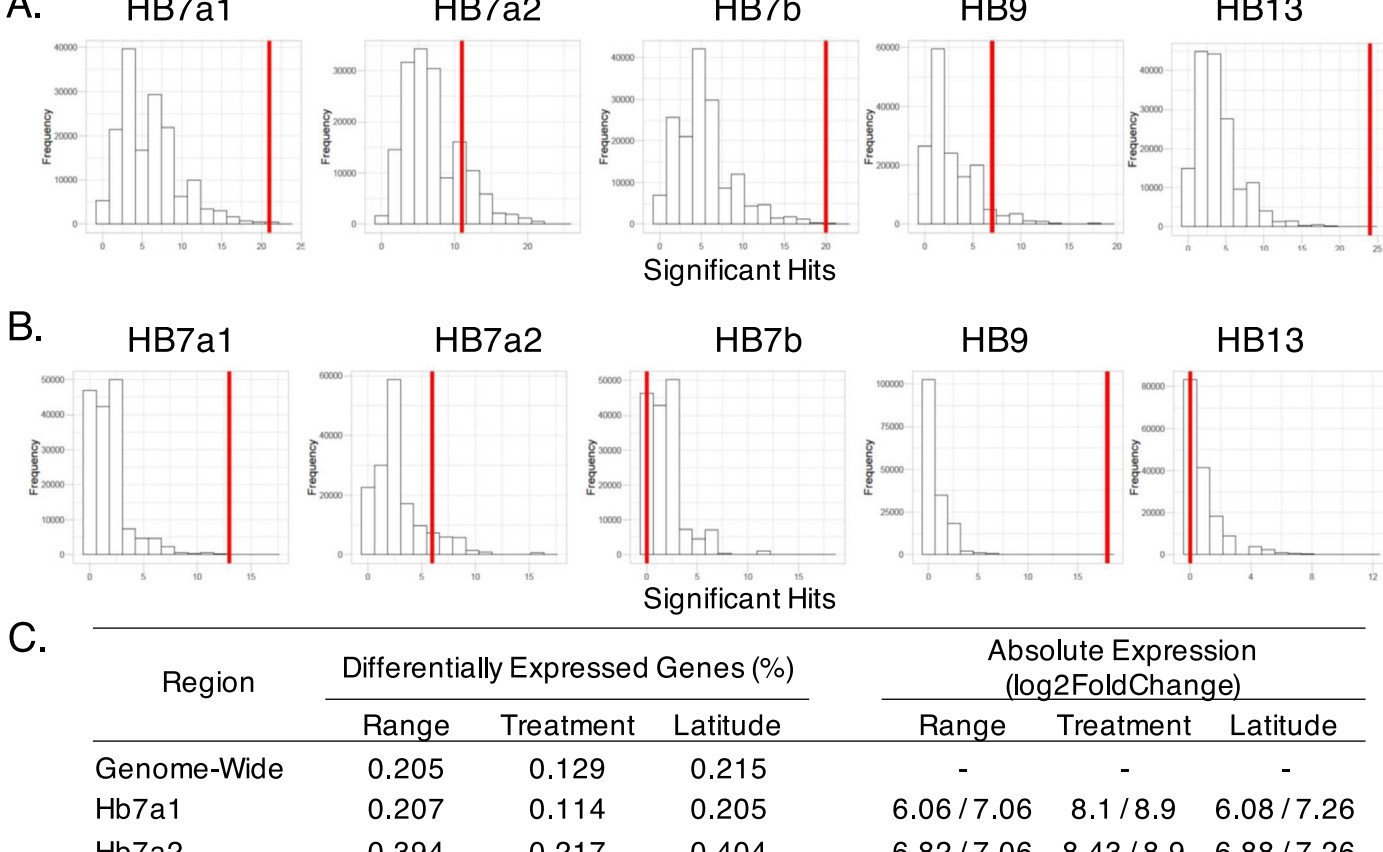

C.

| Region | Differentially Expressed Genes (%) | | | Absolute Expression (log2FoldChange) | | |
|---|---|---|---|---|---|---|
| | Range | Treatment | Latitude | Range | Treatment | Latitude |
| Genome-Wide | 0.205 | 0.129 | 0.215 | - | - | - |
| Hb7a1 | 0.207 | 0.114 | 0.205 | 6.06 / 7.06 | 8.1 / 8.9 | 6.08 / 7.26 |
| Hb7a2 | 0.394 | 0.217 | 0.404 | 6.82 / 7.06 | 8.43 / 8.9 | 6.88 / 7.26 |
| Hb7b | 0.142 | 0.081 | 0.136 | 7.14 / 7.06 | 8.28 / 8.88 | 6.53 / 7.26 |
| Hb9 | 0.053 | 0.035 | 0.053 | 5.44 / 7.04 | 7.58 / 8.88 | 5.71 / 7.27 |
| Hb13 | 0.087 | 0.043 | 0.093 | 7.48 / 7.07 | 8.94 / 8.87 | 7.8 / 7.26 |

**Extended Data Fig. 10 | Characterization of the genic content of haploblock regions. a, b**. Histograms summarizing expected numbers of GWAS hits for each haploblock for survival to flowering (**a**) and total seed mass (**b**) within the North American common gardens across 160,000 simulations. Observed number of GWAS hits are displayed as vertical red lines. **c**. Differential expression analysis of RNAseq data within each haploblock. Percentage of genes differentially expressed and absolute expression (log₂FoldChange) is presented between ranges (Europe vs. North America), between Treatments (Well-Watered vs. Dry Down), and between Latitudes (Low vs. High). Values on either side of paratheses for absolute expression are the observed / expected values. Expected values are derived from a two-sided permutation analysis that re-sampled regions from across the genome with replacement. P-values are greater than 0.05 in all cases.

# Reporting Summary

## Statistics

For all statistical analyses, confirm that the following items are present in the figure legend, table legend, main text, or Methods section.

| n/a | Confirmed | |
|---|---|---|
| ☐ | ☒ | The exact sample size (*n*) for each experimental group/condition, given as a discrete number and unit of measurement |
| ☐ | ☒ | A statement on whether measurements were taken from distinct samples or whether the same sample was measured repeatedly |
| ☐ | ☒ | The statistical test(s) used AND whether they are one- or two-sided<br>*Only common tests should be described solely by name; describe more complex techniques in the Methods section.* |
| ☐ | ☒ | A description of all covariates tested |
| ☐ | ☒ | A description of any assumptions or corrections, such as tests of normality and adjustment for multiple comparisons |
| ☐ | ☒ | A full description of the statistical parameters including central tendency (e.g. means) or other basic estimates (e.g. regression coefficient) AND variation (e.g. standard deviation) or associated estimates of uncertainty (e.g. confidence intervals) |
| ☐ | ☒ | For null hypothesis testing, the test statistic (e.g. $F$, $t$, $r$) with confidence intervals, effect sizes, degrees of freedom and $P$ value noted<br>*Give P values as exact values whenever suitable.* |
| ☒ | ☐ | For Bayesian analysis, information on the choice of priors and Markov chain Monte Carlo settings |
| ☒ | ☐ | For hierarchical and complex designs, identification of the appropriate level for tests and full reporting of outcomes |
| ☐ | ☒ | Estimates of effect sizes (e.g. Cohen's *d*, Pearson's *r*), indicating how they were calculated |

*Our web collection on statistics for biologists contains articles on many of the points above.*

## Software and code

Policy information about availability of computer code

| Data collection | Climate data for sampling locations from the WorldClim dataset was accessed via the raster v3.6-26 package in R v4.2.2 |
|---|---|
| Data analysis | All code from this manuscript is linked to a github repository (github.com/pbattlay/glue-invasions). Briefly, all low coverage whole genome sequences were trimmed, aligned, and processed using fastp v23.2, bwa mem v0.7.17, and samtools v1.10. Quality control on each bamfile was performed using Qualimap v2.2.2, Bamtools v2.5.1, and multiQC v1.14. Population genomic diversity, differentiation, and structure using genotype likelihood approaches implemented in ANGSD v0.929. R v4.2.2 was used for all statistical comparisons including comparisons of summary statistics between ranges, Mantel tests, and PERMANOVAs. EPOS was used to assess effective population size through time. NGSadmix was used to calculate admixture coefficients and PCAngsd was used to conduct principal component analysis. BayPass was used for contrast, outlier, and gene environmental association analyses. Local PCA analysis and haploblocks were determined using customize code based off of lostruct software. Linkage disequilibrium was assessed using ngsLD v1.2.0. Gene ontology analysis was conducted using topGO v2.50.0 in R v4.2.2.<br><br>RNAseq reads trimmed using fastp v023.4. Microbial RNA contamination was removed by aligning to a custom database using bowtie2 v2.5.1. A transcriptome was created using gffread v0.12.7. Read mapping and quantification was done using Salmon v1.10.2. Differential expression analysis was assessed using DESeq2. Custom permutation code was written in R v4.2.2. |

For manuscripts utilizing custom algorithms or software that are central to the research but not yet described in published literature, software must be made available to editors and reviewers. We strongly encourage code deposition in a community repository (e.g. GitHub). See the Nature Portfolio guidelines for submitting code & software for further information.

# Data

Policy information about availability of data

All manuscripts must include a data availability statement. This statement should provide the following information, where applicable:

- Accession codes, unique identifiers, or web links for publicly available datasets
- A description of any restrictions on data availability
- For clinical datasets or third party data, please ensure that the statement adheres to our policy

Low coverage whole genome sequences (fastq files) for all accessions can be found as .fq files in the NCBI SRA database (Bioprojects: PRJNA1081485, PRJNA1179961). Metadata and fitness data from the four-way common garden study can be found on Dryad: https://datadryad.org/stash/share/hBvXInXHWCab2P1kIGKWALwX-Mz9KM6TYLPn9LAySS4, and associated low coverage whole genome sequences are in the NCBI SRA database (Biooproject: PRJNA1098360). Raw fastq files from RNAseq expression experiment can be found in the NCBI SRA database (Bioproject: PRJNA1131002). All code from this manuscript is linked to a github repository (github.com/pbattlay/glue-invasions).

# Research involving human participants, their data, or biological material

Policy information about studies with human participants or human data. See also policy information about sex, gender (identity/presentation), and sexual orientation and race, ethnicity and racism.

| Reporting on sex and gender | Not applicable |
|---|---|
| Reporting on race, ethnicity, or other socially relevant groupings | Not applicable |
| Population characteristics | Not applicable |
| Recruitment | Not applicable |
| Ethics oversight | Not applicable |

Note that full information on the approval of the study protocol must also be provided in the manuscript.

# Field-specific reporting

Please select the one below that is the best fit for your research. If you are not sure, read the appropriate sections before making your selection.

☐ Life sciences  ☐ Behavioural & social sciences  ☒ Ecological, evolutionary & environmental sciences

For a reference copy of the document with all sections, see nature.com/documents/nr-reporting-summary-flat.pdf

# Ecological, evolutionary & environmental sciences study design

All studies must disclose on these points even when the disclosure is negative.

| Study description | We combine a population genomic analysis across worldwide populations of white clover (Trifolium repens L.) with genome-wide association analysis of fitness within native and introduced populations conducted within four field common garden plots (low and high latitude gardens in the European and North American ranges), and RNAseq analysis in a manipulative dry-down experiment. |
|---|---|
| Research sample | The population genomic dataset consists of 2660 samples including 2648 samples from six continents. The majority of these samples (2616) come from sampling around 50 cities. Additional samples come from sampling around 4 cities in Spain as well as 12 popular cultivars.<br><br>The field common garden experiments utilized seeds derived from 93 populations collected along latitudinal gradients across North America and Europe. Seeds from each population were originally pooled from >20 individuals collected >3m apart. Field seeds were planted in a refresher generation and then outcrossed within populations to produce the lines grown in each common garden. Sequence data was obtained from 656 total individuals from these gardens.<br><br>The manipulative dry-down experiment included 51 individuals coming from field seed from 14 of the populations collected across latitudinal gradients in North America and Europe. |
| Sampling strategy | For the population genomics from low coverage whole genome sequencing, samples sizes from each population ranged from 5-120. We included extensively sampled populations for better estimates of site frequency spectra and population-genomic statistics (31 populations; Ave. = 80.74, Std. = 17.7 individuals) and added additional cities with lower sampling that we deemed as important areas for understanding colonization history (19 cities; Ave. = 5.95). We chose a sequencing coverage of ~1X for efficient, accurate and precise estimation of our various summary statistics of interest based on published low coverage whole genome sequencing |

simulations. We downsampled cities that were outliers for coverage.

We sequenced every individual that we could obtain DNA from the field common garden experiments. Of 2000 individuals originally planted, we only were able to obtain DNA from 656. The number of samples from each range (Europe: 190 individuals, North America: 465) is sufficient to detect loci with moderate effect sizes within genome-wide association studies. Sequencing coverage paralleled the population genomic analysis above to provide a compatible dataset for detecting haploblocks.

Our RNAseq experimental design included a control and a dry down treatment. In each treatment, we planted 3 lines from each of the 16 populations (4 population from the low and high latitude populations in the native (Europe) and introduced ranges (North American). Our final sample size for RNAseq included 1-3 individuals per population in each of the treatments due to poor germination. This design provides sufficient biological replication within latitudes, ranges, and across treatments to assess differential expression between each contrast.

| | |
|---|---|
| Data collection | Data collection for 2616 samples for the population genomics was done through the Global Urban Evolution Network (GLUE) by fellow scientists across the world. The collections for this dataset were first reported in Santangelo et al. 2022 (DOI: 10.1126/science.abk0989). Samples for populations from Spain was done by Simon Innes in 2021. The collection of these samples are first reported in Innes et al. 2022 (DOI: 10.1111/evo.14514). Sequencing and bioinformatic analysis was completed by Brandon Hendrickson, Jonas Mendez-Reneau, Paul Battlay, James Santangelo, Jonathan Wilson, Aude E. Caizergues, and Nicholas Kooyers.

Data collection for field common gardens was completed in North American Gardens from March 2020 to September 2021. Work in the Lafayette common garden was led by Nevada King and Courtney Patterson. Work in the Toronto common garden was led by Lucas Albano. Data collection in the European common gardens was conducted from March 2021-September 2022. Work in the Montpellier common garden was led by Cyrille Violle and Francois Vasseur. Work in the Uppsala common garden was led by Adriana Puentes and Amelia Tudoran. Analysis was done by Lucas Albano, Brandon Hendrickson, and Nicholas Kooyers.

RNAseq experiments were conducted in growth chambers from January 2020 – April 2020 with data collection by Caitlyn Stamps and Courtney Patterson. RNA extraction and library construction was completed by Hunter Strickland and Paul Kim. Bioinformatic analysis was completed by Michael Foster, Brandon Hendrickson, and Nicholas Kooyers. |
| Timing and spatial scale | Population genomics samples are collected from worldwide populations as described above.

Seed generation and sourcing for the field common gardens is described above. Field common gardens in North American Gardens ran from March 2020 to September 2021, while the European common garden was conducted from March 2021-September 2022. Data collection was performed weekly, biweekly, monthly or annually depending on which phenotype was being assessed. Data collection was only performed during the growing season.

Samples used in the manipulative growth chamber experiment were obtained from natural population in Europe and North America as described above. |
| Data exclusions | For the population genomics analysis, we excluded 8 samples based on both low coverage (below 0.5x) and quality of sequence data. These samples are not included within any numbers reported in the manuscript (i.e. we started with 2669 samples)

We had sequence data from 656 samples from the field common garden experiments; however, we excluded samples that have very limited numbers of reads post-sequencing. We had sufficient coverage and quality to call genotypes of haploblocks for 586 individuals and sufficient coverage (>0.5X) to conduct GWAS analysis for 569 plants.

We excluded a single individual from the RNAseq analysis as this individual was much larger than the others at the beginning of the experiment. We excluded three other samples from RNAseq analysis that had low read counts. Our total sample size was 47 individuals. |
| Reproducibility | Given the nature of our experiments (worldwide population genomics sampling, multiyear field experiments), cost (manipulative experiment with RNAseq), or availability of additional seed stock (manipulative experiment with RNAseq), we did not replicate our experiments. |
| Randomization | Planting location was randomized within each common garden for the field common garden experiments. Location of pots were randomized within treatment in the manipulative growth chamber experiment. |
| Blinding | Researchers were blinded from knowing the associated metadata (i.e. population or range origin) while collecting data as each plant was given a plant ID based on the randomized location within gardens or treatment prior to planting. |

Did the study involve field work?  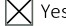 ☒ Yes  ☐ No

# Field work, collection and transport

| | |
|---|---|
| Field conditions | Temperature and precipitation varied both seasonally and between growing seasons. Generally, temperatures were slightly higher than historic climatic norms. There were no extreme events that impacted our gardens. |

| Location | Coordinates of common garden locations are 42°32'40" N, 79°39'38" W (Mississauga, Ontario, Canada), 30°18'23" N, 92°00'3 3" W (Lafayette, Louisiana, USA), 59°49'08" N, 17°38'49" W (Uppsala, Sweden), and 43°38'16" N, 3°51'43" W (Montpellier, France). |
|---|---|
| Access & import/export | All collections were done with permission of the local authorities as described in Santangelo et al. 2022 (DOI: 10.1126/ science.abk0989),  Innes et al. 2022 (DOI: 10.1111/evo.14514) and Albano et al. 2024 (DOI: 10.1101/2024.09.03.611023). Importation of tissue and seed between the EU and Canada as well as between Canada and United States went through required phytosanitary pathways by Canadian Food Inspection Agency or/and the USDA. |
| Disturbance | Common gardens were conducted within field stations in grassy areas that already had substantial human disturbance as white clover grows within lawns and grasslands. Landscape tarps were removed following the common gardens. |

# Reporting for specific materials, systems and methods

We require information from authors about some types of materials, experimental systems and methods used in many studies. Here, indicate whether each material, system or method listed is relevant to your study. If you are not sure if a list item applies to your research, read the appropriate section before selecting a response.

## Materials & experimental systems

| n/a | Involved in the study |
|---|---|
| ☒ ☐ | Antibodies |
| ☒ ☐ | Eukaryotic cell lines |
| ☒ ☐ | Palaeontology and archaeology |
| ☒ ☐ | Animals and other organisms |
| ☒ ☐ | Clinical data |
| ☒ ☐ | Dual use research of concern |
| ☐ ☒ | Plants |

## Methods

| n/a | Involved in the study |
|---|---|
| ☒ ☐ | ChIP-seq |
| ☒ ☐ | Flow cytometry |
| ☒ ☐ | MRI-based neuroimaging |

## Dual use research of concern

Policy information about dual use research of concern

### Hazards

Could the accidental, deliberate or reckless misuse of agents or technologies generated in the work, or the application of information presented in the manuscript, pose a threat to:

| No | Yes | |
|---|---|---|
| ☒ | ☐ | Public health |
| ☒ | ☐ | National security |
| ☒ | ☐ | Crops and/or livestock |
| ☒ | ☐ | Ecosystems |
| ☒ | ☐ | Any other significant area |

### Experiments of concern

Does the work involve any of these experiments of concern:

| No | Yes | |
|---|---|---|
| ☒ | ☐ | Demonstrate how to render a vaccine ineffective |
| ☒ | ☐ | Confer resistance to therapeutically useful antibiotics or antiviral agents |
| ☒ | ☐ | Enhance the virulence of a pathogen or render a nonpathogen virulent |
| ☒ | ☐ | Increase transmissibility of a pathogen |
| ☒ | ☐ | Alter the host range of a pathogen |
| ☒ | ☐ | Enable evasion of diagnostic/detection modalities |
| ☒ | ☐ | Enable the weaponization of a biological agent or toxin |
| ☒ | ☐ | Any other potentially harmful combination of experiments and agents |

## Plants

| | |
|---|---|
| Seed stocks | Collection locations and procedures for samples collected from the GLUE network are reported in Santangelo et al. 2022 (DOI: 10.1126/science.abk0989). Individuals were sampled from multiple urban and rural locations surrounding a focal city. Collection locations and procedures for the field common garden are described in ), Innes et al. 2022 (DOI: 10.1111/evo.14514) and Albano et al. 2024 (DOI: 10.1101/2024.09.03.611023). Individuals were sampled from a single locality for each population in this work and field seed from each location was pooled from ≥20 individuals. Field seed was grown and outcrossed within populations in a greenhouse refresher generations. For both studies, care was taken to sample different genets in the field (i.e. sampling plant at least 3m apart). |
| Novel plant genotypes | Not applicable |
| Authentication | Cultivar lines were obtained from the USDA GRIN database. Specific accessions were: PI 419973, PI 376882, PI 430569, NSL 5462, NSL 6517, NSL 4784, PI 631875, NSL 186524, PI 5459. Three other accessions (Durana, Patriot, and Renovation) were obtained from commercial horticulture distributors. |

