## [Peer Review File · Nature Ecology & Evolution]

Haploblocks contribute to parallel climate adaptation following global invasion of a cosmopolitan plant

Corresponding Author: Dr Nicholas Kooyers

Version 0:

Decision Letter:

16th January 2025

Dear Nic,

Your manuscript entitled "Structural variants underlie parallel adaptation to climate following global invasion of a cosmopolitan plant" has now been seen by three reviewers, whose comments are attached. The reviewers have raised a number of concerns which will need to be addressed before we can offer publication in Nature Ecology & Evolution. We will therefore need to see your responses to the criticisms raised and to some editorial concerns, along with a revised manuscript, before we can reach a final decision regarding publication.

We therefore invite you to revise your manuscript taking into account all reviewer and editor comments. Please highlight all changes in the manuscript text file.

* If you have not done so already please begin to revise your manuscript so that it conforms to our Article format instructions at <http://www.nature.com/natecolevol/info/final-submission>. Refer also to any guidelines provided in this letter.

* Extended Data Figures - please ensure that any supplementary figures and tables that are crucial to the manuscript's conclusions are converted into Extended Data figures and tables to increase visibility of these data. Extended Data figures and tables are online-only (present in the online PDF and full-text HTML versions of the paper), peer-reviewed display items that provide essential background to the article but are not included in the main article due to space constraints. A maximum of ten Extended Data display items (figures and tables) is permitted.

Link Redacted

Nature Ecology & Evolution is committed to improving transparency in authorship. As part of our efforts in this direction, we are now requesting that all authors identified as 'corresponding author' on published papers create and link their Open Researcher and Contributor Identifier (ORCID) with their account on the Manuscript Tracking System (MTS), prior to acceptance. ORCID helps the scientific community achieve unambiguous attribution of all scholarly contributions. You can create and link your ORCID from the home page of the MTS by clicking on 'Modify my Springer Nature account'. For more information please visit www.springernature.com/orcid.

[redacted]

Reviewer expertise:

Reviewer #1: Genomics of adaptation, SVs

Reviewer #2: Genomics of adaptation, SVs

Reviewer #3: Genomics of invasions

Reviewers' comments:

Reviewer #1 (Remarks to the Author):

This manuscript thoroughly analyses the replicated adaptation of an introduced plant (white clover) in very different geographic areas. The span of sampling is astonishing with populations on all continents, and a mix of introduced and native. The analyses are up to expectations for such a dataset. By studying genetic diversity, genetic variation across populations, and association with the environment, the authors put in evidence some population structure and, to a certain extent, some repeatability in the genetic architecture of adaptation. Of particular interest are blocks of high linkage disequilibrium (haploblocks) which may be inversions and gather several candidate loci (and candidate genes) for adaptation.

On top of this study on sampled plants, the manuscript also presents a follow-up transplant experiment which was used to support a significant effect of haploblocks on fitness. In addition, a few candidate genes previously identified as involved in the fitness effect also showed differential RNA expression upon thermal/drought stress. Altogether, those multiple approaches provide very strong evidence for local adaptation to thermal and humidity gradient by white clover upon the colonisation of new areas with a few regions possibly corresponding to chromosomal rearrangements.

The manuscript is generally well-written with clear figures. Methods are very detailed (when reading both the manuscript and the supplementary notes), allowing reproducibility for most steps. The analyses are sound and follow state-of-the-art guidelines (but see a few sanity checks below). The main conclusions are mostly supported by the results although I have a few minor concerns. I mostly regretted the absence of a true discussion, which fails to enlighten us about the true impact of the manuscript. I am providing below a few suggestions to ensure the robustness of the results and replace its findings in a relevant context.

Thank you for inviting me to review this paper, which I enjoyed reading a lot.

Claire Mérot

• Major concerns:

- Absence of discussion: The downside of mixing results and discussion is usually that all discussed points are specific to the study system and do not adequately put the findings in perspective with the rest of the literature. Unfortunately, this is the case here. I think either adding a real discussion paragraph or enriching the conclusion would be very valuable to put in evidence as to why this study is impactful. For example, I was very surprised, given the conclusion that large haploblocks may underlie parallel adaptation to see nothing about the theoretical predictions (Here I think about Westram et al 2022 but there is much more literature on the topic, particularly on adaptation in the face of gene flow, etc). Similarly, the study focuses on white clover and does not mention comparable empirical studies. I can think of many examples of inversion clines in *Drosophila*, parallel adaptation in stickleback, parallel karyotype-environment associations in *Coelopa*, *Littorina* crab/wave examples, and even a study on *Ambrosia* plants by the same author(s). And this is only for the re-use of haploblocks or inversions in parallel adaptation. There is much more literature on parallel adaptation itself (and on introduced/invasive species)

Westram, A. M., Faria, R., Johannesson, K., Butlin, R., & Barton, N. (2022). Inversions and parallel evolution. *Philosophical Transactions of the Royal Society B*, 377(1856), 20210203.

- The LD bias is not really accounted for: this study put in evidence the presence of haploblocks which gather many outliers

in statistical analysis for adaptation. The problem with such haploblocks is the strong LD (due to limited recombination in heterokaryote if this is truly an inversion). So it is not surprising to find a high density of candidate, because as soon as one locus is picked, the other linked loci bear the same signal (due to LD!). This makes it really hard to interpret the enrichment in GWAS (ext Fig 10) and GEA (ext Fig 7) for haploblocks. At the same time, not all SNPs inside a block have a signal so... how should we interpret the density of putative candidates? I have no answer to this and no solution to sort true candidates and hitch-hicked loci inside haploblocks BUT it would be worth at least discussing it (and if the authors have a solution, even better!).

- A related matter about LD: the strong divergence and LD in haploblocks can also drive signals in NGSAdmix and PCA, even on 4-fold degenerate SNPs because SNPs in LD will drive the signal. As a sanity check, it would be relevant to run NGSAdmix/PCA after LD pruning (and/or after removing the identified blocks).

- The large focus on haploblocks makes us forget that there may also be parallel loci outside the blocks: are there? (it seems so from Fig 2). Where is it discussed? Which genes are touched by such parallel loci?

- The 3 clusters are very interesting but hard to catch and interpret. I believe the reader would benefit from a little more information. What could they be? What is known about the biogeography or demographic ancestry of that plant that could help better understand the pattern? Fig 1 is hard to interpret because we don't know which population is which on the admixture plot. Were some reassignments tried for cultivars or introduced species? How was LD controlled? Is this pattern driven by all the genome or a subset of chromosomes?

- Title : the title does not really reflect the findings. First, there is little way to be sure that the blocks of limited recombination are SVs and there is no direct evidence for structural variation. Moreover, there is some evidence for parallelism but also for non-parallelism (which is also interesting meaning that adaptation can find different ways).

- Lack of raw plots: I appreciate keeping the paper simple and extended figures are indeed very nice. That being said, for someone really interested in the paper it would be worth sharing more intermediate outputs. For example, it would be worth showing the MDS and PCA which allowed determining the haploblocks. From extended Fig 6 the three clusters are not always clear. FST pairwise matrix may be of interest, as well as BF values from baypass.

- Minor concerns

- Scripts used in ANGSD/downstream analysis? I could find the scripts used to pre-process the sequence but I couldn't find the scripts used for the analyses (ANGSD, Baypass, etc)

- Data availability: I am slightly surprised that the data are shared as bam files. What if the reference genome changes and people want to re-use the data by re-mapping on a different reference (e.g; a pangenome) or with a different mapper? This kind of data is usually shared as fastq.

- Effect of sample size on pairwise FST: In previous work, my collaborators and I have identified that unbalanced sample sizes were affecting the Fst estimated in ANGSD on low-coverage data (when doing pairwise FST). Notably, populations with a low N were appearing as slightly more differentiated. Have the authors observed the same? How did they control for sample size? (we used to subsample all pop to a similar sample size)

- Strong effect of sample size on diversity estimates: In supplementary figures, the authors show a strong effect of sample size on diversity estimates (thetas statistics). It was not fully clear to me whether, following this observation, populations with low sample size (N=5 for example) were pooled or excluded.

- The dataset joins two datasets obtained and sequenced in different batches. How was the batch effect assessed? (this risk is particularly strong with low coverage data). See Lou & Therkildsen.

Lou, R. N., & Therkildsen, N. O. (2022). Batch effects in population genomic studies with low-coverage whole genome sequencing data: Causes, detection and mitigation. *Molecular Ecology Resources*, 22(5), 1678-1692.

- Somewhere it is mentioned that the reference chromosome names are Chr1_Occ and Chr1_Pal, which were renamed as Chr1 and Chr2. Why so? Is the original reference the result of a recent duplication? If so, how were putatively paralog SNPs filtered? See warning in low coverage data with Dallaire & al

Dallaire, X., Bouchard, R., Hénault, P., Ulmo-Diaz, G., Normandeau, E., Mérot, C., ... & Moore, J. S. (2023). Widespread deviant patterns of heterozygosity in whole-genome sequencing due to autopolyploidy, repeated elements, and duplication. *Genome Biology and Evolution*, 15(12), evad229.

- L455: I don't understand this statement "does not exclude the possibility that haploblocks also include variation in protein-coding regions". Do you mean that different isoforms could be expressed? Wouldn't it be possible to assess allelic variation in RNA and not only quantitative variation of expression?

- L530: How were SNPs polarized for SFS and FST (doMajorminor?)

- L600-601: It can be very misleading for the reader to state that local PCAs are "a precise method to identify SVs". This is

certainly not true. It can be a good clue that a region includes something which maintains non-recombining blocks. We could call them putative inversion, and sometimes confirm them (as in the paper quoted) but usually not all of them, and most confirmations show that putative breakpoints identified from recombination suppression are very approximative. Moreover, the method is strongly biased towards detecting rearrangements which are old/divergent enough, frequent enough (to have a group on PCA), and large enough (to have enough windows of SNPs). I don't mean here that the results are wrong but that precaution should be taken in interpreting and in explaining the method.

- L579-592: how many replicates were run for Baypass? It is usually recommended to run 3 to 5 times and take the median value of XtX or BF (for env associations). See point 6 in the manual.

https://gensoft.pasteur.fr/docs/baypass/2.41/BayPass_manual.pdf I was also wondering why using a WZA associating XtX and BF? Because different environmental variables may provide different information regarding parallelism, what could be learned about looking at each of them? Perhaps raw BF values may be worth sharing in supplementary.

- L648: this is the first mention of GLUE dataset – perhaps explicit it earlier?

- Fig 1A: without interpretation of the 3 clusters, I wonder if the pie chart are informative. I'd really prefer to see sample sizes on a map, and a colour code for populations/continents or name per pop to figure out which pop is which on the admixture plot and the PCA.

- Fig 1C: the PCA is useless without more details to relate to populations. Is one point a population? How did you get read of individual variation? Which point is which?

- Fig 2: "signature of SVs are enriched for patterns of parallel selection": how was it tested? I apologize if I have missed it but I couldn't find the test in the methods and results.

- L637 -642: The PCA were apparently used to genotype samples from the experiment (and also from the wild sampling). How was this performed? People usually use clustering tools, kernels, etc? looking at extended figure 6, PC1 seemed quite continuous for at least a few blocks so I'm wondering where the threshold was put.

- Extended Fig 6 A: The higher heterozygosity in the middle group is unclear. Perhaps consider changing the scale or the shape of the figure. Was the difference in heterozygosity tested?

- Extended Fig 3B is difficult to understand.

Reviewer #2 (Remarks to the Author):

In this study, the authors generated low coverage whole genome sequencing data of 2,660 white clover (*Trifolium repens*) individuals from native and introduced populations across six continents, as well as cultivated lines, and used population genomic approaches to reconstruct the invasion histories of the species, detect selection signals, and identify structural variants. They found high genetic diversity and recent population expansion in both introduced and native populations and revealed distinct colonization histories on different continents despite low genetic differentiation. By examining allele frequency differentiation and genotype-environment association with BayPass, they also discovered shared patterns of differentiation and signatures of climate adaptation between native-introduced comparisons, suggesting parallel selection pressures across introduced regions. Using local population structure method, they identified five large structural variants on three chromosomes that

overlapped with outlier windows for genetic differentiation and revealed that these structural variants were strongly associated with climate variables in all or some ranges. Transcontinental field experiment showed that three of these structural variants significantly influence fitness and align with patterns of local adaptation in both native and introduced ranges, and controlled growth chamber experiments further revealed climate-dependent gene expression patterns in candidate genes, offering insight into their adaptive roles.

Theoretical studies have shown that structural variants, such as inversions, are one of the most influential genetic architectures facilitating parallel genetic evolution. The study in this manuscript demonstrates that structural variants with large effects enable rapid and parallel adaptation helping invasive species thrive in diverse climates.

I find most of the methods and analyses to be sound, and the study presents a comprehensive empirical case of structural variation in species adaptation and parallel genomic evolution, which was a pleasure to read. I only have several comments and suggestions, which I detail below.

- I am not familiar with the *Trifolium* system, so I have several questions regarding the samples:

1. Are the introduced populations in North America, South Africa, Australia, Japan and China originated from the native range or the domesticated cultivars? How did you distinguished clovers that were introduced from native range and cultivars distributed across the world?

2. It is stated that *Trifolium repens* is native to Eurasia ("Here we consider all of Eurasia as the native range" in Supplemental Note). However, in this study, the populations in China and Japan were treated as "introduced". Are there native populations in China and Japan, and how do you distinguished native and introduced populations? If not, I suppose the native range of

the species is actually only restricted to Europe?

3. Please mark which populations are “native” and “cultivar” on the map in Fig. 1A.

- I think it is important to state in the main text that white clover is an allopolyploid, as this is relevant to almost all downstream analyses, including SNP calling, the estimation of genetic diversity and the evaluated gene expression. How divergent are the two sub-genomes? I wonder, given the low sequencing depth (~1X), whether the Illumina reads can be correctly mapped to each sub-genome and whether the variations between paralogs can be reliably distinguished. Could the genetic structure be dominated by the paralogs from the two sub-genomes, potentially explaining the lack of population structure across continents? Did the authors verify the SNPs in these tetraploid samples?

- The authors used the local PCA method described by Li and Ralph, combined with PCA clustering, heterozygosity, and LD patterns, to de novo identify putative structural variants. Local population structure along each chromosome was projected onto five MDS axes to identify outliers. Did the authors perform any filtering (e.g., by size) on the outliers, or the five haploblocks were the only MDS outliers identified in the analysis? Were any MDS outliers filtered out due to not displaying appropriate PCA, heterozygosity, or LD patterns?

Among five haploblocks (putative structural variants), HB7a1 did not exhibit a clear pattern of higher LD, and the PCA clusters and elevated heterozygosity are marginal; the PCA for HB13 appears continuous, while the elevated LD is restricted to only part of the region (Extended Data Fig. 6). How did the author identify the clusters in PCA, and what criteria were used to designate a MDS outlier region as a putative structural variant?

As inversion and translocation have distinct effects on recombination and genetic differentiation (see Bock et al. (2023) <https://doi.org/10.1016/j.xplc.2023.100599>). The authors refer to the haploblocks broadly as structural variants (i.e., inversions and translocations) rather than specifically identifying them as inversions. Is there a particular reason why these cannot be conclusively confirmed as inversions?

- The local population structure method for identifying inversions relies on the principle that inversions suppress recombination within the inverted regions. This suppression often results in large, non-recombining haplotype blocks (haploblocks) in GWA and GEA analyses (see Todesco et al. (2020)). However, the GWAS from the common garden experiments in this study identified sporadic loci strongly associated with fitness measures within each haploblock, using a Bonferroni corrected significance threshold specific to each haploblock. Given the lack of recombination within haploblocks, I wonder whether the outliers identified by this method are stochastic signals. How does the GWA pattern look like across the whole genome? Why not use a genome-wide significance threshold?

Minor comments/edits:

Line 277: “North America, France and New Zealand” should be “North America, Australia, New Zealand, Spain and France”? - based on Extended Data Fig. 3.

Figure 1A: The pie chart for Cape Town, South Africa does not align to the black circle. Although there is only one population from South Africa, is it possible to plot it on a map as in other continents?

Figure 1C: The legend could be placed outside the plot, as it is not easily distinguishable from the data points within the plot. Additionally, population names should consistently use either city or country names. I suggest using country names, as cities like Brighton, Reading, and Antwerp may not be immediately recognizable to all readers.

Figure 5: The genomic coordinates (chromosome + positions) should be marked on the x-axis for each haploblock. Along y-axis, I suggest labeling the GWAS analyses (“North America” to “Montpellier”) and gene expression comparisons differently to improve readability. In addition, Mississauga and Lafayette gardens were merged in group “North America”, while the result of group “Europe” (Extended Data Fig. 9) that summarized Uppsala and Montpellier gardens were not included here. Is there any reason to exclude it?

Extended Data Fig. 3B: Please clarify that the pie charts above map inserts represent the cultivars, and that each cultivar can be originated from an area (dotted lines) or a population (a pie chart). For a number of cultivars, the arrows are not pointed to a clear location (e.g., LA-S1), please correct.

Reviewer #3 (Remarks to the Author):

In this manuscript, Battlay and colleagues report on a comprehensive investigation of evolutionary and genetic drivers of invasion success in a cosmopolitan plant. The authors rely on a large dataset (e.g. lines 97-97: “the largest whole-genome sequence dataset for an invasive plant to date”), as well as an extremely thorough set of analyses that include genome scans for selection, population-genomic inferences of structural variants, genotype-environment and genotype-phenotype association analyses, replicate common gardens at low and high latitudes for both the native and the invasive ranges, measurements of fitness under realistic natural settings, and gene expression analyses following the manipulation of water availability under controlled settings.

Results provided evidence for the contribution of five large structural variants to parallel climate adaptation in this invasive species. This is an important finding, given that structural variants can reconcile two apparently contradicting results on the genetics of biological invasions: a small number of large-effect loci are often inferred to drive invasions, and at the same time most traits that drive invasion success are expected to be highly polygenic. Inversions and other regions of reduced

recombination that link together many co-adapted small effect alleles can underlie both patterns, as the authors mention (lines 132-138).

The manuscript has the potential to become a key reference in invasion genetics, because it addresses a fundamental question on the success of invasive species, it relies on extremely thorough data and analyses, and it has a very high standard of writing and overall presentation.

Major comments:

1. My main concern is that the manuscript places too much emphasis on the five structural variants relative to variants elsewhere in the genome. For example, on lines 468-471, the authors mention that "while there were clearly many regions across the genome with evidence of selection, the strongest and most parallel signatures of adaptation come from just a few haploblocks that also exhibit classic genomic signatures of structural rearrangements". However, unless I am misinterpreting these results, Fig. 2 and Extended Data Fig. 7 indicate that most parallel and non-parallel outlier regions occur in non-haploblock genomic regions. For example, in contrasts between Europe and the other ranges (Extended Data Fig. 7A), only two of the five comparisons showed a significant enrichment of outlier windows in haploblocks (lines 353-356). Thus, across extensive parts of the invasive range of this species (Oceania, China, Japan), haploblocks do not seem to show disproportionately stronger evidence of selection. Along the same lines, I think Fig. 2 would benefit from an additional panel that focuses only on parallel outlier windows and shows (perhaps as a piechart) the breakdown of haploblock and non-haploblock repeated outlier windows. If this interpretation is correct, the manuscript would need to be revised to make it more balanced, and to reflect the large relevance of the rest of the genome as well.

2. Are the five genomic regions structural variants, and do the population genomic signals agree with this interpretation? For example, in Extended Data Fig. 6, I would have expected elevated heterozygosity in the intermediate PC1 group of samples. The authors mention that indeed this was the case (lines 338-340), but no information on statistical significance is provided. Just by looking at the plots, it doesn't seem that this is the case, at least for some of these five genomic regions. Local patterns of LD are also not aligned closely with the inferred boundaries of these structural variants (Extended Data Fig. 6C).

*****END*****

Version 1:

Decision Letter:

4th April 2025

Dear Nic,

Thank you for submitting your revised manuscript "Haploblocks contribute to parallel climate adaptation following global invasion of a cosmopolitan plant" (NATECOLEVOL-24113345A). It has now been seen again by the original reviewers and their comments are below. The reviewers find that the paper has improved in revision, and therefore we'll be happy in principle to publish it in Nature Ecology & Evolution, pending minor revisions to satisfy the reviewers' final requests and to comply with our editorial and formatting guidelines.

If you have not done so already, please ensure that you also email us completed copies of the Reporting summary and Editorial policy checklists:

Reporting summary: https://www.nature.com/documents/nr-reporting-summary.pdf

Editorial policy checklist: https://www.nature.com/documents/nr-editorial-policy-checklist.pdf

[redacted]

Reviewer #1 (Remarks to the Author):

I am happy to see the revisions of this manuscript. The revisions and complementary analysis make the results stronger, and the discussion is much improved. I think that the additions give the manuscript a broader interest.

A minor note for accessibility to the abstract- L103. I'm not sure haploblock has a clear enough definition to be used in an abstract without further explanation - "large blocks of non-recombining haplotypes"? and L107 "putative SV?" or "putative inversions?" "or putative non-recombining SV"? (One has to keep in mind here (like at the end of the intro) that the local PCA method does target only one specific type of signature (reduced recombination leading to long linked divergent haplotypes).

All of my previous comments have been addressed, and I reckon they were numerous. The authors have done an impressive effort at polishing and checking all analyses. This is a very rich study, including so many analysis, data, and results. I believe it will be a landmark in the study of biological invasions, and repeated adaptation through putative SV.

Best regards,
Claire Mérot

Reviewer #2 (Remarks to the Author):

This is the revised version of a manuscript I previously reviewed. Overall, I think the authors have done a great job with the revisions and addressed most of the points I raised earlier, as well as those from the other two reviewers, many of which overlapped, especially regarding the assignment of cultivars and introduced samples, the quality of the SNPs on a duplicated reference genome, and the local population structure patterns used to identify haploblocks.

I only have a couple comments:

- Despite the detailed description of the "invasion"/introduced history of *Trifolium repens*, it is still unclear whether the introduced populations in North America, South America, South Africa, Australia, Japan and China originated from "domesticated" white clover or wild strains. It is to be regretted that with such a wide span of sampling of native, introduced and cultivar samples on all continents, the evolutionary history (domestication, introgression, etc.) of white clover remains unsolved. If the domesticated/historical cultivars/landraces and true wild populations are indistinguishable, please clarify this in the main text to avoid confusion.

- Related to the above problem, I am unsure whether *Trifolium repens* should be treated as an invasive species, as stressed in the first several paragraphs of the main text. Whether the accessions found in other continents are accidental introductions from the wild with costly impacts on ecosystems, agriculture, and human health (similar to *Ambrosia*)? Or are they beneficial domesticated strains intentionally distributed and planted as forage and cover crop? Not that this challenges the study's significance in evolutionary biology, but the focus of the manuscript (including the title, key words, etc.) may need some adjustment.

- Line 539-540: "native across Eurasia" -> "native to Europe and western Asia"; Line 547: "in Eurasia" -> "in Europe and western Asia".

Reviewer #3 (Remarks to the Author):

The authors have thoroughly addressed most of my comments. However, the contribution of non-haploblock variants could be highlighted more strongly. For instance, the authors could note that, while haploblocks show significant enrichment for outlier SNPs, the majority of inferred outlier loci actually lie outside these regions. As a result, statements like the one in the abstract's conclusion—"Our results provide strong evidence that large-effect structural variants underlie rapid and parallel adaptation of an introduced species throughout the world"—do not fully align with the findings. A more precise statement might be: 'Our results provide strong evidence that large-effect structural variants contribute significantly, though not predominantly, to rapid and parallel adaptation of an introduced species throughout the world.'

That said, I appreciate why the authors opted to center the manuscript on haploblocks, as this focus delivers a tighter, more compelling narrative. I have no additional comments. Congratulations to all authors on a valuable contribution.

Dear Editors and Reviewers,

We appreciate the time and effort that has gone into reviewing our manuscript NATECOLEVOL-24113345 "**Haploblocks contribute to parallel climate adaptation following global invasion of a cosmopolitan plant**". We have addressed all of the editor's and reviewers' suggestions and believe that our manuscript is clearer and substantially strengthened as a result of these revisions. We provide a point-by-point response below, but summarize the main changes here:

- 1). We modified our conclusion section to add more discussion regarding the broader significance of our findings. We view our results as both novel and timely for the invasion biology community as well as the broader community of evolutionary biologists interested in the genetic basis of rapid adaptation.
- 2). We added several analyses to validate our previous findings. We reran population structure analyses both by pruning the dataset to account for LD and removing haploblocks to ensure our genome-wide results are not confounded by haploblocks. We added a more detailed characterization of the haploblock regions, including an additional analysis to detect haploblocks in a higher coverage dataset. We have added an additional analysis to partition relative variation in fitness metrics between haploblocks and genomic relatedness matrices in our introduced common gardens. These analyses suggest that our results are robust and further emphasize the importance of the haploblocks.
- 3). We clarified many different points, not just to reviewers, but in the main text and supplemental note. These were often important details that are essential to contextualizing the results and understanding our assumptions. Critically, we justified our focus on haploblock regions. There certainly are additional regions of the genome that are under selection, including many under parallel selection in more than one invasive region. But the haploblocks are strongly differentiated across all introduced regions and exhibit much more parallel signatures than expected by chance. They are the key regions that any objective researcher would approach first because the results are the clearest and expected to have the largest effects on fitness, which we experimentally confirm.
- 4). Finally, we used the constructive suggestions from reviewers to improve our figures and our figure legends. We believe these better highlight the key results and the narrative of the paper.

In short, we hope you find our revisions thorough and our manuscript deserving of acceptance into Nature Ecology and Evolution.

Sincerely,
The Authors

Reviewers' comments:

Reviewer #1 (Remarks to the Author):

This manuscript thoroughly analyses the replicated adaptation of an introduced plant (white clover) in very different geographic areas ...

The manuscript is generally well-written with clear figures. Methods are very detailed (when reading both the manuscript and the supplementary notes), allowing reproducibility for most steps. The analyses are sound and follow state-of-the-art guidelines (but see a few sanity checks below). The main conclusions are mostly supported by the results although I have a few minor concerns. I mostly regretted the absence of a true discussion, which fails to enlighten us about the true impact of the manuscript. I am providing below a few suggestions to ensure the robustness of the results and replace its findings in a relevant context.

Thank you for inviting me to review this paper, which I enjoyed reading a lot.

Claire Mérot

Response: We appreciate the reviewer's time and detail that they put into reviewing our paper. Their comments were overwhelmingly fair and pertinent and we address all of their comments below.

• Major concerns:

Reviewer 1 comment 1: Absence of discussion: The downside of mixing results and discussion is usually that all discussed points are specific to the study system and do not adequately put the findings in perspective with the rest of the literature. Unfortunately, this is the case here. I think either adding a real discussion paragraph or enriching the conclusion would be very valuable to put in evidence as to why this study is impactful. For example, I was very surprised, given the conclusion that large haploblocks may underlie parallel adaptation to see nothing about the theoretical predictions (Here I think about Westram et al 2022 but there is much more literature on the topic, particularly on adaptation in the face of gene flow, etc). Similarly, the study focuses on white clover and does not mention comparable empirical studies. I can think of many examples of inversion clines in *Drosophila*, parallel adaptation in stickleback, parallel karyotype-environment associations in *Coelopa*, *Littorina* crab/wave examples, and even a study on *Ambrosia* plants by the same author(s). And this is only for the re-use of haploblocks or inversions in parallel adaptation. There is much more literature on parallel adaptation itself (and on introduced/invasive species)

Westram, A. M., Faria, R., Johannesson, K., Butlin, R., & Barton, N. (2022). Inversions and parallel evolution. *Philosophical Transactions of the Royal Society B*, 377(1856), 20210203.

Response: We have now revised our manuscript to include a discussion section (lines 479-526) and short conclusion (lines 528-535) rather than a relatively long conclusion section. This revision allowed us to better emphasize our results in the content of the literature on parallel adaptation of invasions and the genetic basis of rapid adaptation. We appreciate the literature that the reviewer has provided - much of it is now used within our revised discussion. At the

same time, we are conscious of word limits and were judicious in our use of space. We have also added more discussion to a number of the sections (see comments below). We believe this aids in the interpretation of our results.

Reviewer 1 comment 2: The LD bias is not really accounted for: this study put in evidence the presence of haploblocks which gather many outliers in statistical analysis for adaptation. The problem with such haploblocks is the strong LD (due to limited recombination in heterokaryote if this is truly an inversion). So it is not surprising to find a high density of candidate, because as soon as one locus is picked, the other linked loci bear the same signal (due to LD!). This makes it really hard to interpret the enrichment in GWAS (ext Fig 10) and GEA (ext Fig 7) for haploblocks. At the same time, not all SNPs inside a block have a signal so... how should we interpret the density of putative candidates? I have no answer to this and no solution to sort true candidates and hitch-hicked loci inside haploblocks BUT it would be worth at least discussing it (and if the authors have a solution, even better!).

Response: This is an excellent comment and we agree that it may not be surprising to find a high density of fitness-associated SNPs within the haploblock region. It is surprising how much polymorphism is contained within the different haploblock alleles. The presence of this variation suggests that the structural variants are old and have had time to accumulate variation and experience recombination within each of the haploblock alleles. This variation gives us some power to investigate how allelic variation within haploblocks impacts fitness. Indeed, we see moderately sharp peaks within the haploblocks associated with fitness, which suggests that particular variants within haploblocks are indeed adaptive. However, we are careful to avoid stating that these fitness-associated SNPs in the haploblocks are the only variants with fitness effects. For instance, particular variants within haploblocks may be underrepresented within our panel of accessions represented in the common gardens, and thus may not show strong associations. As indicated by the reviewer, a specific analysis to sort the true candidates is not possible, but we did improve the Discussion of these limitations within the main text (lines 418-421).

We also added one link to the paragraph that initially characterizes the haploblocks (lines 354-356): *“While there is elevated LD within haploblocks, there are still high levels of polymorphism with each block suggesting that haploblocks are old.”*

We also added a sentence to the paragraph identifying fitness associated loci within haploblock regions (lines 418-421): *“This method is likely limited for specific genes underlying fitness differences as there is elevated linkage disequilibrium within haploblocks; however, there is substantial variation within haploblocks, which allowed us to identify distinct peaks of phenotype:genotype association.”*

We also added the following caveat (lines 423-425): *“..and the analysis is likely only detecting a subset of fitness-associated genes due to limited sample size of some haplotype genotypes.”*

Reviewer 1 comment 3: A related matter about LD: the strong divergence and LD in haploblocks can also drive signals in NGSAdmix and PCA, even on 4-fold degenerate SNPs because SNPs in LD will drive the signal. As a sanity check, it would be relevant to run NGSAdmix/PCA after LD pruning (and/or after removing the identified blocks).

Response: We agree with this suggestion. We constructed two modified datasets to ensure that our genome-wide 4-fold degenerate site analyses were not influenced by including sites with significant linkage-disequilibrium or by including haploblocks in the dataset. The first new dataset consisted of all 4-fold degenerate sites minus the haploblock regions. The second dataset consisted of distance-pruned 4-fold degenerate sites and still included the haploblocks. We pruned the dataset to keep one SNP every 10kb, as LD decays rapidly ($r^2 < 0.05$ within 5-10kb; Inostroza et al. 2018, Kuo et al. 2024) in this highly-diverse obligate-outcrossing species. We trust this measure more than strict LD pruning because calculating LD with low-coverage data across many populations seems more problematic. Neither pruned dataset has any substantial impact on the population structure within PCAs (i.e. the relative distances between populations and nesting patterns of native within invasive populations is the same). We added a new figure to our supplemental note (Supplemental Note Fig. 2) depicting PC space for each of these datasets and with points representing individual-level variation vs. population means.

We also added this sentence to the main text of the methods (lines 610-611): “*Additionally, distance-based pruning of our dataset and removing haploblocks do not alter population structure (Supplemental Note Fig. 2).*” These similar results despite the presence/absence of pruning haploblocks and linked SNPs indicate that our results are robust.

Reviewer 1 comment 4: The large focus on haploblocks makes us forget that there may also be parallel loci outside the blocks: are there? (it seems so from Fig 2). Where is it discussed? Which genes are touched by such parallel loci?

Response: Yes, there are other loci underlying climate adaptation and parallel signatures of adaptation in other parts of the genome. We choose to focus on the haploblocks because they are strongly overrepresented both in terms of the windows under selection in our XtX-EAA and contrast analyses (which is even stronger when considering outlier windows that show parallel signatures across multiple invaded regions), and they are likely to have demonstrative effects on plant fitness and phenotypes given the number of genes in LD within the haploblocks. This expectation is strengthened by the field experiments.

Given the reviewers comments we have made an effort to highlight some more general features of our selection scans that are not specific to haploblocks. Specifically, we discuss the significant correspondence between population-level differentiation (XtX) and climate correlations in all ranges:

“In each range, between 15 and 52% of XtX outlier windows were also outliers for correlations with at least one of six minimally-correlated climate variables (XtX-EAA windows). In all ranges

this overlap was far more than would be expected by chance (hypergeometric test p-values < 7.01e-31)”

This indicates that adaptation to local climate is an important driver of differentiation in the species' native and invaded ranges. We also highlight the parallelism observed between the selection scans in distinct ranges:

“Across the six ranges, we observed signatures of genetic parallelism in climate adaptation - the outlier XtX-EAA climate adaptation windows overlapped between ranges more often than expected by chance for all between-range comparisons (hypergeometric test p-values < 0.013).”

To further respond to this comment we conducted a GO enrichment analysis for parallel windows outside haploblock regions, but we did not recover any notable patterns and have opted to leave it out of the manuscript. If the editor feels that the inclusion of these results are important we could add it to the supplementary information.

Reviewer 1 comment 5: The 3 clusters are very interesting but hard to catch and interpret. I believe the reader would benefit from a little more information. What could they be? What is known about the biogeography or demographic ancestry of that plant that could help better understand the pattern? Fig 1 is hard to interpret because we don't know which population is which on the admixture plot. Were some reassignments tried for cultivars or introduced species? How was LD controlled? Is this pattern driven by all the genome or a subset of chromosomes?

Response:

We use the NGSadmix in combination with our PCA and population genomic summary statistics to make our interpretations about population structure and colonization history. We acknowledge that our clustering reflects the reality of a species with high levels of genetic diversity within populations and relatively limited diversity between populations on a worldwide scale. But these clusters obviously provide some interesting conclusions on patterns of variation across different invasive regions. We have now better highlighted the geographic patterns associated with these clusters.

To provide biogeographical context of the NGSadmix results we include the following text (lines 236-238): *“The extreme east and west native populations (i.e., Tehran and Spain) primarily represented distinct ancestries (purple and green, respectively), and a third ancestry increased with latitude (orange).”*

All of the populations are on the admixture plot (i.e. all 2661 individuals are represented), which we state on lines 1026-27.

Cultivars largely looked like the Spanish populations in the NGSadmixture analysis, just as in the PCA. We did not reassign individuals from wild collected populations as cultivars (or vice-versa) because those collections have not undergone artificial breeding or domestication to the best of our knowledge. We do acknowledge that the evolutionary history of wild white clover populations is intertwined with hybridization and introgression from

We did not control for LD in the NSGadmixture analysis because we only used 4-fold degenerate sites, and moreover distance-pruning of 4-fold degenerate sites or removing haploblock regions did not change the clusters or interpretation (also see our response to Reviewer 1, comment #3). Thus, our patterns are driven by genome-wide patterns of population structure rather than by specific haploblocks or chromosomes.

Reviewer 1 comment 6: Title : the title does not really reflect the findings. First, there is little way to be sure that the blocks of limited recombination are SVs and there is no direct evidence for structural variation. Moreover, there is some evidence for parallelism but also for non-parallelism (which is also interesting [finding] meaning that adaptation can find different ways).

Response: We have had substantial internal discussions regarding the title. We made a slight revision to the title by revising “underlie” to “contribute to”. Our logic for the title is that it highlights the most prominent conclusion from our study - that haploblocks seem to be under strong selection in the introduced ranges and these patterns are more parallel than expected either from genomic regions not under selection or regions under selection. We changed ‘Structural Variants’ to ‘Haploblocks’ in response to the reviewer’s comment.

Reviewer 1 comment 7: Lack of raw plots: I appreciate keeping the paper simple and extended figures are indeed very nice. That being said, for someone really interested in the paper it would be worth sharing more intermediate outputs. For example, it would be worth showing the MDS and PCA which allowed determining the haploblocks. From extended Fig 6 the three clusters are not always clear. FST pairwise matrix may be of interest, as well as BF values from baypass.

Response: We are pleased that the reviewer found our methods interesting and have endeavored to represent them as much as possible in the figures, but we are limited to ten extended data figures. The MDS plots used to identify the haploblocks are in Fig. 2C. The relevant axis from the local PCAs used to genotype the haploblocks are included in extended data Fig. 6A, and we have improved the readability of these plots by cropping out outliers. We have now also included a second version of these plots based on SNPs called in a single population with high-coverage data (extended data Fig. 6B). We have also added WZA scores for XtX, EAA (τ) and contrast statistics for each 20 kbp window in the genome in Supplementary Table 2. The results and figures can also be replicated by accessing https://github.com/pbattlay/glue-invasions/angsd_baypass_EAA and https://github.com/pbattlay/glue-invasions/angsd_baypass_invasion_contrast

• Minor concerns

Reviewer 1 comment 8: Scripts used in ANGSD/downstream analysis? I could find the scripts used to pre-process the sequence but I couldn't find the scripts used for the analyses (ANGSD, Baypass, etc)

Response: We appreciate the reviewer's effort to help us make our work transparent and reproducible. The scripts used for ANGSD and BayPass are stored in larger text files that detail each step of the analysis from BAM files to WZA scores or BAM files to haploblock genotypes. They were previously poorly named and we apologize for the lack of clarity! The names of these files have been updated to the following:

https://github.com/pbattlay/glue-invasions/angsd_baypass_EAA
https://github.com/pbattlay/glue-invasions/angsd_baypass_invasion_contrast
https://github.com/pbattlay/glue-invasions/angsd_local_PCA_haploblock_analysis

We have also added additional code to support downstream population genomic and common garden analyses. All of our results can be recreated from the scripts in the github.

Reviewer 1 comment 9: Data availability: I am slightly surprised that the data are shared as bam files. What if the reference genome changes and people want to re-use the data by re-mapping on a different reference (e.g; a pangenome) or with a different mapper? This kind of data is usually shared as fastq.

Response: Thank you. We now include all low coverage genomes as fastq files. We believe this was a misunderstanding in our initial draft, as our NCBI bioproject we cited does have all of the fastq files, but our previous wording was confusing. The previous paper with GLUE dataset had only .bam files.

Reviewer 1 comment 10: Effect of sample size on pairwise FST: In previous work, my collaborators and I have identified that unbalanced sample sizes were affecting the Fst estimated in ANGSD on low-coverage data (when doing pairwise FST). Notably, populations with a low N were appearing as slightly more differentiated. Have the authors observed the same? How did they control for sample size? (we used to subsample all pop to a similar sample size)

Response: We had not previously considered this possibility. We examined the relationship between combined sample sizes of both populations and pairwise Fst using both linear models and linear mixed models (to account for population-associated pseudoreplication). Our results suggest there is an association here ($r^2 = 0.11$), but in the opposite direction that you have observed. Populations with low combined sample sizes are less differentiated from one another (low pairwise Fsts). This correlation disappears when combined sample size is $> \sim 30$ (see below graph). When we remove populations with > 10 individuals sampled, our mantel tests results are similar and consistent with our previous interpretation. Our outlier analyses are based on empirical p-values, with outliers relative to the overall distribution of each statistic. Thus, we

believe these tests should be robust (or even conservative) in identifying SNPs that are highly differentiated across regions.

Reviewer 1 comment 11: Strong effect of sample size on diversity estimates: In supplementary figures, the authors show a strong effect of sample size on diversity estimates (thetas statistics). It was not fully clear to me whether, following this observation, populations with low sample size (N=5 for example) were pooled or excluded.

Response: Thank you. We have performed additional analyses that are detailed in the supplement note to confirm that our conclusions hold regardless of variation in population sample size or number of SNPs on diversity estimates. We also conducted additional analyses to determine whether our comparisons of native vs. introduced range statistics were impacted by sample size or number of sites. First, we ran univariate general linear models with different population summary statistics as response variables (P_i , Waterson's theta, Tajimas D). Range (native/introduced) was treated as a fixed effect and population sample size and number of SNPs as covariates. Because this linear model includes multiple different invaded regions, we conducted similar linear mixed models that include 'invaded region' as a random effect. Finally, we removed low sample size populations (N<10). All of the above analyses produce nearly identical results to the original results. We now refer to these additional analyses with the addition of the following:

Lines 585-586: *"Models including covariates for population sample size and number of SNPs do not qualitatively change conclusions."*

Supplemental Note: We also now make it clear that *all* populations were included in the comparisons of the diversity estimates between the introduced and native ranges presented in the main text.

Reviewer 1 comment 12: The dataset joins two datasets obtained and sequenced in different batches. How was the batch effect assessed? (this risk is particularly strong with low coverage data). See Lou & Therkildsen.

Lou, R. N., & Therkildsen, N. O. (2022). Batch effects in population genomic studies with low-coverage whole genome sequencing data: Causes, detection and mitigation. *Molecular Ecology Resources*, 22(5), 1678-1692.

Response: We agree that it would be useful to assess whether batch sequencing effects could influence our results. Generally speaking, our dataset should have limited batch effects because: 1) our dataset is large, and 2) is the result of many library preps across many different lanes of sequencing. This should minimize any effect of odd sequencing bias. In our analyses, the most important effect that batch effects could have would be on our population structure analysis. Thus, we decided to evaluate whether batch effects impacted our population component analysis. To evaluate the variation associated with batch effects, we first needed to run an individual-scale analysis rather than the population-scale analysis. We moved to a multivariate linear mixed modeling framework (rather than a PERMANOVA) using the *lmer()* function within *lme4* library for linear models and the *Anova()* function in the *car* library for significance testing. Our initial model had the first four genetic PCs as the multivariate response variable, region (native/introduced/cultivar) as a fixed factor, and population as a random factor in the model. The region factor was statistically significant in this model ($X^2 = 8.01$, $df = 2$, $p = 0.018$) indicating that the individual-scale and population scale models produced qualitatively similar results. We then added a batch term as a random variable; we consider each sequencing lane a different batch. Each individual in our analysis was only sequenced on a single lane. The batch random effect accounted for 1.8% of the variance in PC space. The region factor became slightly more statistically significant ($X^2 = 8.12$, $df = 2$, $p = 0.017$) with the batch term in the model. These results suggest that the batch term has negligible impact on our results.

We now have modified the methods in the main text and provided more extensive methods in the supplemental note to indicate the different sequencing runs had negligible effect on our population structure analyses.

Main text: (lines 608-609): “Differences in number of samples, sequencing coverage, batch effects from sequencing runs have limited impact on our inferences of population structure (see Supplemental Note).”

Reviewer 1 comment 13: Somewhere it is mentioned that the reference chromosome names are Chr1_Occ and Chr1_Pal, which were renamed as Chr1 and Chr2. Why so? Is the original reference the result of a recent duplication? If so, how were putatively paralog SNPs filtered? See warning in low coverage data with Dallaire & al

Dallaire, X., Bouchard, R., Hénault, P., Ulmo-Diaz, G., Normandeau, E., Mérot, C., ... & Moore, J. S. (2023). Widespread deviant patterns of heterozygosity in whole-genome sequencing due to autopolyploidy, repeated elements, and duplication. *Genome Biology and Evolution*, 15(12), evad229.

Response: Yes, we mention this in the supplemental note. White clover is an allotetraploid species that formed ~15,000-28,000 years ago. The parental genomes are highly diverged ($K_s \sim 0.04$ genome wide; Kuo et al. 2024) and our reference genome has both haplotypes and subgenomes resolved at a chromosome level (Santangelo et al. 2023). We changed the name of the chromosomes to the more standard notation (Chromosome 1-16) rather than the naming system in the previous paper for simplicity for the reader. We note that NCBI renames the chromosomes in the same manner as in our manuscript. Since there is quite a bit of divergence between the parental genomes, our mapping statistics are exceptional (i.e., most reads are properly paired and align only once). Our ANGSD filtering removes all reads that are not properly paired and also filters by mapping quality. Interestingly, there has not been extensive rediploidization of the *T. repens* genome - two copies of most genes still remain in the genome (80%+ still duplicated; Griffiths et al. 2019; Santangelo et al. 2023) and the genome remains highly syntenic. This makes our genome somewhat unlike the fish species in the study above as they lack the re-diploidized regions that were the biggest issues.

Reviewer 1 comment 14: L455: I don't understand this statement "does not exclude the possibility that haploblocks also include variation in protein-coding regions". Do you mean that different isoforms could be expressed? Wouldn't it be possible to assess allelic variation in RNA and not only quantitative variation of expression?

Response: Yes, we do mean that different isoforms could be expressed and thus that the adaptive variation in the haploblocks could be produced by directly modifying the coding regions of proteins with no change in expression. Indeed, it is possible to assess allelic variation in the RNA, but we do not have genomic data from these samples to call haploblocks (nor phenotypic/fitness data to correlate to differences within isoforms). We did modify our text to make it clearer for the reader (lines 475-477):

"This result is consistent with cis-regulatory changes underlying rapid adaptation following introduction, but does not exclude the possibility that haploblocks also include ecologically important variation in protein-coding regions."

Reviewer 1 comment 15: L530: How were SNPs polarized for SFS and FST (doMajorMinor?)

Response: Clearly you have spent some time doing these same analyses - good catch. In the main text, we only described the command for calculating genotype likelihoods and 1DSFS for the genetic diversity metrics. The flags for calculating genotype likelihoods for 2DSFS and Fst were slightly different and previously only in the supplementary note. Indeed, we used -doMajorMinor 4 to polarize by the allele found in the reference genome. We have now revised the main text to read as follows (lines 577-579):

“We recalculated genotype likelihoods and SAFs for each population using the reference genome to assign major and minor alleles (-GL 1 -doMaf 2 -minMaf 0.05 -doCounts 1 -dumpCounts 2 -baq 2 -minQ 20 -minMapQ 30 -doSaf 5 -doMajorMinor 4 -sites 4fold.sites) to estimate differentiation using Hudson’s Fst (realSFS fst index -whichFst 1).”

Reviewer 1 comment 16: L600-601: It can be very misleading for the reader to state that local PCAs are “a precise method to identify SVs”. This is certainly not true. It can be a good clue that a region includes something which maintains non-recombining blocks. We could call them putative inversion, and sometimes confirm them (as in the paper quoted) but usually not all of them, and most confirmations show that putative breakpoints identified from recombination suppression are very approximative. Moreover, the method is strongly biased towards detecting rearrangements which are old/divergent enough, frequent enough (to have a group on PCA), and large enough (to have enough windows of SNPs). I don’t mean here that the results are wrong but that precaution should be taken in interpreting and in explaining the method.

Response: Thank you, we agree our wording should be toned down. We have deleted the sentence and now make the point in the paragraph’s opening sentence (lines 648-649) that the method has been successful in the past, including with low-coverage data:

We identified haploblocks – population-genomic signatures of large structural variants – using local principal component analysis, which has proven reliable in a range of genomic datasets (e.g.,^{15,54,55}) including those with low-coverage whole-genome sequencing data⁴⁸.

Reviewer 1 comment 17: L579-592: how many replicates were run for Baypass? It is usually recommended to run 3 to 5 times and take the median value of XtX or BF (for env associations). See point 6 in the manual. https://gensoft.pasteur.fr/docs/baypass/2.41/BayPass_manual.pdf

Response: We only ran a single replicate for Baypass because, according to the manual “For large enough data sets, estimations are generally reproducible for most parameters and statistics” (our dataset is very large), and also it is the BF statistic that is particularly unstable (we did not use BFs). We used a Kendall’s Tau summarized across windows with the WZA to measure EAAs, which has greater power and a lower false discovery rate than Baypass’s BF in most situations (Booker, Yeaman, Whiting & Whitlock (2024) *Molecular Ecology Resources*, 24(2), e13768). Further, our use of the WZA to summarize XtX data over 20 kbp windows will reduce the effects of false-positives at individual loci.

Reviewer 1 comment 18: I was also wondering why using a WZA associating XtX and BF? Because different environmental variables may provide different information regarding parallelism, what could be learned about looking at each of them? Perhaps raw BF values may be worth sharing in supplementary.

Response: We considered only windows that had outlier WZA window scores for XtX and Tau as a conservative test for any kind of local climate adaptation, which was our primary goal in this

analysis. While we agree that it would be interesting to look at patterns in individual variables, given the scale of our dataset we believe it is beyond the scope of this paper. We have, however, included WZA scores (tables for individual SNPs would be enormous) for XtX and Tau for each variable separately as a supplementary data file.

Reviewer 1 comment 19: L648: this is the first mention of GLUE dataset – perhaps explicit it earlier?

Response: We have now removed all mentions of the GLUE dataset from the main text as we do not want readers to get these two datasets confused. Our dataset has significantly expanded sampling from the original GLUE paper. We do cite the original GLUE paper (Santangelo et al. 2022) several times including when describing its large contribution to our dataset, and explicitly define its contribution in the supplemental note.

Reviewer 1 comment 20: Fig 1A: without interpretation of the 3 clusters, I wonder if the pie chart are informative. I'd really prefer to see sample sizes on a map, and a colour code for populations/continents or name per pop to figure out which pop is which on the admixture plot and the PCA.

Response: Thank you, we agree this addition would be useful . We have now added population numbers to Fig1A-1C so that readers can identify populations in 1B and 1C more easily. We used italics in 1A to identify relative sampling size (i.e. populations with more than 32 individuals are normal typeset, while those with fewer than 10 samples are in italics). We acknowledge that the interpretation of the three clusters is nuanced, but we believe that the pie charts provide the reader with important information about the ancestries that make up each introduced population. That is, there is little northern or eastern European ancestry in the introduced regions and most introduced populations look similar to western European populations and modern cultivars. Also, there is differentiation in ancestry across latitude (and climate) in the introduced regions that looks parallel across different invaded regions. In short, we think this is a key part of the figure and we have kept it in the manuscript.

Reviewer 1 comment 21: Fig 1C: the PCA is useless without more details to relate to populations. Is one point a population? How did you get read of individual variation? Which point is which?

Response: Each point within Fig. 1C does represent a population and we clarify this in the legend. We added a number of different versions of the PCA plot to the supplemental note, including those that show all individuals and those that show population means with standard error on PC1 and PC2 axes. We also now include numbers to each of the points so that readers can identify different populations.

Reviewer 1 comment 22: Fig 2: “signature of SVs are enriched for patterns of parallel selection”: how was it tested? I apologize if I have missed it but I couldn’t find the test in the methods and results.

Response: You are right that we left this out of the text. We apologize for the oversight and have now added the following sentence (Line 369-371):

“...29% and 10% of parallel windows (i.e. windows that were outliers in more than one range) for XtX-EAA and contrast scans, respectively, were found within haploblocks, marking a substantial enrichment in these regions relative to the rest of the genome (hypergeometric test $p \leq 9.09e-16$).”

Reviewer 1 comment 23: L637 -642: The PCA were apparently used to genotype samples from the experiment (and also from the wild sampling). How was this performed? People usually use clustering tools, kernels, etc? looking at extended figure 6, PC1 seemed quite continuous for at least a few blocks so I’m wondering where the threshold was put.

Response: Clusters were assigned manually (i.e., by eye). We would ideally remove this human interpretation aspect, however we have attempted to use a range of clustering algorithms for this task (including in other systems with higher coverage data and SVs verified by a diploid assembly) and found them to be unreliable. We are confident that our genotyping of these haploblocks is accurate, as the heterozygosity and LD patterns we observed are based on the genotypes assigned by clustering. We have added an additional check on our cluster assignments by comparing them with PC/heterozygosity plots generated from SNPs called in a single GLUE population with higher-coverage sequencing data (~10X) now displayed as EFig. 6B.

Reviewer 1 comment 24: Extended Fig 6 A: The higher heterozygosity in the middle group is unclear. Perhaps consider changing the scale or the shape of the figure. Was the difference in heterozygosity tested?

Response: We had originally tested this as follows: “We defined elevated heterozygosity as when the standard error of the mean (SEM) for each homozygote class did not overlap the SEM for the heterozygote class.” Based on the reviewers comment we have added a Wilcoxon test to specifically compare each homozygote class with its corresponding heterozygote class—all tests are significant at $p < 0.0003$. This is now stated in the methods (lines 666-668) as well as the caption for EFig. 6. To improve the visualization of clustering and differences in heterozygosity, we have cropped outliers on the y-axes in EFig. 6A, which we also state in the figure legend.

Reviewer 1 comment 25: Extended Fig 3B is difficult to understand.

Response: We have modified our legend text to better represent the figure and also redrawn some of the arrows for better viewing. The figure legend now reads as follows:

“B. A map of cultivar origins and NGSadmixture estimated ancestries. Each pie chart within map inserts reflects the average ancestries ($K=3$) from a given wild population (not from cultivars).

Pie charts outside the map reflect the ancestry for each cultivar. Arrows indicate the relative locations where wild populations were collected to generate each cultivar. Note that some cultivars were generated from crosses between wild populations from multiple areas. Dotted boxes reflect when the location of the originating wild population is a general location. Note that Durana and Renovation stem from plants collected in Georgia, USA. LA-S1 and Patriot originate from plants in Louisiana, USA and Mississippi, USA, respectively. Pitua is the result of crossing between Spanish and New Zealand accessions.”

Reviewer #2 (Remarks to the Author):

In this study, the authors generated low coverage whole genome sequencing data of 2,660 white clover (*Trifolium repens*) individuals from native and introduced populations across six continents, as well as cultivated lines, and used population genomic approaches to reconstruct the invasion histories of the species, detect selection signals, and identify structural variants...

I find most of the methods and analyses to be sound, and the study presents a comprehensive empirical case of structural variation in species adaptation and parallel genomic evolution, which was a pleasure to read. I only have several comments and suggestions, which I detail below.

Response: Thank you, we are glad the reviewer found merit in our work and we thank them for their constructive feedback.

- I am not familiar with the *Trifolium* system, so I have several questions regarding the samples:

Reviewer 2 comment 1: Are the introduced populations in North America, South Africa, Australia, Japan and China originated from the native range or the domesticated cultivars? How did you distinguished clovers that were introduced from native range and cultivars distributed across the world?

Response: We provide a detailed evolutionary and invasive history of white clover in the supplemental note (see section “Study system”). In short, white clover was introduced from European populations after the cultivation of white clover in Spain between 1000-1200 AD and spread of ‘domesticated’ white clover around western Europe. Wild accessions from western Europe were likely highly outbred with local wild strains of white clover as this rotational/pasture crop was brought into use in different areas. Other than that, we can say very little about whether historical cultivars/landraces, or seed from wild populations, were introduced. White clover was likely introduced to the US as ‘English Hay’, harvested from pastures with multiple grass species and red clover. The cultivars we refer to within our manuscript are modern cultivars from advanced breeding designs produced from the 1960’s to the present. Our PCA shows that these cultivars are actually relatively diverged from the wild strains we collected in the field.

Reviewer 2 comment 2: It is stated that *Trifolium repens* is native to Eurasia (“Here we consider all of Eurasia as the native range” in Supplemental Note). However, in this study, the

populations in China and Japan were treated as “introduced”. Are there native populations in China and Japan, and how do you distinguished native and introduced populations? If not, I suppose the native range of the species is actually only restricted to Europe?

Response: There are no native populations in China and Japan as white clover is widely considered to be introduced to these areas (IUCN Global Invasive Species Database). But white clover is thought to be native to many countries in western Asia. We have replaced Eurasia with a more accurate description (“*Europe and western Asia*”) in both the main (line 160) and supplemental note.

Reviewer 2 comment 3: Please mark which populations are “native” and “cultivar” on the map in Fig. 1A.

Response: None of the pie charts in Fig. 1A are cultivars - these are all wild populations. We have now specified this within the legend.

Reviewer 2 comment 4: I think it is important to state in the main text that white clover is an allopolyploid, as this is relevant to almost all downstream analyses, including SNP calling, the estimation of genetic diversity and the evaluated gene expression. How divergent are the two sub-genomes? I wonder, given the low sequencing depth (~1X), whether the Illumina reads can be correctly mapped to each sub-genome and whether the variations between paralogs can be reliably distinguished. Could the genetic structure be dominant by the paralogs from the two sub-genomes, potentially explaining the lack of population structure across continents? Did the authors verify the SNPs in these tetraploid samples?

Response: Please see Comment 13 from Reviewer 1. Notably, there is significant differentiation between the subgenomes ($K_s = \sim 4\%$) and we have a chromosome-level reference genome that is both haplotype- and subgenome resolved that is highly contiguous and complete (e.g., it is close to the predicted genome size). Our metrics for alignment suggest a typical level of reads are properly-paired with low-levels of reads mapping to different chromosomes (i.e. 96.0% of mapped reads were properly-paired). We filter out any non-properly paired reads in ANGSD. In short, our mapping and filtering strategy is not contributing to high levels of heterogeneity across populations. Additionally, our diversity estimates match previous studies of genetic diversity and population structure assessed via homoeologous SSR markers (Kooyers et. al 2012, Kooyers & Olsen 2013, Wu et al. 2021).

Reviewer 2 comment 5: The authors used the local PCA method described by Li and Ralph, combined with PCA clustering, heterozygosity, and LD patterns, to de novo identify putative structural variants. Local population structure along each chromosome was projected onto five MDS axes to identify outliers. Did the authors perform any filtering (e.g., by size) on the outliers, or the five haploblocks were the only MDS outliers identified in the analysis? Were any MDS outliers filtered out due to not displaying appropriate PCA, heterozygosity, or LD patterns? Among five haploblocks (putative structural variants), HB7a1 did not exhibit a clear pattern of higher LD, and the PCA clusters and elevated heterozygosity are marginal; the PCA for HB13

appears continuous, while the elevated LD is restricted to only part of the region (Extended Data Fig. 6). How did the author identify the clusters in PCA, and what criteria were used to designate a MDS outlier region as a putative structural variant?

Response: We added these details to the *Haploblock identification* section of the methods with the following sentences (lines 656-660):

“We selected MDS scan regions for further analysis based on the presence of clusters of a particular corner outlier in a chromosomal region, i.e., stretches of a chromosome where the population structure was both similar and extreme. In total, ten such regions were analyzed, but five were excluded based on lack of clustering in the local PCA and/or patterns of heterozygosity incongruous with a structural variant.”

We have removed a small number of outliers from the heterozygosity plots so now the elevated heterozygosity is more apparent. For each haploblock the nucleotide diversity (heterozygosity) of the heterozygote class is significantly greater than the homozygote classes based on a Wilcoxon test. The HB7a1 LD plot is complicated by the fact that it overlaps with HB7a2, so the reduced LD is only apparent at the beginning of the haploblock where there is no overlap. There are two regions in HB13 where LD is reduced but the first region has stronger LD than the second which makes the latter more difficult to see.

Clusters were assigned manually, as we have found clustering algorithms to be unreliable for this task even in datasets with less noisy data and SVs verified by a diploid assembly. We relied on the heterozygosity and LD patterns as checks on our genotyping. We now include another check on our genotyping, leveraging much clearer PC/heterozygosity plots generated from SNPs called in a single GLUE population with higher sequencing coverage (EFig. 6B).

Reviewer 2 comment 6: As inversion and translocation have distinct effects on recombination and genetic differentiation (see Bock et al. (2023) <https://doi.org/10.1016/j.xplc.2023.100599>). The authors refer to the haploblocks broadly as structural variants (i.e., inversions and translocations) rather than specifically identifying them as inversions. Is there a particular reason why these cannot be conclusively confirmed as inversions?

Response: Our method for haploblock detection relies on population-genomic signatures of recombination repression. While this is usually interpreted as evidence of an inversion, similar patterns could result from a translocation polymorphism – Bock et al. (2023) suggests that translocations *can* have effects on recombination suppression that are distinct from inversions, but they can also have effects that are similar to inversions (i.e., recombination suppression across the length of the SV; recombination suppression at the SV breakpoint). We therefore believe it is reasonable to suggest the possibility that the SVs could be inversions or translocations.

Reviewer 2 comment 7: The local population structure method for identifying inversions relies on the principle that inversions suppress recombination within the inverted regions. This

suppression often results in large, non-recombining haplotype blocks (haploblocks) in GWA and GEA analyses (see Todesco et al. (2020)). However, the GWAS from the common garden experiments in this study identified sporadic loci strongly associated with fitness measures within each haploblock, using a Bonferroni corrected significance threshold specific to each haploblock. Given the lack of recombination within haploblocks, I wonder whether the outliers identified by this method are stochastic signals. How does the GWA pattern look like across the whole genome? Why not use a genome-wide significance threshold?

Response: This is similar to Comment 2 raised by Reviewer 1 regarding the utility of an association study within our haploblocks. We note that while LD is elevated within haploblocks, it is not extremely tight linkage across the entire block (See Extended Data Fig. 6). Additionally, these haploblocks have substantial polymorphism within them, as would be expected if haploblock were relatively old and had an opportunity to recombine within a given haploblock allele when they are homozygous within an individual. Thus, we do not find it odd that we find solid peaks within some of our haploblock regions and think it is highly unlikely the signals we detect are “stochastic”. We believe the best significance threshold is using the Bonferroni corrected significance threshold based on the number of SNPs within the haploblock regions. This is a conservative method because there is linkage between markers that makes them non-independent observations. But we also acknowledge that the full genome significance threshold would be higher ($\alpha = 1e-6$ to $1e-8$ depending on garden and fitness metric). In our opinion, using these thresholds would be overly conservative and create false negatives. We note that many (but not all) of our peaks would still be significant with the more stringent threshold. We do not evaluate or report any of the regions outside the haploblocks here as our intention is just to characterize the haploblock regions that show strong selective signals and parallel signals relative to the rest of the genome. Our analysis would look slightly different if we intended on presenting a GWAS for the whole genome (likely higher filtering of SNPs to reduce false negatives for a first pass).

Minor comments/edits:

Reviewer 2 comment 8: Line 277: “North America, France and New Zealand” should be “North America, Australia, New Zealand, Spain and France”? - based on Extended Data Fig. 3.

Response: Thank you for catching this - this is correct. Revised.

Reviewer 2 comment 9: Figure 1A: The pie chart for Cape Town, South Africa does not align to the black circle. Although there is only one population from South Africa, is it possible to plot it on a map as in other continents?

Response: Revised. We added a map of southern Africa with a Cape Town pie chart.

Reviewer 2 comment 10: Figure 1C: The legend could be placed outside the plot, as it is not easily distinguishable from the data points within the plot. Additionally, population names should

consistently use either city or country names. I suggest using country names, as cities like Brighton, Reading, and Antwerp may not be immediately recognizable to all readers.

Response: We have added a box around our legend to distinguish it from the points within the plot. Because we had very few samples from multiple different Spanish cities (and there was no population structure between Spanish cities), we pooled them into a single population. This is detailed in the supplementary note, but we now explain this in the legend too.

Reviewer 2 comment 11: Figure 5: The genomic coordinates (chromosome + positions) should be marked on the x-axis for each haploblock. Along y-axis, I suggest labeling the GWAS analyses (“North America” to “Montpellier”) and gene expression comparisons differently to improve readability. In addition, Mississauga and Lafayette gardens were merged in group “North America”, while the result of group “Europe” (Extended Data Fig. 9) that summarized Uppsala and Montpellier gardens were not included here. Is there any reason to exclude it?

Response: We are not able to add genomic coordinates to the figure without substantially increasing the figure size (and decreasing readability). But we have now included new text within the legend: “HB7a1, HB7a2, and HB7b are all found on chromosome 7, while HB9 is on chromosome 9 and HB13 is on chromosome 13. Position on the chromosome is found beneath differential expression heatmaps for each chromosome.” We now bolded the various DEG comparisons on Fig. 5 to make them stand out from the GWAS groupings. The reason to exclude Europe is the combined analysis of Uppsala and Montpellier datasets had no fitness-associated peaks in the GWAS (Extended Data Figure S9)

Reviewer 2 comment 12: Extended Data Fig. 3B: Please clarify that the pie charts above map inserts represent the cultivars, and that each cultivar can be originated from an area (dotted lines) or a population (a pie chart). For a number of cultivars, the arrows are not pointed to a clear location (e.g., LA-S1), please correct.

Response: Please note the Comment 25 from Reviewer 1 that explains our detailed revisions to this legend and figure.

Reviewer #3 (Remarks to the Author):

In this manuscript, Battlay and colleagues report on a comprehensive investigation of evolutionary and genetic drivers of invasion success in a cosmopolitan plant. ...

The manuscript has the potential to become a key reference in invasion genetics, because it addresses a fundamental question on the success of invasive species, it relies on extremely thorough data and analyses, and it has a very high standard of writing and overall presentation.

Response: Thank you for your interpretation and effort.

Major comments:

Reviewer 3 comment 1: My main concern is that the manuscript places too much emphasis on the five structural variants relative to variants elsewhere in the genome. For example, on lines 468-471, the authors mention that “while there were clearly many regions across the genome with evidence of selection, the strongest and most parallel signatures of adaptation come from just a few haploblocks that also exhibit classic genomic signatures of structural rearrangements”. However, unless I am misinterpreting these results, Fig. 2 and Extended Data Fig. 7 indicate that most parallel and non-parallel outlier regions occur in non-haploblock genomic regions. For example, in contrasts between Europe and the other ranges (Extended Data Fig. 7A), only two of the five comparisons showed a significant enrichment of outlier windows in haploblocks (lines 353-356). Thus, across extensive parts of the invasive range of this species (Oceania, China, Japan), haploblocks do not seem to show disproportionately stronger evidence of selection.

Along the same lines, I think Fig. 2 would benefit from an additional panel that focuses only on parallel outlier windows and shows (perhaps as a piechart) the breakdown of haploblock and non-haploblock repeated outlier windows. If this interpretation is correct, the manuscript would need to be revised to make it more balanced, and to reflect the large relevance of the rest of the genome as well.

Response: It is true that most outlier windows fall outside haploblock regions. However, given the size of our dataset, the extent of the analyses conducted and the limited space to report and discuss results in this manuscript, we have chosen to focus on the most striking feature of our results—the haploblocks.

Local adaptation (XtX-EAA) scans in all ranges are enriched in haploblock regions; it is the contrast scans which are only enriched in North and South America. We have now clarified and expanded upon this in the results:

Line 364-368: “Despite covering less than 2% of the genome, haploblocks contain respectively 14.8% and 6% of outlier windows for XtX-EAA and contrast scans. This represents a significant

enrichment for XtX-EAA scans in all ranges (hypergeometric test $p \leq 0.028$) and for contrast scans between the native range and North and South America”

We also note haploblock enrichment becomes more striking when we consider parallel outlier windows:

Line 369-372: “29% and 10% of parallel windows (i.e. windows that were outliers in more than one range) for XtX-EAA and contrast scans, respectively, were found within haploblocks, marking a substantial enrichment in these regions relative to the rest of the genome (hypergeometric test $p \leq 9.09e-16$).”

Figure 2 already shows non-haploblock regions—the haploblock regions are in blue, non-haploblock regions in black. We have clarified this in the legend for Fig. 2:

“Blue portions of bars in A and B correspond to genomic windows within haploblocks, while black portions of bars represent non-haploblock regions.”

We also now highlight genome-wide (i.e., not confined to haploblocks) features of our selection scans including the significant correspondence between population-level differentiation (XtX) and climate correlations in all ranges (indicative of adaptation to local climate; (lines 313-317) as well as the remarkable parallelism observed between the selection scans (lines 317-320).

Please also see our response to Reviewer 1 comment 4 for further discussion of this issue.

Reviewer 3 comment 2: Are the five genomic regions structural variants, and do the population genomic signals agree with this interpretation? For example, in Extended Data Fig. 6, I would have expected elevated heterozygosity in the intermediate PC1 group of samples. The authors mention that indeed this was the case (lines 338-340), but no information on statistical significance is provided. Just by looking at the plots, it doesn't seem that this is the case, at least for some of these five genomic regions. Local patterns of LD are also not aligned closely with the inferred boundaries of these structural variants (Extended Data Fig. 6C).

Response: We interpret the five haploblocks as SVs based on their population genomic signatures. We apologize that the increased heterozygosity in the middle clusters was so difficult to see, and as described in response to Reviewer 1 comment 24 we have cropped outliers on the y-axes in EFig. 6A to improve the visualization of clusters and heterozygosity differences. We have also now used a Wilcoxon test to specifically compare each homozygote class with its corresponding heterozygote class; all tests are significant at $p < 0.0003$. Furthermore, we repeated clustering from SNPs called in a single GLUE (Toronto) population with higher sequencing coverage and now include these (much clearer) plots (EFig. 6B). Beyond the issue with hb7a1 which we explained in response to Reviewer 2 comment 5 (it overlaps HB7a2 which complicates the LD analysis), we may not expect clear LD patterns near SV breakpoints because these regions are usually enriched for repetitive sequence (Huang & Rieseberg [2020] *Frontiers in plant science*, 11, 296). However, in the interest of being conservative, we have now changed our title to ‘Haploblocks’ instead of “Structural Variants”

and revised our text throughout to use the term ‘haploblock’ instead of ‘structural variant’, unless we specifically mean ‘structural variant’.

Dear Dr. Domingues,

We have now addressed all of the revisions below within our manuscript. We provide the point-by-point description of our changes below. We thank the reviewers for their effort and comments.

Best,
Nic Kooyers on behalf of all authors.

Reviewer #1:

I am happy to see the revisions of this manuscript. The revisions and complementary analysis make the results stronger, and the discussion is much improved. I think that the additions give the manuscript a broader interest.

A minor note for accessibility to the abstract- L103. I'm not sure haploblock has a clear enough definition to be used in an abstract without further explanation - "large blocks of non-recombining haplotypes"? and L107 "putative SV?" or "putative inversions?" "or putative non-recombining SV"? (One has to keep in mind here (like at the end of the intro) that the local PCA method does target only one specific type of signature (reduced recombination leading to long linked divergent haplotypes).

The editorial staff at Nature E&E has revised our abstract, but we agree that including this definition is useful. Thus, we have included a brief revision in our abstract (L101): *Five large haploblocks – large haplotypes with limited recombination - on three chromosomes exist as standing genetic variation within the native and introduced ranges, and exhibit strong signatures of parallel climate-associated adaptation across continents.*

All of my previous comments have been addressed, and I reckon they were numerous. The authors have done an impressive effort at polishing and checking all analyses. This is a very rich study, including so many analysis, data, and results. I believe it will be a landmark in the study of biological invasions, and repeated adaptation through putative SV.

Best regards,
Claire Mérot

We appreciate your effort and comments.

Reviewer #2:

Remarks to the Author:

This is the revised version of a manuscript I previously reviewed. Overall, I think the authors have done a great job with the revisions and addressed most of the points I raised earlier, as well as those from the other two reviewers, many of which overlapped, especially regarding the assignment of cultivars and introduced samples, the quality of the SNPs on a duplicated reference genome, and the local population structure patterns used to identify haploblocks.

Thank you.

I only have a couple comments:

- Despite the detailed description of the “invasion”/introduced history of *Trifolium repens*, it is still unclear whether the introduced populations in North America, South America, South Africa, Australia, Japan and China originated from “domesticated” white clover or wild strains. It is to be regretted that with such a wide span of sampling of native, introduced and cultivar samples on all continents, the evolutionary history (domestication, introgression, etc.) of white clover remains unsolved. If the domesticated/historical cultivars/landraces and true wild populations are indistinguishable, please clarify this in the main text to avoid confusion.

We added an additional statement that clarifies the introduction history (L120): *“Introductions to North and South America, South Africa, Australia, Japan and China occurred by the late 1800s via European colonial expansion and likely involved both landraces and wild accessions.”*

- Related to the above problem, I am unsure whether *Trifolium repens* should be treated as an invasive species, as stressed in the first several paragraphs of the main text. Whether the accessions found in other continents are accidental introductions from the wild with costly impacts on ecosystems, agriculture, and human health (similar to *Ambrosia*)? Or are they beneficial domesticated strains intentionally distributed and planted as forage and cover crop? Not that this challenges the study’s significance in evolutionary biology, but the focus of the manuscript (including the title, key words, etc.) may need some adjustment.

We have addressed this issue frequently publishing on white clover, and prefer to keep the language as written. We note that white clover is listed as an invasive species in various databases (e.g. Global Invasive Species Database) – but this does not really cover the issue. The definition of invasive species depends on the stakeholder. Our definition of invasive species is a non-native species that expands its range from the site of its original human-mediated introduction with the potential to cause harm.

The introduction of white clover as a forage and cover crop was human-mediated and intentional. However, subsequent spread and naturalization has occurred nearly everywhere that white clover has been introduced. Community and ecosystem consequences are more difficult to evaluate, but white clover certainly has the potential to change communities and population dynamics of other species.

- Line 539-540: “native across Eurasia” -> “native to Europe and western Asia”; Line 547: “in Eurasia” -> “in Europe and western Asia”.

Done.

Reviewer #3:

Remarks to the Author:

The authors have thoroughly addressed most of my comments. However, the contribution of non-haploblock variants could be highlighted more strongly. For instance, the authors could note that, while haploblocks show significant enrichment for outlier SNPs, the majority of inferred

outlier loci actually lie outside these regions. As a result, statements like the one in the abstract's conclusion—'Our results provide strong evidence that large-effect structural variants underlie rapid and parallel adaptation of an introduced species throughout the world'—do not fully align with the findings. A more precise statement might be: 'Our results provide strong evidence that large-effect structural variants contribute significantly, though not predominantly, to rapid and parallel adaptation of an introduced species throughout the world.'

That said, I appreciate why the authors opted to center the manuscript on haploblocks, as this focus delivers a tighter, more compelling narrative. I have no additional comments. Congratulations to all authors on a valuable contribution.

We understand the reviewer's critique here and agree that non-haploblock variants could indeed be important too. We have adjusted our language in the abstract to read: Our results provide strong evidence that large-effect structural variants contribute significantly to rapid and parallel adaptation of an introduced species throughout the world.